# Deep neural networks divide and conquer dihedral multiplication

**Sihui Wei** [* 1 2] **Gavin McCracken** [* 1 2] **Gabriela Moisescu-Pareja** [1 2] **Harley Wiltzer** [1 2] **Doina Precup** [1 2 3]
**Irina Rish** [4 2] **Jonathan Love** [5]

## Abstract

We find multilayer perceptrons and transformers both universally learn an instantiation of the same divide-and-conquer algorithm that requires only a logarithmic number of neural representations to solve dihedral multiplication. Clustering neurons based on similar activation behaviour reveals remarkably clear structure: each neural representation corresponds to a Cayley graph. To our knowledge, this is the first work that fully characterizes and describes all neural representations that are learnable on a dataset, while prior work on group multiplications studied neuron-level behavior, or preliminarily investigated cluster behavior. Thus, we can understand the algorithm networks universally learn at three levels of abstraction: 1) Neurons activate on coset or approximate coset structure of the dihedral group. 2) Groups of neurons together form neural representations that act to divide the dataset into different subproblems, being Cayley graphs, where the equivalence class of the answer is computed. 3) The global algorithm then linearly combines each neural representation (subproblem) together at the logits. This work provides the community with a deep case study and a well-understood toy model for interpretability, and makes progress toward proving the conjecture that networks trained via stochastic gradient methods divide and conquer all group multiplication tasks.

## 1. Introduction

One of the most important open problems in deep learning is the quest to create interpretability tools that are general,

---

[*]Equal contribution [1]McGill University [2]Mila – Quebec Artificial Intelligence Institute [3]Google DeepMind [4]Université de Montréal [5]Leiden University. Correspondence to: Sihui Wei <sihui.wei@mail.mcgill.ca>, Gavin McCracken <gavin.mccracken@mail.mcgill.ca>.

*Proceedings of the 43rd International Conference on Machine Learning*, Seoul, South Korea. PMLR 306, 2026. Copyright 2026 by the author(s).

and result in insights that apply across different random seeds, architectures, and ultimately, over different training data. Ubiquitous to this goal is the universality hypothesis—that deep neural networks (DNNs) trained on different (but related) datasets will make use of *similar algorithmic strategies* (Li et al., 2015; Olah et al., 2020). If true, there's hope for researchers to discover methods to automatically detect circuits in large models, which would be a great advancement towards safe, interpretable artificial intelligence.

Unfortunately, little is understood about universality: 1. We aren't confident that the hypothesis is true. Chughtai et al. (2023a) investigated universality by studying DNNs learning cyclic, dihedral, and permutation group multiplications and claimed DNNs learn a universal algorithm for all three tasks. They claimed a DNN trained on any group multiplication would learn their universal algorithm, but Stander et al. (2024) showed this wasn't true on permutation groups, and McCracken et al. (2025b) showed it wasn't true on cyclic groups. Additionally, Zhong et al. (2023) claimed two totally different circuits were learned in transformers, but Moisescu-Pareja et al. (2025) resolved that, proving only one circuit was learned. 2. We understand little about the *nature* of universality. Moisescu-Pareja et al. (2025) showed that some architectures learn topologically and geometrically different neural representations, but proved a theorem that all representations were either a universal manifold, or linear projections of the universal manifold. Relatedly, McCracken et al. (2025b) conjectured DNNs learn a divide-and-conquer algorithm to solve any group multiplication.

Following the scientific method, our work revisits all prior theories using the dihedral group to test which theories make testable predictions outside their original settings. We find that DNNs learn approximate cosets (confirming Stander et al. (2024)) to divide and conquer dihedral group multiplication (confirming McCracken et al. (2025b)). Additionally, we make a novel discovery by learning that the universal manifold described by Moisescu-Pareja et al. (2025) is a *minimal Cayley graph*. Our contributions are:

1. *A new example of universality across datasets.* We find neurons in DNNs trained on the non-commutative dihedral group multiplication learn either *coset* structure or *approximate coset* structure. This finding means DNN neurons

universally learn approximate cosets to solve the permutation group Stander et al. (2024), cyclic group (McCracken et al., 2025b) and dihedral group multiplications.

2. *A novel perspective.* The neural representations of DNNs trained to generalization on dihedral multiplication (and modular addition) tasks encode Cayley graphs.

3. *The neural representations divide and conquer.* These Cayley graphs act as simpler computational subproblems to help the DNN compute the correct answer. Indeed, we find $\mathcal{O}(\log(n))$ neural representations are learned, implying a very efficient divide-and-conquer algorithm is learned.

To our knowledge, this is the first work to characterize the classes of neural representations that can be learned across random seeds and architectures and interprets DNNs at three levels of abstraction: 1. the neuron level 2. neural representation level and 3. the global algorithm level.

## 2. Background

The *dihedral group* $D_n$ is the set of symmetries of a regular $n$-gon, containing $2n$ elements: $n$ rotations $r^k$ for $k \in \{0, \ldots n-1\}$ that rotate the $n$-gon by $2\pi/n$ radians, and $n$ reflections $sr^k$ reflecting about $n$ distinct axes. The rotation $r^0$ is the *identity* element, denoted $e$, for which $ex = xe = x$ for any

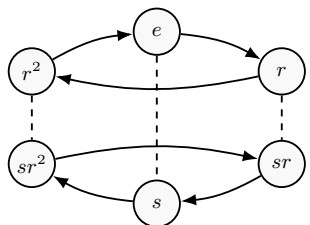

*Figure 1.* A $D_3$ Cayley Graph. Solid arrows apply left multiplication by $r$; dashed lines apply left multiplication by $s$. Note non-commutativity $r \cdot s \neq s \cdot r$.

$x \in D_n$. These operations form a non-commutative group multiplication when $n \geq 3$, meaning the order in which operations are multiplied matters, for instance, $sr \neq rs$. **Group multiplication**, $a \cdot b = \mathcal{C}$, $a, b \in D_n$ involves composing two symmetries in sequence (from right to left):

$$r^a r^b = r^{(a+b) \bmod n}, \qquad sr^a r^b = sr^{(a+b) \bmod n},$$
$$r^a sr^b = sr^{(b-a) \bmod n}, \quad sr^a sr^b = r^{(b-a) \bmod n}. \quad (1)$$

**Cayley graphs** geometrically encode a group's structure. The Cayley graph of $D_n$ may be expressed via a *generator set* $\{r, s\}$, where nodes are group elements and edges are labeled by $\{r, s\}$. Particularly, an edge labeled $x \in \{r, s\}$ between nodes $a, b$ exists if $b = xa$. A Cayley graph for $D_3$ is depicted in Fig. 1.

**Group Fourier Transform (GFT).** Analogously to the classical Fourier transform, the group Fourier transform (GFT) decomposes a complex-valued function on a finite group $G$ into components indexed by the group's unitary irreducible representations (irreps); see Appendix A.2. In this paper we index the GFT components by the standard

irreps of $D_n$: $\mathrm{triv}$ is the *DC* component (unchanged by all group elements), i.e., $r \mapsto +1$, $s \mapsto +1$; sign flips under reflections but not rotations, i.e., $r \mapsto +1$, $s \mapsto -1$; For even $n$, rp flips under the rotations but not reflections ($r \mapsto -1$, $s \mapsto +1$), and srp flips under both ($r \mapsto -1$, $s \mapsto -1$); Each $2D_k$ (2D_k) corresponds to a cosine–sine pair at angular frequency $\frac{2\pi k}{n}$: rotations rotate this pair, and reflections flip it, $k = 1, \ldots, \lfloor \frac{n-1}{2} \rfloor$.

**Cosets / approximate cosets.** Given a subgroup $H \leq G$, the left cosets $gH$ all have the same cardinality and form a partition of $G$. In our setting, this corresponds to the case where a learned "frequency" is well aligned with the group structure, so that the neuron's response is (approximately) constant on each coset block. By contrast, when the frequency does not divide the group size (for example, attempting to "divide" the dihedral group $D_{18}$ by 5), no such exact partition exists and each element is in its own coset. We refer to this pattern of responses as an *approximate coset*.

## 3. Related work

Group multiplication tasks have emerged as standard evaluation settings across both the mechanistic interpretability community (Nanda et al., 2023; Chughtai et al., 2023b;a; He et al., 2024; Tao et al., 2025; Doshi et al., 2024; Stander et al., 2024; McCracken et al., 2025a) and theoretical interpretability community (Gromov, 2023; Morwani et al., 2024; Mohamadi et al., 2023; McCracken et al., 2025b). These tasks have served as a shared foundation for empirically and theoretically oriented researchers to test and validate their hypotheses. In particular, studies on group multiplication have become central to examining the *Universality Hypothesis* (Li et al., 2015; Olah et al., 2020; Chughtai et al., 2023a; Huh et al., 2024), which asserts that neural networks trained on related problems tend to develop internal structures that are similar. The hypothesis further suggests that, independent of factors such as model architecture, initialization, or training setup, deep neural networks will discover comparable representational mechanisms grounded in common underlying principles.

The first interpretability work on modular addition offered an algorithmic explanation for how transformer models learn the task (Nanda et al., 2023). Then Chughtai et al. (2023a) conducted a broader empirical test of the Universality Hypothesis by extending the analysis to both cyclic and permutation group multiplication tasks claiming DNNs consistently developed internal mechanisms equivalent to learning matrix representations of groups and performing matrix multiplications to compute results. However, subsequent research challenged these conclusions. Zhong et al. (2023) claimed to demonstrate that distinct transformer architectures learned two fundamentally different computational

circuits. Then, Stander et al. (2024) failed to reproduce Chughtai et al. (2023a)'s findings, instead identifying coset circuits as the structures underlying models trained on permutation group tasks. Consequently, the initial claims of universality were undermined—modular addition appeared to admit two incompatible circuit interpretations and neither matched the coset-based mechanism observed for permutation groups, a related setting.

More recently, these inconsistencies were resolved. Mc-Cracken et al. (2025b) provided a theorem that DNNs trained on modular addition universally use approximate coset structures to implement the Chinese Remainder Theorem. Shortly after, Moisescu-Pareja et al. (2025) was unable to reproduce the findings of Zhong et al. (2023), which had suggested the existence of two distinct circuit types. Resultantly, empirical and theoretical evidence converged to show DNNs learn a universal algorithm to solve modular addition. Furthermore, a universal principle was discovered, DNNs used approximate cosets to learn both modular addition and the permutation group. Thus, McCracken et al. (2025b) conjectured that DNNs would use approximate cosets to learn all group multiplications, which remains an open question our work steps toward answering, by closing the question on dihedral groups.

## 4. Methodology

We train one- and two-layer perceptrons (MLPs) to 100% test accuracy using 512 ReLU neurons per layer, with separate embedding matrices for the left and right factors $a$ and $b$, on dihedral multiplication pairs $(a, b) \in D_n \times D_n$ for $n \in \{18, 19\}$. We also train one- and two-layer transformers by casting dihedral multiplication as a length-two language modeling task: given the token sequence $(a, b)$, predict the output token $a \cdot b$. These transformers use a single shared token embedding and include positional embeddings and residual connections. Training pairs are sampled uniformly from $D_n$ in both architectures. We focus on $D_{18}$ in the main text because its richer coset structure (relative to odd or prime $n$) yields a clearer contrast between coset and approximate coset preferences in Fig. 4.

We cluster neurons by identifying all units that activate on the same Fourier basis with the GFT; for each neuron in the cluster we build a $2n \times 2n$ matrix whose entry $(a, b)$ is the neuron's preactivation on datum $(a, b)$, flatten each matrix, and stack the resulting vectors to form a $|\text{cluster} f| \times (2n)^2$ "cluster of preactivations" matrix. We then perform principal component analysis (PCA) on this matrix of neuron preactivations and project all $(2n)^2$ data $(a, b)$ onto the principal components (PCs). When cosets are learned, data in the same joint equivalence class—e.g., all points with $(a \equiv 0 \pmod 3, b \equiv 0 \pmod 3)$—collapse to the same coordinate. For example, in $D_{18}$ there are $36^2 = 1296$ points,

but for neurons learning $2D_6$ (corresponding to learning frequency 6 and since $\gcd(6, 18) = 6 \neq 1$) only 36 points are plotted (Figure 3 (A)). These 36 points correspond to the 36 joint equivalence classes. For each fixed $a$, there are 6 points determined by $b$: three with $b < 18$ (rotations) and $b \equiv 0, 1, 2 \pmod 3$ and three with $b \geq 18$ (reflections) and $b \equiv 18, 19, 20 \pmod 3$; e.g., for $a \equiv 0 \pmod 3$ one may take $b \in \{0, 1, 2\}$ and $b \in \{18, 19, 20\}$ as representatives.

## 5. Results

Here, we demonstrate what's learned by the network at three levels of abstraction. The preactivations of three *individual* neurons are presented in Fig. 2 with their group Fourier transform, two neural representations are presented in Fig. 3, one for an approximate coset, and one for a coset, and finally, the universally learned algorithm is presented in Fig. 7. The remaining main results quantitatively demonstrate what 1-layer multilayer perceptrons (MLPs) and transformers learn over 1000 random seeds, and over different sizes of the problem, *i.e.* different $n$ in $D_n$.

**Setup.** We focus on the dihedral group on 36 elements, $D_{18}$. Let elements 0 to 17 be rotations $r^k$ and elements 18 to 35 be reflections $sr^k$; Note: rotations $r^k$ are in the sign $+1$ region of a Cayley graph and reflections $sr^k$ are in the sign $-1$ region of a Cayley graph. The following definitions are adapted from McCracken et al. (2025b).

**Definition 5.1** (Step size). $d := \left(\frac{f}{\gcd(f,n)}\right)^{-1}$ $\pmod{\frac{n}{\gcd(f,n)}}$, where the modular inverse is used.

**Definition 5.2** (Remapping: frequency normalization). Consider the function $h(x) = \cos\left(\frac{2\pi f x}{n}\right)$ with frequency $f$. We define a new function $g$, allowing us to perform something analogous to a change of variables using the step size $d$: $g(d \cdot x) = h(x) \iff g(x) = h(d^{-1} \cdot x)$.

**Individual neurons.** Fourier analysis gives that all functions on dihedral groups are composed by projections of dihedral irreps. Since GFTs of neuron activations concentrate on one irrep, this gives $\frac{n+1}{2} + \frac{3}{4}\left(1 + (-1)^n\right)$ possible neuron classes. We select three example neurons from different irrep classes in Fig. 2, to show that neurons learn rotation generators (step size $d$) and reflection generators (different phase shifts of the sine function on rotation vs. reflection data).

**Important note.** Although one can fit the first-layer neuron preactivations by treating each quadrant independently by $y(a, b) \approx \sum_{q \in \mathcal{Q}} Q_q(a, b)\left[\beta_{0,q} + A_q^{(a)} \sin(\omega_{\text{dom}} a_0 + \phi_q^{(a)}) + A_q^{(b)} \sin(\omega_{\text{dom}} b_0 + \phi_q^{(b)})\right]$, where $\mathcal{Q} := \{\text{BL}, \text{BR}, \text{TR}, \text{TL}\}$ (bottom-left, bottom-right, top-right, top-left). This introduces substantially more degrees of freedom (separate amplitudes/phases per quadrant)

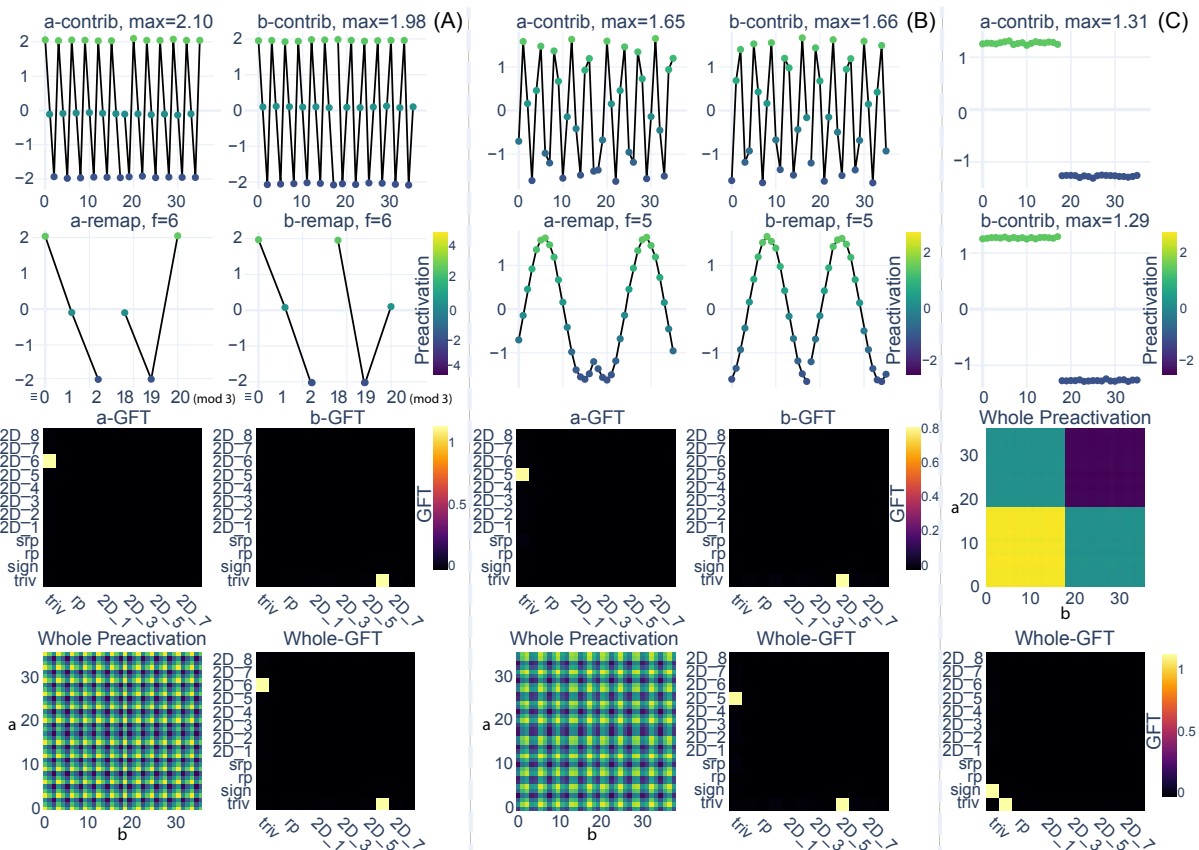

*Figure 2.* DNN trained on $D_{18}$. **(A)** a frequency 6 neuron learns precise cosets; $a$-contrib and $b$-contrib take values on three discrete level sets. Remapping collapses the points in row 1 onto six cosets of $D_{18}$ in row 2; three are in rotation class ($x < 18$): $x \equiv 0, 1, 2 \pmod 3$, and three are in reflection class ($x \geq 18$): $x \equiv 18, 19, 20 \pmod 3$, $x \in \{a, b\}$. All GFTs show concentration on one irrep ($2D_6$). **(B)** The same plots are shown for a neuron learning frequency 5 giving approximate cosets because $\gcd(18, 5) = 1$—there are no precise equivalence classes. **(C)** A neuron learned the sign +1 coset: it is an indicator for when $a$ and $b$ are rotations (sign +1 of the group). See D.2 for all possible neurons.

and can obscure the global structure. Instead, we use a single *coupled* first-order variable model at the dominant frequency $f_{\text{dom}}$, fit jointly over the full grid:

$$y(a, b) \approx \beta_0 + A_1^{(a)} \sin(\omega_{\text{dom}} a_0 + \phi_1^{(a)}) + I_a A_I^{(a)} \sin(\omega_{\text{dom}} a_0 + \phi_I^{(a)}) + A_1^{(b)} \sin(\omega_{\text{dom}} b_0 + \phi_1^{(b)}) + I_b A_I^{(b)} \sin(\omega_{\text{dom}} b_0 + \phi_I^{(b)}),$$

where $a_0 = a \bmod n$, $b_0 = b \bmod n$, $\omega_{\text{dom}} = \frac{2\pi f_{\text{dom}}}{n}$, and the half-indicators are

$$I_a(a) = \begin{cases} +1, & a < n, \\ -1, & a \geq n, \end{cases} \qquad I_b(b) = \begin{cases} +1, & b < n, \\ -1, & b \geq n. \end{cases}$$

This parameterization captures the shared sinusoidal signal while allowing a minimal, interpretable correction between the rotation and reflection halves on each variable.

First-layer neuron preactivations in both architectures are well-approximated by coupled first-order sinusoids fit over the full grid ($R^2_{avg} = 0.9988$ across architectures; Appendix D.1, Table 5).

**Neural representations encode Cayley graphs.** A key consequence of this coupling is that each variable admits

*two* effective phases at the dominant frequency $f_{\text{dom}}$: one associated with the rotation half $\phi_+^{(a)}$ when $I_a(a) = +1$ and one with the reflection half $\phi_-^{(a)}$ when $I_a(a) = -1$, rather than independent phases per quadrant (and analogously $\phi_+^{(b)}, \phi_-^{(b)}$).

Empirically, over 1000 runs we find two relations are almost always learned on the phases. The $a$ phases sum to a constant, and the difference between $b$ phases is a constant: $A_{\text{sum}} = (\phi_{-, f_{\text{dom}}}^{(a)} + \phi_{+, f_{\text{dom}}}^{(a)}) \bmod 2\pi$ and $B_{\text{dif}} = (\phi_{-, f_{\text{dom}}}^{(b)} - \phi_{+, f_{\text{dom}}}^{(b)}) \bmod 2\pi$. This is confirmed quantitatively by ensuring a strict concentration threshold ($\geq 0.999$) is exceeded (in 99.81% of cases it is on average). The concentration level measures how strongly neurons within the cluster agree on the same $A_{sum/dif}$ or $B_{sum/dif}$, with a number near 1 meaning almost all neurons match closely, while near 0 means they disagree and spread out (Appendix A.5). Whereas other cases are essentially absent ($A_{\text{dif}}$: 0.00%, $B_{\text{sum}}$: 0.03%). Thus, in typical clusters these two quantities concentrate sharply to a common con-

stant within a cluster, showing stable input $a$ and input $b$ phase transformations when switching between the rotation/reflection halves (see Appendix C for the dihedral interpretation under the $sr^k$ convention).

**These phase relations are because every neuron has the same generators within a cluster.** To get the generators from the phase relations, we substitute $A_{\mathrm{sum}}$ or $B_{\mathrm{dif}}$ into $\Delta\phi$ to get the reflection generator index $k$ via $k \equiv q^{-1}\big(\big[(\pi - \Delta\phi)\frac{g}{2\pi}\big] \bmod g\big) \pmod g \in \mathbb{Z}_g$, $h = \gcd(n, f_{\mathrm{dom}})$, $g = \frac{n}{h}$, $q = \frac{f_{dom}}{h}$.

When both $A_{\mathrm{sum}}$ and $B_{\mathrm{dif}}$ pass the $0.999$ concentration threshold, mapping their concentrated constants to a reflection index yields the same $k$ in $100\%$ of cases, i.e. both variables recover the same consistent reflection generator. This gives that in $99.81\%$ of runs, Cayley graphs have one reflection and one rotation generator for $a$ and $b$.

The caption of Fig. 3 walks through the geometric structure of a neural representation of frequency 6 neurons in row (A). The goal is to understand that data points in layer 1 representations are spaced out based on the equivalence class of the $a$ and $b$ inputs and whether the input is a rotation or reflection. In downstream neural representations (layers $> 1$ or at the logits), the data points are spaced out based on the equivalence class of $\mathcal{C}$ and whether $\mathcal{C}$ is a reflection or rotation element.

For the frequency-6 neurons in row (A), coloring by $\mathcal{C} \bmod 3$ does not yet reveal clear separation, indicating that layer-1 representations are still organized primarily by the input coordinates $a$ and $b$, rather than by $\mathcal{C}$. This $\mathcal{C}$-level organization emerges more clearly downstream: PCA of the logit contributions shows six $C_3$ structures, with the three reflection-related $C_3$s orthogonal to the three rotation-related $C_3$s. Thus, the representation already distinguishes whether $\mathcal{C}$ is a rotation or reflection, while narrowing the answer to one of three candidates within that type.

For the frequency-5 representation in the second row of Fig. 3, the cluster contributions to logits show the same qualitative pattern: reflection $\mathcal{C}$s and rotation $\mathcal{C}$s are stored in orthogonal subspaces.

At this point, we've qualitatively shown that each vertex on the Cayley graphs contains all points in the same coset equivalence class. Appendices D.2 and D.3 qualitatively show every possible type of neuron and the Cayley graph corresponding to their neural representation.

We now test this quantitatively with transformers and MLPs over many random seeds. We test if the PCA manifold is *consistent with* the Cayley graph on the recovered variable-specific generators by checking three graph-theoretic predictions that should hold if PCA preserves the generator-induced adjacency and cyclic actions:

(i) *Coset clustering score and disjointness*:

1. **Coset clustering score $s_{\mathrm{cluster}}$.** We compute the mean distance between points within the same coset and the mean distance between points in different cosets. We define

$$s_{\mathrm{cluster}} = 1 - \frac{\mathbb{E}[\text{distance between points in the same coset}]}{\mathbb{E}[\text{distance between points across cosets}]},$$

so that larger values indicate tighter and better-separated coset clusters.

2. **Coset disjointness.** For each coset, we compute the fraction of other cosets whose enclosing balls are strictly disjoint from it; we report the mean fraction across cosets.

They measure vertex identifiability in PCA space: each vertex should appear as a compact cluster, and different vertices should remain well separated.

(ii) *Cyclic subgroup traces.* Repeatedly applying the recovered rotation generator along a fixed input variable should produce a coherent cycle in the embedding. We measure cyclic regularity by the Spearman correlation $|\rho|$ between step index and cumulative polar angle (Appendix A.6) and by the number of self-intersections of the closed polygonal trace.

(iii) *Global generation via closure.* If the two input variable-specific generator pairs generate the full product group $(\mathbb{Z}_{2g})^2$, then alternating closure under the $a$-variable and $b$-variable generator pairs should reach all vertices; we report the resulting coverage fraction.

Averaged across 1000 seeds, using the resulting variable-specific rotation and reflection generator pairs for $a$ and $b$, the PCA embedding supports the intended two-input graph structure: generator-induced neighbor relations remain well separated ($s_{\mathrm{cluster}}$ and disjointness score in Table 1); rotation traces form coherent loops with near-monotone angular progression ($|\rho| = 0.9999 \pm 0.0001$) and almost no self-intersections ($0.0031 \pm 0.0072$); and nested closure reaches $100\%$ of vertices in $(\mathbb{Z}_{2g})^2$. Randomly permuting the cluster index (while preserving vertex presence) destroys these signatures ($|\rho| = 0.0667 \pm 0.0099$, self-intersections $25.4125 \pm 18.4687$). Together, these results show that the PCA geometry is well explained by the Cayley graph defined by the recovered generators, rather than arising from a trivial cyclic artifact.

*Table 1.* $s_{\mathrm{cluster}}$ and disjointness statistics (mean $\pm$ std); 1k seeds.

| Arch | $n$ | $s_{\mathrm{cluster}}$ | Disjointness |
|---|---|---|---|
| MLP | 18 | $0.993 \pm 0.001$ | $0.999 \pm 0.001$ |
| Transformer | 18 | $0.974 \pm 0.071$ | $0.977 \pm 0.082$ |

We train 1000 MLPs and transformers on $D_{18}$, and record the rate of occurrence of the different Cayley graphs (Fig. 4).

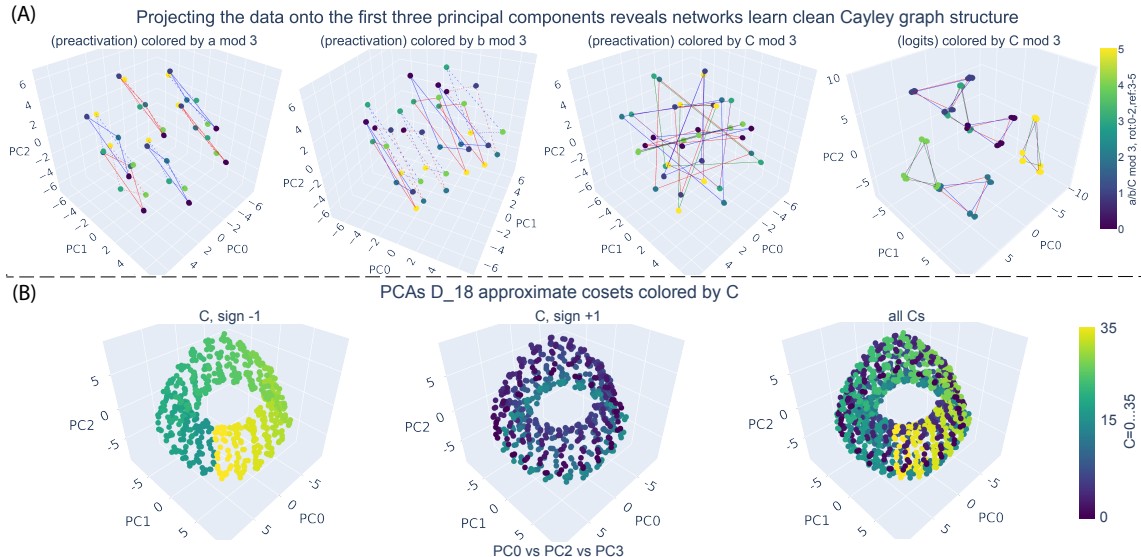

*Figure 3. Cayley graphs revealed.* Data points are colored by their coset equivalence classes for $a, b, \mathcal{C}$; each vertex is all data points in the same coset equivalence class, so tightly clustered that they appear as one point. **(A)** PCA of the preactivations of the frequency-6 neuron cluster shows a DNN decomposed $D_{18}$ into six $D_3$ Cayley graph subproblems. Coloring by the coset of $a$ (column 1) or $b$ (column 2) produces six $D_3$. Note a $D_3$ connected by red edges means $b$ is a reflection and a $D_3$ connected by blue edges means $b$ is a rotation; furthermore, dashed lines mean $a$ is a reflection and solid lines mean $a$ is a rotation. The plotted line segments are obtained by iteratively applying the rotation generator in PCA space to move between coset clusters along each input variable until the cycle closes; the reflection generator is not drawn here, since it is recovered from the half-phase relations. There is one $D_3$ Cayley graph for each variable, replicated across the other's six coset clusters, with two generators per input. In the third column, coloring by $\mathcal{C} \bmod 3$ does not produce clearly separated structures. This suggests at this stage, the representation is organized mainly around the input coordinates $a$ and $b$ respectively, with $\mathcal{C}$-level structure emerging only downstream in the next layer or logits (column 4). The logits space shows six well-separated cosets–three corresponding to answers with sign $+1$ and three with sign $-1$. **(B)** PCAs of a frequency-5 (approximate coset) cluster's contributions to the logits, show the network stores information in sign -1 orthogonally to information in sign +1.

On $D_{18}$, we see DNNs have a preference for learning Cayley graphs that correspond to smaller subproblems, *e.g.* MLPs learn frequency 6 giving $D_{3_6}$ in 87.7% of the runs. We find smaller Cayley graphs (dark blue) are mostly preferred to full-sized Cayley graphs (green).

Furthermore, we know the rotation generator for each Cayley graph is the frequency, and in Fig. 5 we show the reflection generators are uniformly likely across random seeds, and this is true for transformers as well (see Appendix D.4). Finally, we explore how many neural representations are learned as a function of the size $n$ of the dihedral group and find both transformers and MLPs learn a logarithmic number ($\mathcal{O}(\log(n))$) of Cayley graphs (Fig. 6).

### 5.1. The global algorithm DNNs universally learn

We now present the algorithm DNNs universally learn. We know every neuron activates individually on an approximate coset, and we know every neural representation in a trained DNN encodes a Cayley graph. Thus, we give Algorithm 1.

*Remark* 5.3. *The sinusoidal neuron-based Cayley-graph decomposition.* The dihedral group $D_n = \langle r, s \mid r^n = e, s^2 = e, srs = r^{-1} \rangle$ has two types of generators: rotation $r$ and reflection $s$. Multiplication splits into four quadrants depending on the pair of input types (rotation or reflection).

---

**Algorithm 1** The abstract algorithm DNNs universally learn

1. Construct $\mathcal{O}(\log n)$ neuron clusters, each realizing a Cayley graph $C_k$ generated by either $\{r\}$ or $\{r, s\}$.
2. For each cluster $k$, use step size $d_k$ to realize the generator $r$, with phases chosen appropriately to realize $s$ when $C_k$ is generated by $\{r, s\}$. Use $C_k$ to separate input pairs $(a, b)$ according to the equivalence classes induced by $C_k$.
3. After layer 1, transform the representation from Cayley graphs over raw inputs $(a, b)$ to Cayley graphs over the output equivalence class $\mathcal{C}$.
4. With $\mathcal{O}(\log n)$ Cayley graphs in the final layer, the correct logit has logarithmic height, as each graph outputs mass onto all logits in the same equivalence class as $\mathcal{C}$.

---

Empirically, neurons in each layer specialize to sinusoidal features whose preactivations cluster along Cayley-graph structure for either one or two generators. Principal component analysis of these clusters reveals interpretable graphs in which the generators $\{r\}$ or $\{r, s\}$ are directly visible. Thus, each cluster of neurons corresponds to a distinct Cayley graph, implementing one simpler "subproblem" of the global group operation.

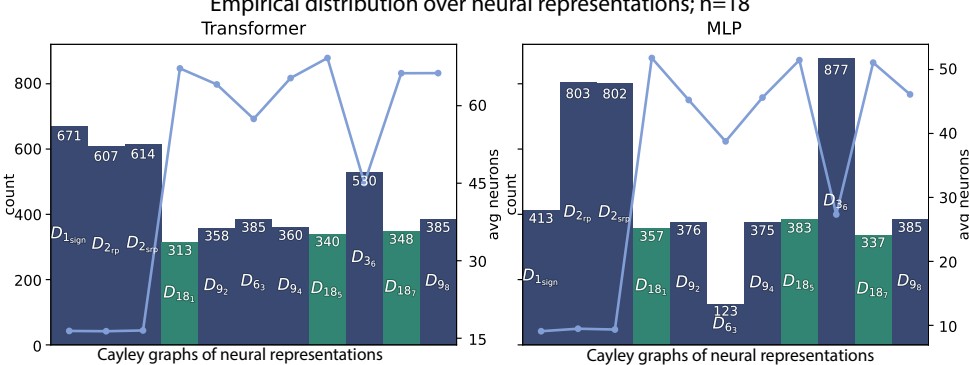

*Figure 4.* We study transformers and MLPs over 1000 random seeds on $D_{18}$. **Barplot.** Number of times each Cayley graph was learned (bar labels are $D_{x_y}$, where $x$ is the number of cosets and $y$ is the irreps index). **Line plot.** Average number neurons used to learn each Cayley graph representation.

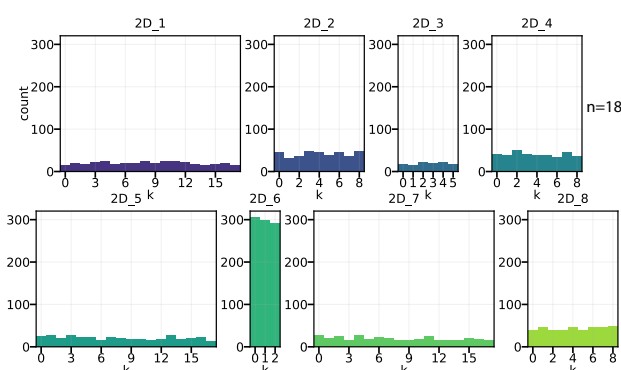

*Figure 5.* Histograms of the number of times each generator ($sr^k$) was learned for each type of Cayley graph in MLPs across 1000 seeds. The distributions are all approximately uniform. This matches our intuitions from group theory that there are no benefits to learning specific generators for the same Cayley graph. Networks seem to just want to learn the Cayley graph, and don't prefer particular generators to be used.

# 6. Discussion and conclusion

We give interpretations that complement and agree at three levels of abstraction. We find: 1) neurons activate on coset or approximate coset structures 2) neural representations act to divide the data manifold into different subproblems (Cayley graphs), each computing the equivalence class of the answer with respect to the generators of the graph 3) the global algorithm merges the computational results of each neural representation to maximize the correct logit with logarithmically many linear combinations of neural representations. Because different representations output their maximum values on different equivalence classes (due to having different generators), the correct logit is uniquely maximized. We empirically find this divide-and-conquer algorithm is remarkably efficient, using $\mathcal{O}(\log(n))$ neural representations. This picture gives insight into the nature of three influential hypotheses about deep learning.

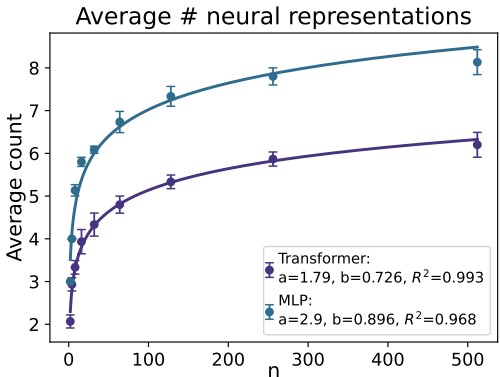

*Figure 6.* DNNs trained over orders of magnitude ($D_2$ to $D_{512}$) with transformers learning fewer neural representations vs MLPs, but both scaling curves are fit well by log functions, providing evidence that a very representation-efficient divide-and-conquer algorithm is universally learned.

## 6.1. Contributing to universality

The universality hypothesis: DNNs trained on similar data make use of similar principles (Li et al., 2015; Olah et al., 2020). McCracken et al. (2025b)'s conjecture only relates to this hypothesis, but our work illuminates two more.

We present three levels at which universality can potentially exist. 1) The fact that individual neurons have been observed to learn approximate cosets across three different group multiplication tasks gives reason to believe that neurons may learn these structures on all group multiplication tasks. Can this be proved formally, or falsified experimentally? 2) Neural representations have been found to be Cayley graphs in the cyclic group and dihedral group, **however,** both of these groups share the cyclic group as their finite simple group. Are neural representations universally Cayley graphs? Other simple groups (*e.g.* permutation, Lie type, etc) should be checked experimentally. Furthermore, can this be proven rigorously? 3) Will DNNs trained on every group multiplication learn a divide-and-conquer algorithm that requires a logarithmic number of neural representations

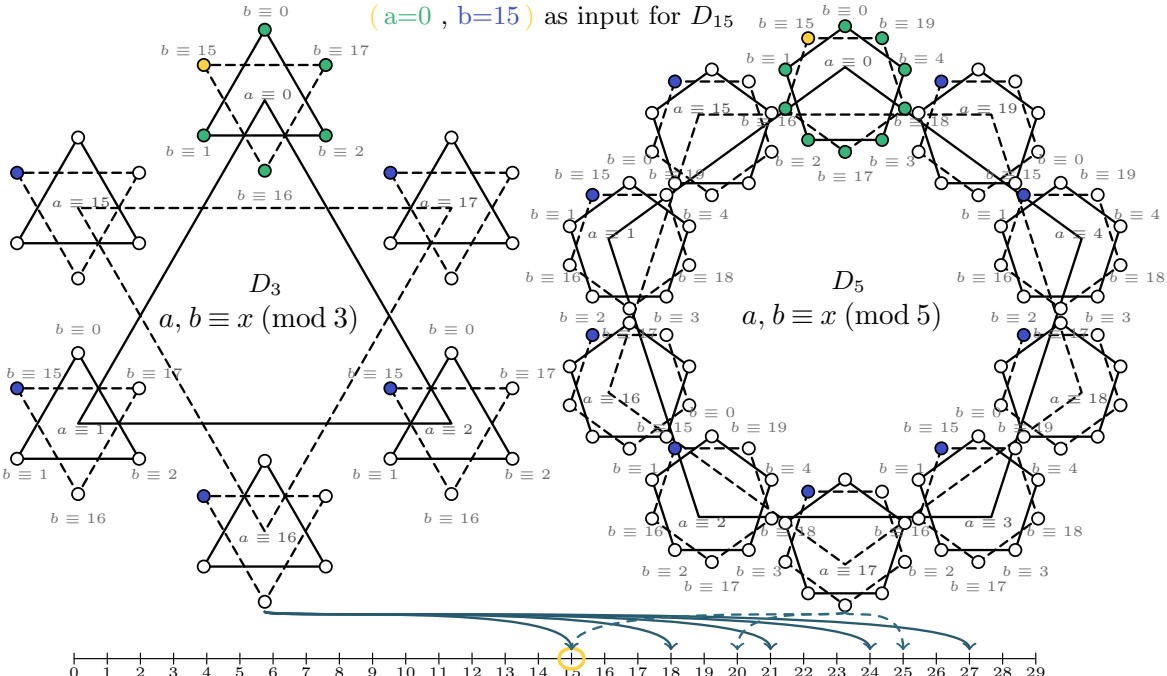

*Figure 7. A DNN breaks $D_{15}$ into simpler subproblems.* We visualize the computation within a DNN with $a = 0, b = 15$ input as it maximizes the correct logit ($\mathcal{C} = 15$) via linear combination of neural representations in the DNN. These are: six $D_3$ subgroups and ten $D_5$ subgroups. The $D_3$ representation positively outputs on logits where $a \equiv 0 \pmod{3}$ and $b \equiv 15 \pmod{3}$ because the network stores information about the cosets of $a$ and the cosets of $b$ independently, narrowing down the cosets containing $\mathcal{C}$. The $D_3$ neural representation outputs maximally on reflection elements 15, 18, 21, 24, 27. Thus, it narrowed down membership of $\mathcal{C}$ to these five elements, all in reflections of $D_{15}$. The $D_5$ neural representation outputs maximum mass onto reflection elements 15, 20, 25. The intersection of these sets is 15, and thus 15 has a higher logit value than any other logit, and the correct answer is selected by argmax.

compared to the size of the group? A formal proof of this would be enormous and make giant leaps toward explaining the efficiency of deep learning, but for the time being this has only been observed, again, on groups sharing the cyclic group as their finite simple group.

## 6.2. Contributing to the manifold hypothesis

The manifold hypothesis: DNNs efficiently learn (despite the curse of dimensionality typically applying to such high dimensional objects) by learning low dimensional manifolds that the dataset lives on (Goodfellow et al., 2016).

Each neural representation corresponds to a manifold (Moisescu-Pareja et al., 2025), but that manifold is a Cayley graph we can describe via its generators. We discover that dihedral groups of size $n$ have $\frac{n+1}{2} + \frac{3}{4}\left(1 + (-1)^n\right)$ classes of possible neural representations. Moisescu-Pareja et al. (2025) saw that some local minima were lower-dimensional linear projections of the universal modular addition manifold. We find the same thing can occur (Appendix D.7), but discover a new type of local minima that didn't exist within modular addition. This occurs when a neural representation can't span the entire group. This means the Cayley graph has < 4 generators. As seen in the results, in the vast majority of cases Cayley graphs have one reflection and one

rotation generator for $a$ and $b$, and thus span the group. The reason this type of local minima didn't exist within modular addition is because every Cayley graph for modular addition must have at least 1 generator, which is enough to span a Cayley graph with coset equivalence classes as vertices.

## 6.3. Contributing to Platonic Representation hypothesis

The platonic representation hypothesis: architectures trained on different objectives learn a platonic representation, with distances between data points in different classes of data "converging" as models get larger (Huh et al., 2024).

The discovery that we can identify the generators of Cayley graphs allows for a direct inspection of the platonic representation hypothesis at the neural representation level. The generators explicitly give how datapoints are distanced in each neural representation that's learned. Since there's a variance in which Cayley graphs are learned across random seeds, this implies that platonic representations aren't learned at the neural representation level. Note, it could still be true that a platonic representation is learned at the algorithm level, but to our knowledge there's no way (yet) to directly inspect it.

**Limitations and future work.** We believe our methodology will transfer to other group multiplications since it works

on both cyclic and dihedral groups. Since both these groups share the cyclic group as their base building block (called a simple finite group), however, it could be that groups composed of different simple groups may require extensions beyond simple application. Another limitation is that we study DNNs trained on pure dihedral group multiplication, and not DNNs trained on a mixture of groups. We see this as a good direction for future work, because it could provide a playground for studying not just the superposition of neurons, but superposition of neural representations as well.

Furthermore, while we explain what the local minima are, which will help with deriving a theoretical proof of every possibility, such a derivation is left to future work. We believe this would be a major contribution if solved, as it would unify theory with empirical interpretation, and provide a solved interpretability model for study.

Our methodology relies on the fact we can use the Group Fourier Transform to identify neuron behaviors. After using it, we gain the ability to form the neuron clusters and study the neural representations qualitatively and quantitatively with great precision. For the time being however, there is no such method available for use on natural datasets. That said, part of the goal of studying toy models is to hopefully, once studying enough of them, gain enough insight into how DNNs work to be able to generalize the community's collective insights onto natural datasets.

Similar to the work of Moisescu-Pareja et al. (2025), we find the solutions learned by transformers are noisier (see Table 1) and we find the neural representations can be lower dimensional (matrix rank) compared to the MLP neural representations in deeper networks. An example of when this happens is when a transformer with 2 or more layers learns the sign neural representation, allowing for the network to learn Cayley graphs that don't have a reflection generator for the other neural representations in layer 1. In the second layer, such a network uses the sign representation to augment the non-sign neural representations from the first layer so that they now understand reflections. We aren't sure whether this is a better local minima, or a shortcut that is being learned by gradient descent. Thus, it's an open problem to study why gradient descent is learning some lower-dimensional representations in the first layer, if there's depth, instead of learning neural representations of consistently higher dimension that are Cayley graphs with both a reflection and a rotation generator being learned. Similarly, in models that learn this solution, the concentrated relation need not take exactly the $A_{\text{sum}}$ and $B_{\text{dif}}$ form described in the main analysis; depending on the architecture and layer, the corresponding signal can instead appear through $A_{\text{dif}}$ or $B_{\text{sum}}$. We therefore leave a systematic treatment of these phase-convention variants to future work.

## Impact Statement

This paper presents work whose goal is to advance interpretability of the field of deep learning. There are many potential societal consequences of our work, none of which we feel must be specifically highlighted here.

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

# Table of Contents for the Appendix

# A. Additional Background

## A.1. Dihedral group multiplication table

See Figure 8.

|  | Rotations | | | Reflections | | |
|---|---|---|---|---|---|---|
| $g \backslash h$ | $e$ | $r$ | $r^2$ | $s$ | $sr$ | $sr^2$ |
| $e$ | $e$ | $r$ | $r^2$ | $s$ | $sr$ | $sr^2$ |
| $r$ | $r$ | $r^2$ | $e$ | $sr^2$ | $s$ | $sr$ |
| $r^2$ | $r^2$ | $e$ | $r$ | $sr$ | $sr^2$ | $s$ |
| $s$ | $s$ | $sr$ | $sr^2$ | $e$ | $r$ | $r^2$ |
| $sr$ | $sr$ | $sr^2$ | $s$ | $r^2$ | $e$ | $r$ |
| $sr^2$ | $sr^2$ | $s$ | $sr$ | $r$ | $r^2$ | $e$ |

Entries are $g \cdot h$.
$r^i r^j = r^{(i+j) \bmod 3}$,
$r^i (sr^j) = sr^{(j-i) \bmod 3}$,
$(sr^i) r^j = sr^{(i+j) \bmod 3}$,
$(sr^i)(sr^j) = r^{(j-i) \bmod 3}$

*Figure 8.* Cayley table for $D_3$ ($= S_3$).

## A.2. Irreducible Representations and Group Fourier Transform

**Representation and irreducible representation.** Let $G$ be a group. A (finite-dimensional) representation of $G$ is a homomorphism
$$\rho : G \to \mathrm{GL}(V),$$
where $V$ is a complex vector space. We say that $\rho$ (or $V$) is *irreducible* if $V$ has no nontrivial $G$-invariant subspaces, i.e., the only subspaces $W \subseteq V$ with $\rho(g)W \subseteq W$ for all $g \in G$ are $W = \{0\}$ and $W = V$. A complete set of pairwise inequivalent irreducible representations is denoted by $\widehat{G}$.

**Group Fourier transform.** Let $G$ be a finite group and let $\widehat{G}$ denote a complete set of unitary irreducible representations (irreps) $\rho : G \to \mathrm{U}(d_\rho)$. For a function $f : G \to \mathbb{C}$, we define the (matrix-valued) *group Fourier transform* by

$$\widehat{f}(\rho) \;=\; \frac{1}{|G|} \sum_{g \in G} f(g)\, \rho(g)^\dagger \;\in\; \mathbb{C}^{d_\rho \times d_\rho}, \tag{2}$$

and the inverse transform by

$$f(g) \;=\; \sum_{\rho \in \widehat{G}} d_\rho \operatorname{Tr}\!\Big( \rho(g)\, \widehat{f}(\rho) \Big). \tag{3}$$

**Schur orthogonality (matrix elements).** For irreducible representations $\rho^\lambda$ and $\rho^\mu$ of dimensions $d_\lambda$ and $d_\mu$,

$$\frac{1}{|G|} \sum_{g \in G} \rho^\lambda_{ij}(g)\, \overline{\rho^\mu_{i'j'}(g)} \;=\; \frac{\delta_{\lambda\mu}\, \delta_{ii'}\, \delta_{jj'}}{d_\lambda}. \tag{4}$$

With the normalization in (2), the corresponding Parseval/Plancherel identity is

$$\frac{1}{|G|} \sum_{g \in G} |f(g)|^2 \;=\; \sum_{\rho \in \widehat{G}} d_\rho \, \|\widehat{f}(\rho)\|_F^2. \tag{5}$$

**Product groups.** For $F : G \times G \to \mathbb{C}$ we use the tensor-product of irreducible representations $\rho \otimes \sigma$ of dimension $d_\rho d_\sigma$. The two-variable Fourier transform and its inverse are

$$\widehat{F}(\rho, \sigma) \;=\; \frac{1}{|G|^2} \sum_{(g_1, g_2) \in G \times G} F(g_1, g_2) \left( \rho(g_1)^\dagger \otimes \sigma(g_2)^\dagger \right), \tag{6}$$

$$F(g_1, g_2) \;=\; \sum_{\rho, \sigma} d_\rho d_\sigma \operatorname{Tr}\!\Big( \big( \rho(g_1) \otimes \sigma(g_2) \big) \widehat{F}(\rho, \sigma) \Big). \tag{7}$$

For this convention, the Parseval identity takes the form

$$\frac{1}{|G|^2} \sum_{g_1,g_2 \in G} |F(g_1,g_2)|^2 \;=\; \sum_{\rho,\sigma} d_\rho d_\sigma \, \|\widehat{F}(\rho,\sigma)\|_F^2. \tag{8}$$

### A.3. Irreducible Representations of the Dihedral Group $D_n$

We use the presentation

$$D_n \;=\; \langle r,s \mid r^n = e,\; s^2 = e,\; srs = r^{-1}\rangle.$$

**One-dimensional irreducible representations.** For all $n$, there are the trivial and sign characters:

$$\chi_{\mathrm{triv}}(r^m) = 1, \quad \chi_{\mathrm{triv}}(sr^m) = 1; \qquad \chi_{\mathrm{sign}}(r^m) = 1, \quad \chi_{\mathrm{sign}}(sr^m) = -1.$$

If $n$ is *even*, there are two additional one-dimensional irreps obtained by sending $r \mapsto -1$ and $s \mapsto \pm 1$. Explicitly, for $m \in \{0,\dots,n-1\}$,

$$\chi_{rp}(r^m) = (-1)^m, \quad \chi_{rp}(sr^m) = (-1)^m,$$
$$\chi_{srp}(r^m) = (-1)^m, \quad \chi_{srp}(sr^m) = -(-1)^m.$$

**Two-dimensional irreducible representations** ($2D_k$)**.** For $k = 1,\dots,\lfloor (n-1)/2 \rfloor$, define

$$R_k(r^m) = \begin{pmatrix} \cos\frac{2\pi km}{n} & -\sin\frac{2\pi km}{n} \\ \sin\frac{2\pi km}{n} & \cos\frac{2\pi km}{n} \end{pmatrix}, \qquad R_k(sr^m) = \begin{pmatrix} 1 & 0 \\ 0 & -1 \end{pmatrix} R_k(r^m). \tag{9}$$

These are real orthogonal (hence unitary) representations and satisfy the defining relation $srs = r^{-1}$. Substituting these matrices into the general formulas (2)–(7) yields the transforms used in our implementation.

### A.4. Approximate cosets

Approximate cosets are intuitively, the generalization of cosets to "almost a coset". They arise when neurons in a network learn to divide a group using a structure that doesn't actually divide the group. For example, $D_{18}$ has 36 elements: 18 rotations and 18 reflections. Frequency 6 induces a nontrivial collapse of the rotation indices modulo 3: the rotations form three classes, each containing 6 rotations, and the reflections form three corresponding classes, each containing 6 reflections. Thus, the representation distinguishes elements by their rotation index modulo 3 and by whether they are rotations or reflections.

Suppose alternatively that we were trying to divide $D_{18}$ by 5. Since the $\gcd(5,18) = 1$, 5 doesn't factorize 18 into anything smaller. Thus, there are no non-trivial cosets; a neuron learning frequency 5 has learned the full group structure of $D_{18}$. Such a neuron has 5 peaks (maximum values), and if a peak is located at $a$, the next peaks are located $a \pm \frac{18}{5} = 3.6$. Naturally, because the problem is discrete, this results in every point having a different activation value and can be seen in Fig. 2 by contrasting (A) the coset case with (B) the approximate coset case. In (A), the neuron activates with 3 strengths depending on a and 3 strengths depending on $b$, whereas in (B) the neuron activates with 18 different strengths for $a$ and 18 different strengths for $b$.

We offer the reader the following intuition: approximate cosets are simply when a neuron has learned something that doesn't allow it to cleanly divide the Cayley graph into smaller pieces. A natural response is to think "perhaps neurons would prefer to learn things that cleanly divide the group?" and indeed, this is observed later in Fig. 4. For the mathematical definition, please see Section 3 and 4 in McCracken et al. (2025b).

### A.5. Concentration level metric

The concentration level $R \in [0,1]$ is the *mean resultant length* (MRL) of a set of angles $\{\theta_i\}_{i=1}^N$:

$$R \;:=\; \left|\langle e^{i\theta}\rangle\right| \;=\; \frac{\left\| \sum_{i=1}^N e^{i\theta_i} \right\|}{N}.$$

Intuitively, $R \approx 1$ means the angles concentrate tightly around a common value, whereas $R \approx 0$ means they are broadly dispersed.

In our setting, each $\theta_i$ is one observed value of a half-phase combination (computed at the dominant frequency $f_{\text{dom}}$), namely

$$A_{\text{sum}} = \left(\phi^{(a)}_{-,f_{\text{dom}}} + \phi^{(a)}_{+,f_{\text{dom}}}\right) \bmod 2\pi, \qquad A_{\text{dif}} = \left(\phi^{(a)}_{-,f_{\text{dom}}} - \phi^{(a)}_{+,f_{\text{dom}}}\right) \bmod 2\pi,$$

and analogously $B_{\text{sum}}$ and $B_{\text{dif}}$.

### A.6. Spearman rank correlation

Spearman's rank correlation coefficient $\rho$ is a nonparametric measure of monotone association between two sequences, defined as the Pearson correlation of their rank-transformed values. In our setting, we compute $|\rho|$ between the step index along a rotation trace and the cumulative polar angle of the embedded points (after projecting to a best-fit 2D subspace). Values near 1 indicate a nearly monotone angular progression consistent with an ordered cycle, while values near 0 indicate a scrambled or non-monotone ordering.

## B. Experimental settings

### B.1. Training setup

All experiments were trained for 5000 epochs using Adam. All experiments were done on CPUs except for the scaling dihedral group size $n$ plot, which required RTX8000 GPUs to make the experiment faster.

If specific hyperparameters aren't listed, MLPs use $\text{lr} = 0.001$ and $\text{wd} = 0.0001$, and transformers use learning rate $= 0.0005$ and L2 weight decay $= 10^{-7}$.

### B.2. Figure-specific conventions

B.2.1. FIGURE 3: COLOR AND LINE CONVENTIONS; $D_{18}$

**Preactivation PCA and Color Coding by $a$ or $b$.**    In the preactivation PCA plots of Figure 3 (A), we color each point according to the coset of $a$, $b$, or $C$. For all cases we use a continuous colormap (viridis) split into two halves:

- the lower half of the colormap is used for *rotation* cosets (sign $+1$);

- the upper half is used for *reflection* cosets (sign $-1$).

This makes the rotation vs. reflection structure visible as a change in hue even when plotting a single scalar residue.

**Lines When Coloring by $a$.**    When we color by $a$, we additionally draw line segments to reveal how residues in the same coset are arranged:

- We fix a value of $b$ and order the corresponding points by stepping $a \mapsto a + d \pmod{g}$, where $g = n/\gcd(n, f)$ and $d$ is defined in 5.1, starting from the smallest residue present in that stripe.

- For a given fixed $b$, we obtain up to four "stripes" depending on whether $a$ and $b$ lie in the rotation block or the reflection block.

- **Blue vs. red** encodes whether $b$ is in the rotation block (blue) or the reflection block (red).

- **Solid vs. dashed** encodes whether $a$ is in the rotation block (solid) or the reflection block (dashed).

- Within each stripe, points are connected in the order given by repeated addition of the step size $d$ modulo $g$, so each polyline traces out a full cycle of residues inside that coset.

**Lines When Coloring by $b$.**    The construction for "color by $b$" is symmetric:

- We fix $a$ and order points by stepping $b \mapsto b + d \pmod{g}$.

- As before, we obtain up to four stripes depending on whether $a$ and $b$ are in rotation block or reflection block, and we connect points in each stripe according to the modular step order.

These color and line conventions are purely for visualization: they do not affect the PCA itself, but make the underlying $D_3$ coset structure and the separation between rotation and reflection components visually apparent.

### B.2.2. FIGURE 3: CLUSTER CONTRIBUTIONS TO THE LOGITS

For each frequency cluster, we define its *cluster contribution to the logits* by isolating the post-activations of the neurons in that cluster at the penultimate layer and propagating only those activations to the logits.

Concretely, let $H_{\text{cluster}} \in \mathbb{R}^{(2n)^2 \times m}$ collect the post-activations of all neurons in the cluster across the $(2n)^2$ input points (rows index inputs, columns index neurons in the cluster), and let $W_{\text{cluster}} \in \mathbb{R}^{m \times (2n)}$ be the slice of the final weight matrix restricted to those neurons. The logit contribution matrix for this cluster is

$$L_{\text{cluster}} = H_{\text{cluster}} W_{\text{cluster}} \in \mathbb{R}^{(2n)^2 \times (2n)}, \tag{10}$$

which has the same shape as the full logit matrix.

All PCA plots of the "cluster's contributions to the logits" in Figure 3 are obtained by running PCA on matrix $L_{\text{cluster}}$.

### B.3. Additional experiments and parameters

### B.3.1. FIGURES 4 AND 5: SPLIT AND DATASETS

90%/10% train/test split, $D_{18}$, $D_{19}$.

### B.3.2. FIGURE 6: SCALING SWEEP HYPERPARAMETERS

90%/10% train/test split $D_{2^k}$, with $k \in [1, 9]$.

*Table 2.* Transformer scaling experiments: learning rate (LR) and weight decay (WD) values per $n$.

| $n$ | LR | WD |
|---|---|---|
| 2 | 1e-3 | 1e-5 |
| 4 | 5e-4 | 1e-4 |
| 8 | 5e-4 | 1e-4 |
| 16 | 2e-3 | 1e-4 |
| 32 | 2e-4 | 1e-6 |
| 64 | 2e-4 | 1e-7 |
| 128 | 2e-5 | 1e-7 |
| 256 | 2e-5 | 1e-7 |
| 512 | 2e-5 | 1e-7 |

*Table 3.* MLP scaling experiments: learning rate (LR) and weight decay (WD) selections per $n$.

| $n$ | LR | WD |
|---|---|---|
| 2 | 5e-5 | 1e-4 |
| 4 | 5e-5 | 1e-6 |
| 8 | 1e-3 | 1e-4 |
| 16 | 2e-3 | 1e-4 |
| 32 | 2e-3 | 1e-4 |
| 64 | 5e-4 | 1e-4 |
| 128 | 2e-3 | 1e-5 |
| 256 | 1e-4 | 1e-6 |
| 512 | 1e-5 | 1e-7 |

## B.3.3. RNN

*Table 4.* Results for RNNs using the same settings as in the main paper: $R^2$, coset clustering score ($s_{cluster}$), and coset disjointness, averaged across 300 seeds. Note that replacing each neuron's preactivation (layerwise) by its fitted sinusoid(s) yields 100% test accuracy in all runs, and rotation traces form coherent loops with near-monotone angular progression ($|\rho| = 0.9968 \pm 0.0189$).

| $s_{cluster}$ | coset disjointness | $R^2$ |
|---|---|---|
| $0.9968 \pm 0.0032$ | $0.9999 \pm 0.0009$ | $0.9934 \pm 0.0078$ |

### B.3.4. HYPERPARAMETER TUNING HEATMAPS FOR DIFFERENT ARCHITECTURES, DEPTHS, WIDTHS, AND ACTIVATION FUNCTIONS

See Fig. 9, Fig. 10, Fig. 11, Fig. 12, Fig. 13 and Fig. 14. The networks achieve high scores on all of the metrics, except at the edge of the hyperparameter region, which corresponds to where networks struggle to learn to generalize to the test set. This makes sense because the solutions at the edge correspond to bad hyperparameters, and it's well known that hyperparameter quality influences the quality of the learned local minima and ability for trained networks to generalize to the test set. Furthermore, these plots show a green check mark on the hyperparameter combinations where deleting every neuron and replacing it by its best fit resulted in the network maintaining 100% full dataset (train + test) accuracy. We mark a cell with a red X if the network fails to learn, i.e. reach 100% full dataset accuracy. If it does generalize, but replacing all neurons with their best fitted sinusoid functions does not maintain 100% accuracy on the full dataset, we mark the cell with a purple dot. Note, purple dots are our way of showing how robust our interpretation is, because if a single datum becomes misclassified (MLP) or mispredicted (transformer) a purple dot will appear.

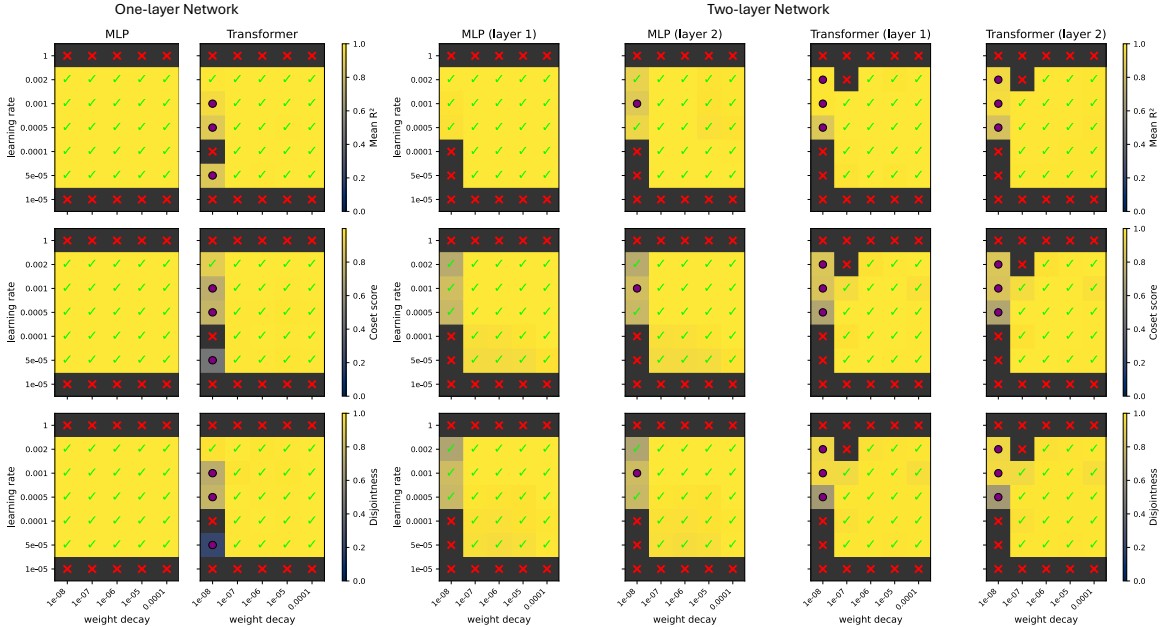

*Figure 9.* Hyperparameter tuning results for one- and two-layer MLPs and transformers with ReLU as activation function. Each heatmap shows a sweep over learning rate (y-axis) and weight decay (x-axis) for MLPs and transformers. Rows correspond to different metrics: the top row reports the mean coefficient of determination $R^2$ between network preactivations and the sinusoidal fits, the middle row reports the coset clustering score, and the bottom row reports the disjointness metric. Colors indicate the metric value averaged over random seeds; we only fill a cell when all seeds reach 100% train and test accuracy during training; runs where any seed fails to generalize to 100% accuracy are marked with a red "×". Among successful runs, a purple "○" denotes that replacing the neuron preactivations by the sinusoidal fits results in full dataset accuracy < 99.9999%, while a green "✓" denotes replacing neuron's preactivation by the sinusoidal fit results in full dataset accuracy > 99.9999%.

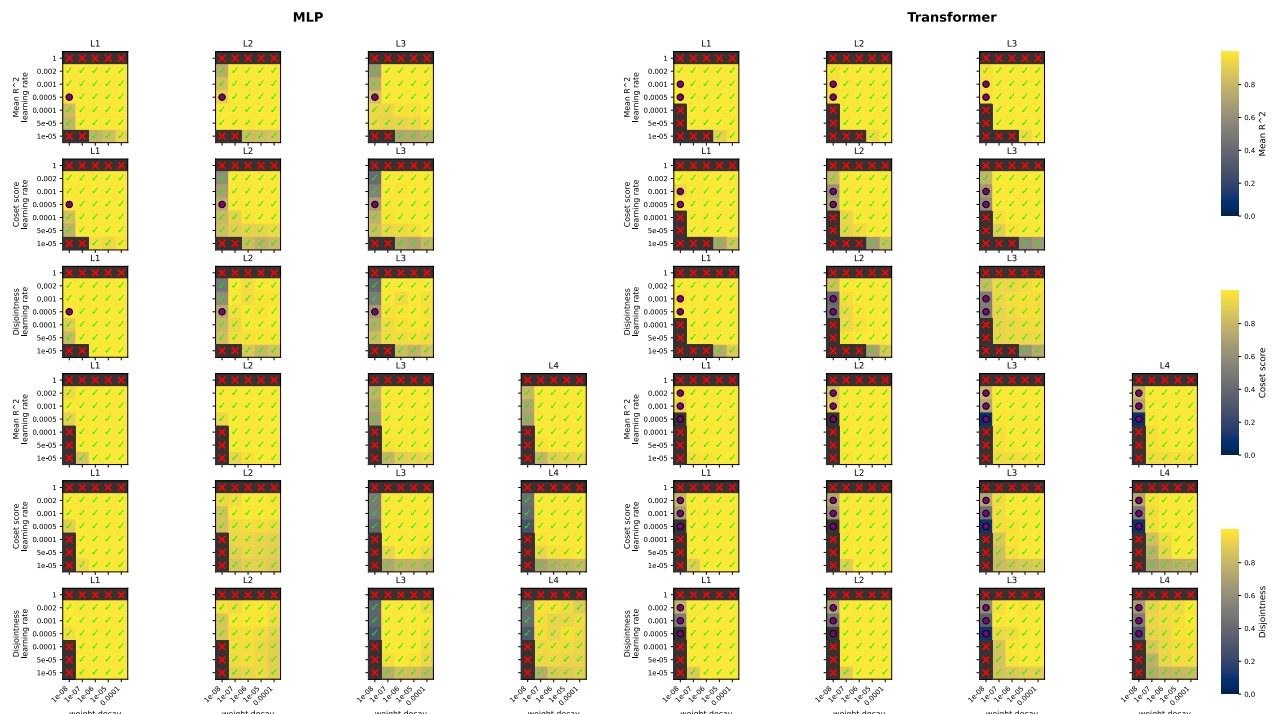

*Figure 10.* Hyperparameter tuning results for three- and four-layer MLPs and transformers with ReLU as activation function. As in Figure 9, each heatmap shows a sweep over learning rate (y-axis) and weight decay (x-axis), and the color scale and markers (red "×", purple "○", and green "✓") follow the same conventions.

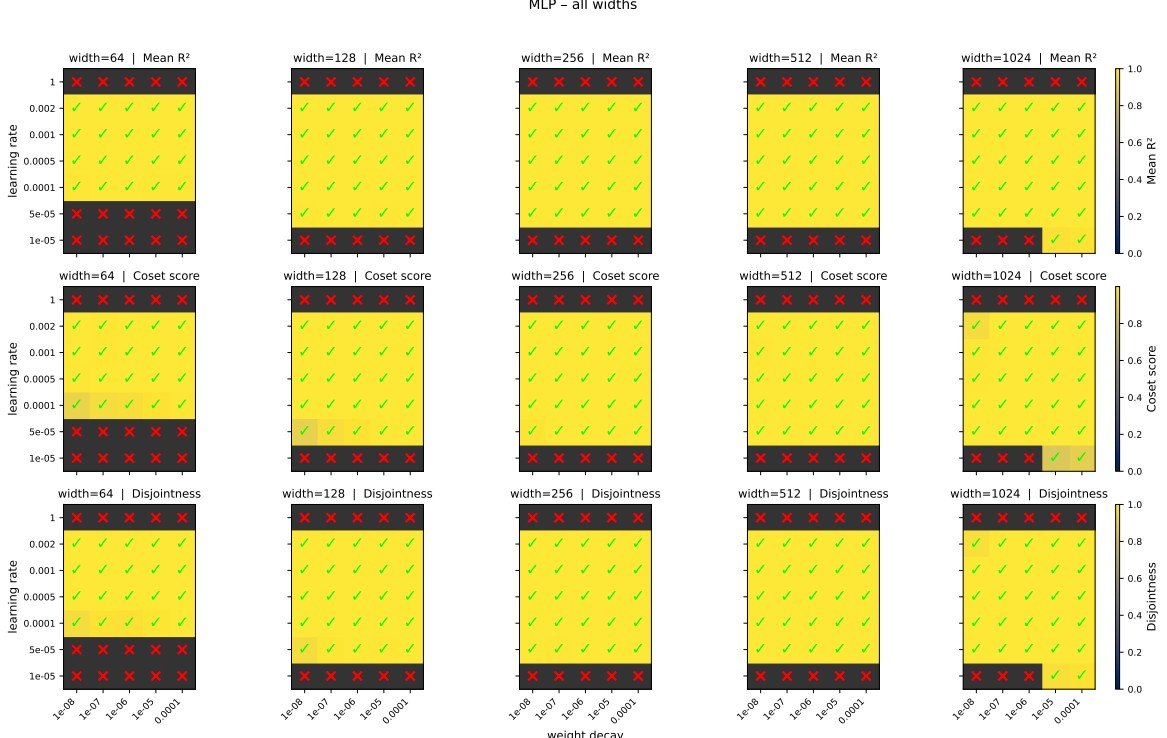

*Figure 11.* Hyperparameter tuning results over different widths of one-layer MLPs. As in Figure 9, each heatmap shows a sweep over learning rate (y-axis) and weight decay (x-axis), and the color scale and markers (red "×", purple "○", and green "✓") follow the same conventions.

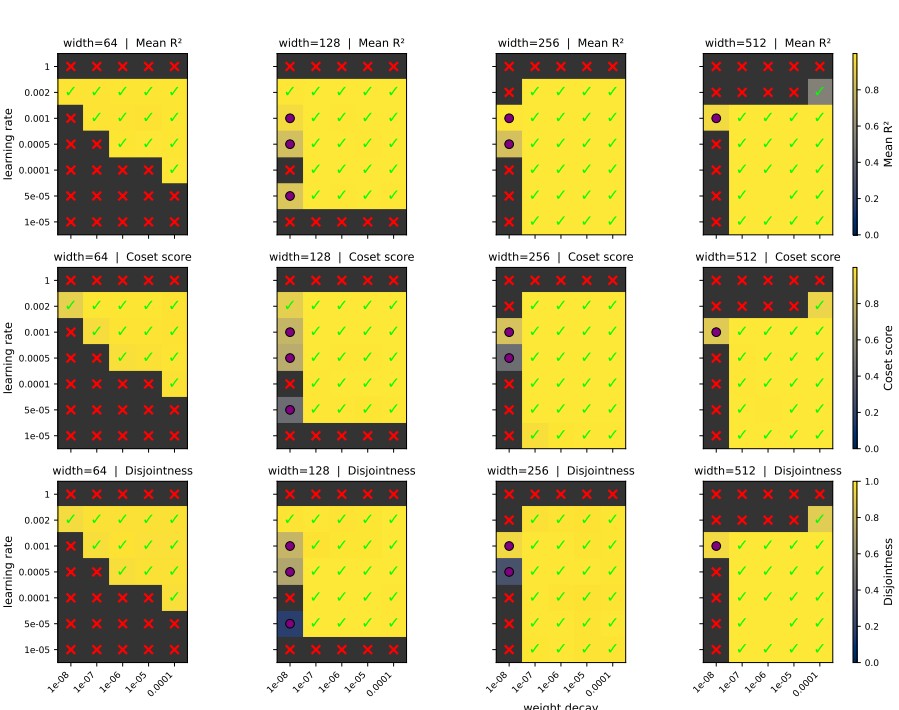

*Figure 12.* Hyperparameter tuning results over different widths (dmodel) of one-layer transformers. As in Figure 9, each heatmap shows a sweep over learning rate (y-axis) and weight decay (x-axis), and the color scale and markers (red "×", purple "∘", and green "✓") follow the same conventions.

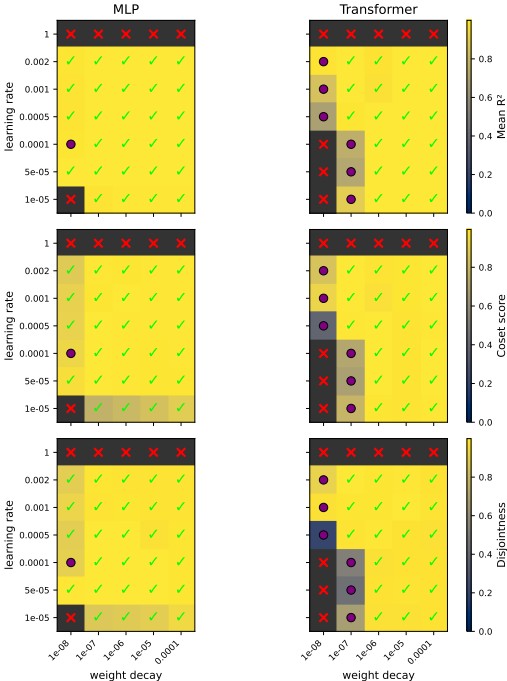

*Figure 13.* Hyperparameter tuning results for one-layer MLPs and transformers with pure quadratic activation function. As in Figure 9, each heatmap shows a sweep over learning rate (y-axis) and weight decay (x-axis), and the color scale and markers (red "×", purple "∘", and green "✓") follow the same conventions.

RNN

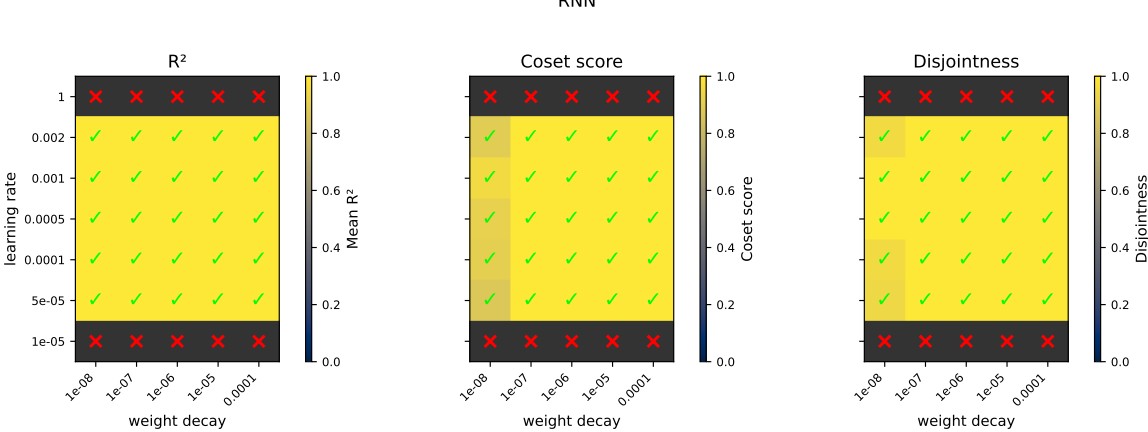

*Figure 14.* Hyperparameter tuning results for RNN. Each heatmap shows a sweep over learning rate (y-axis) and weight decay (x-axis). Columns correspond to different metrics: the first column reports the mean coefficient of determination $R^2$ between network preactivations and the sinusoidal fits, the second reports the coset clustering score, and the third reports the disjointness metric. Colors and markers (red "×", purple "∘", and green "✓") follow the same conventions.

### B.3.5. RANDOM MULTIPLICATION VS. DIHEDRAL GROUP MULTIPLICATION

In this section we replace the true dihedral group multiplication by a random multiplication table where each entry is sampled independently at random over the same output space. We keep the same one-layer architecture as in the main text and train the network either on the true dihedral multiplication or on the random multiplication, using the same training hyperparameters and number of seeds. This provides a negative-control setting in which no coherent group structure is expected.

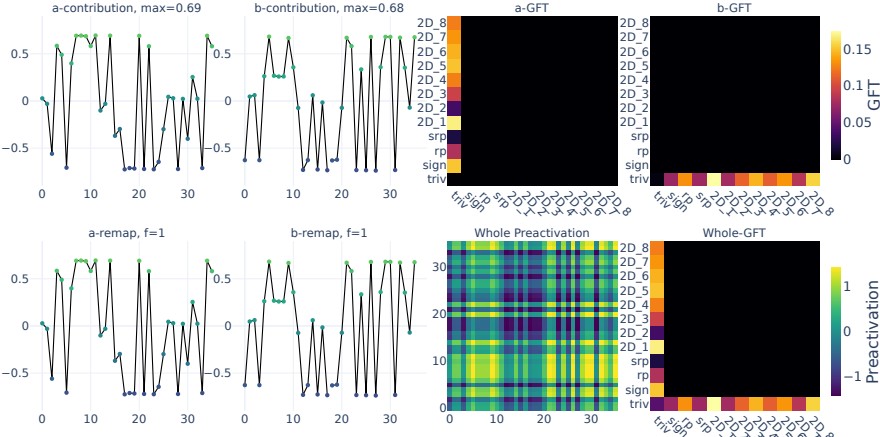

*Figure 15.* Preactivations of a single neuron under the random multiplication task. Unlike in the dihedral setting, the preactivation does not exhibit structured sinusoidal behavior or concentration on a small number of Fourier modes.

For each neuron, we fit first-order sinusoids to its preactivations (as a function of the group element) and report the resulting average coefficient of determination $R^2$. Higher values indicate that the neuron is well approximated by a small number of Fourier modes.

For the coset-level geometry, we use the same low-dimensional representation space as in the main paper. In addition to the coset clustering score $s_{\text{cluster}}$ and the coset disjointness score in the main paper, we also compute a coset *radius* statistic:

- **Coset radius:** for each coset, we compute the radius of its minimal enclosing ball in PCA space and report the mean radius across cosets.

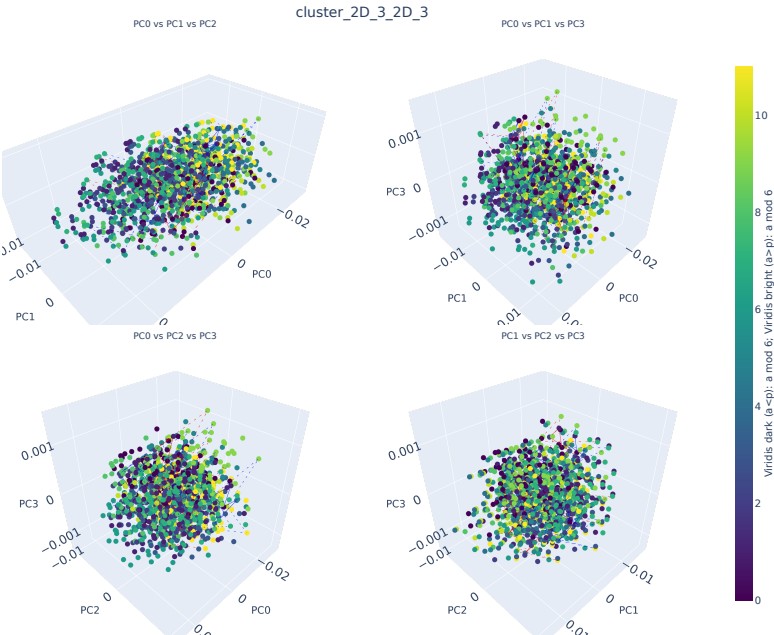

*Figure 16.* PCA visualization of a representative cluster in the random multiplication task, using the same pipeline as in the main paper. Unlike in the dihedral setting, the points do not form geometrically structured clusters in PCA space, indicating the absence of coherent group-induced feature organization.

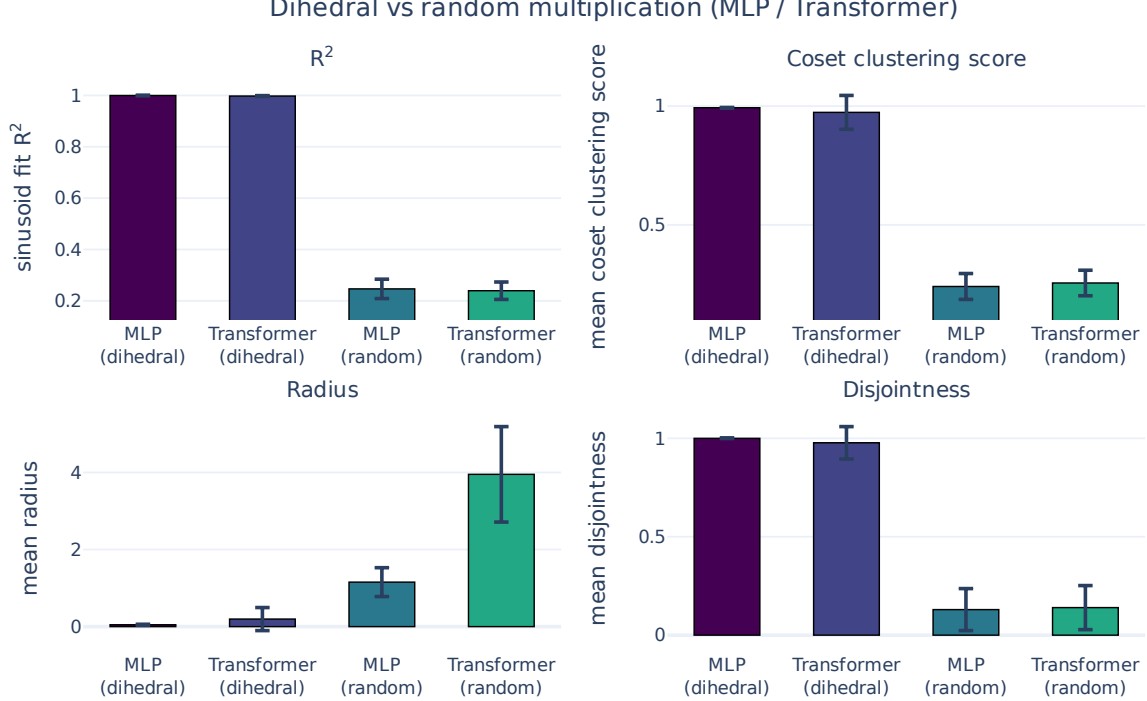

*Figure 17.* We train one-layer MLPs and one-layer transformers either on the true dihedral multiplication law or on a random multiplication table, using the same training protocol and number of random seeds for each setting. We aggregate the sinusoidal-fit $R^2$ and coset-level geometric statistics over 1000 seeds.

As shown in Figure 17, for both architectures the true dihedral law yields substantially higher sinusoidal-fit $R^2$ and coset clustering scores, as well as smaller coset radii and larger coset disjointness, than the random-multiplication baseline. In

other words, when we destroy the group structure while keeping the architecture, training protocol, and analysis pipeline fixed, both the simple sinusoidal behavior at the neuron preactivation level and the coset-level geometry largely disappear. This negative-control experiment therefore supports the interpretation that the structures we observe in the main paper reflect the learned dihedral multiplication law itself, rather than generic artifacts of the PCA or group Fourier analysis.

## C. Left/Right Multiplication and Half-Phase Relations

In the $sr^k$ convention, the reflection element $sr^k$ acts differently depending on which side it appears in the product, and this is the concrete source of the "flip" in the $a \cdot b$ table (Eq. (1)). Viewing the input (a, b) as left vs. right actions, fixing a reflection on the *right* and varying the *left* rotation gives

$$r^a \cdot sr^k \;=\; sr^{(k-a) \bmod n},$$

so if the resulting reflection half is indexed by $i = (k - a) \bmod n$, then $k \equiv i + a \pmod n$ is constant along that input variable. Conversely, fixing a reflection on the *left* and varying the *right* rotation gives

$$sr^k \cdot r^b \;=\; sr^{(k+b) \bmod n},$$

so if the resulting reflection half is indexed by $j = (k + b) \bmod n$, then $k \equiv j - b \pmod n$ is constant. These two index relations ($k = i + a$ versus $k = j - b$) are inequivalent precisely because $sr^k$ does not commute with rotations, and they identify which side the reflection generator effectively acts on when one moves along $a$ versus $b$.

In the first-layer coupled sinusoid fits, each variable has two effective half-phases at the dominant frequency $f_0$, one from the rotation half and one from the reflection half. Writing these as $\phi^{(a)}_{\pm,f_0}$ and $\phi^{(b)}_{\pm,f_0}$, we summarize the observed within-cluster constraints by the half-phase combinations

$$A_{\mathrm{sum}} = \big(\phi^{(a)}_{-,f_0} + \phi^{(a)}_{+,f_0}\big) \bmod 2\pi, \qquad B_{\mathrm{dif}} = \big(\phi^{(b)}_{-,f_0} - \phi^{(b)}_{+,f_0}\big) \bmod 2\pi,$$

(with the complementary $A_{\mathrm{dif}}$ and $B_{\mathrm{sum}}$ defined analogously). Empirically, across most clusters $A_{\mathrm{sum}}$ and $B_{\mathrm{dif}}$ are highly concentrated to cluster-specific constants, consistent with the side-dependent relations above: along $a$, the reflection generator appears on the right ($k \equiv i + a$), yielding an additive constraint between rotation and reflection halves, whereas along $b$ it appears on the left ($k \equiv j - b$), yielding a difference-type constraint.

## D. More Figures

### D.1. Additional results over 1000 seeds

*Table 5.* Mean of cluster-average $R^2$ per Fourier basis for 1-layer MLP and transformer preactivations. We only fit the coupled first-order sinusoids. For both architectures, replacing each neuron's preactivation (layerwise) by its fitted sinusoid(s) leaves test accuracy at 100% in 100% of runs.

| Fourier basis | MLPs | Transformers |
|---|---|---|
| rp | 0.9999 | 0.9998 |
| srp | 0.9999 | 0.9998 |
| 2D_1 | 0.9995 | 0.9964 |
| 2D_2 | 0.9999 | 0.9962 |
| 2D_3 | 0.9999 | 0.9993 |
| 2D_4 | 0.9999 | 0.9967 |
| 2D_5 | 0.9996 | 0.9966 |
| 2D_6 | 0.9999 | 0.9994 |
| 2D_7 | 0.9999 | 0.9969 |
| 2D_8 | 0.9999 | 0.9962 |

Here, we show the Cayley graph distribution on $D_{19}$ (Fig. 18) and that 2-layer networks also learn Cayley graphs (Table 6 and Fig. 19).

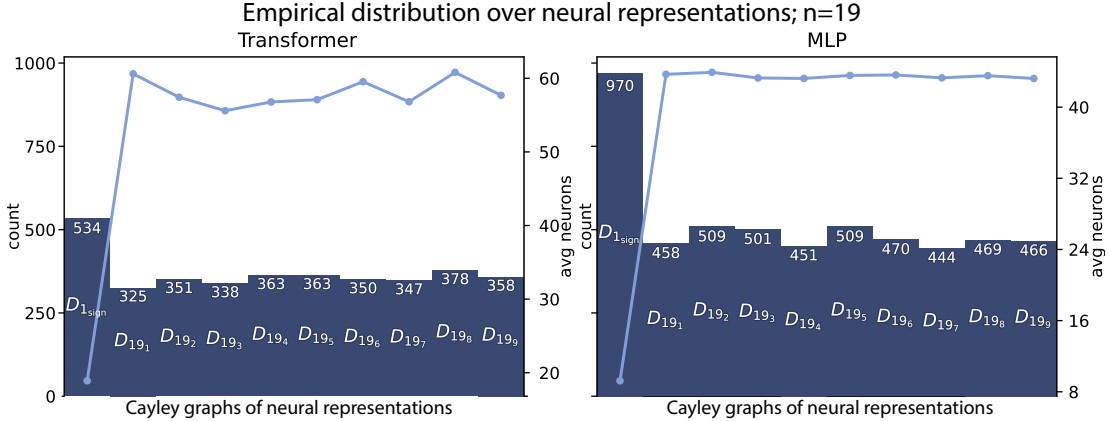

*Figure 18.* We study transformers and MLPs over 1000 random seeds on $D_{19}$ (no non-trivial coset structure). **Barplot.** Number of times each Cayley graph was learned. **Line plot.** Average number neurons used to learn each Cayley graph representation.

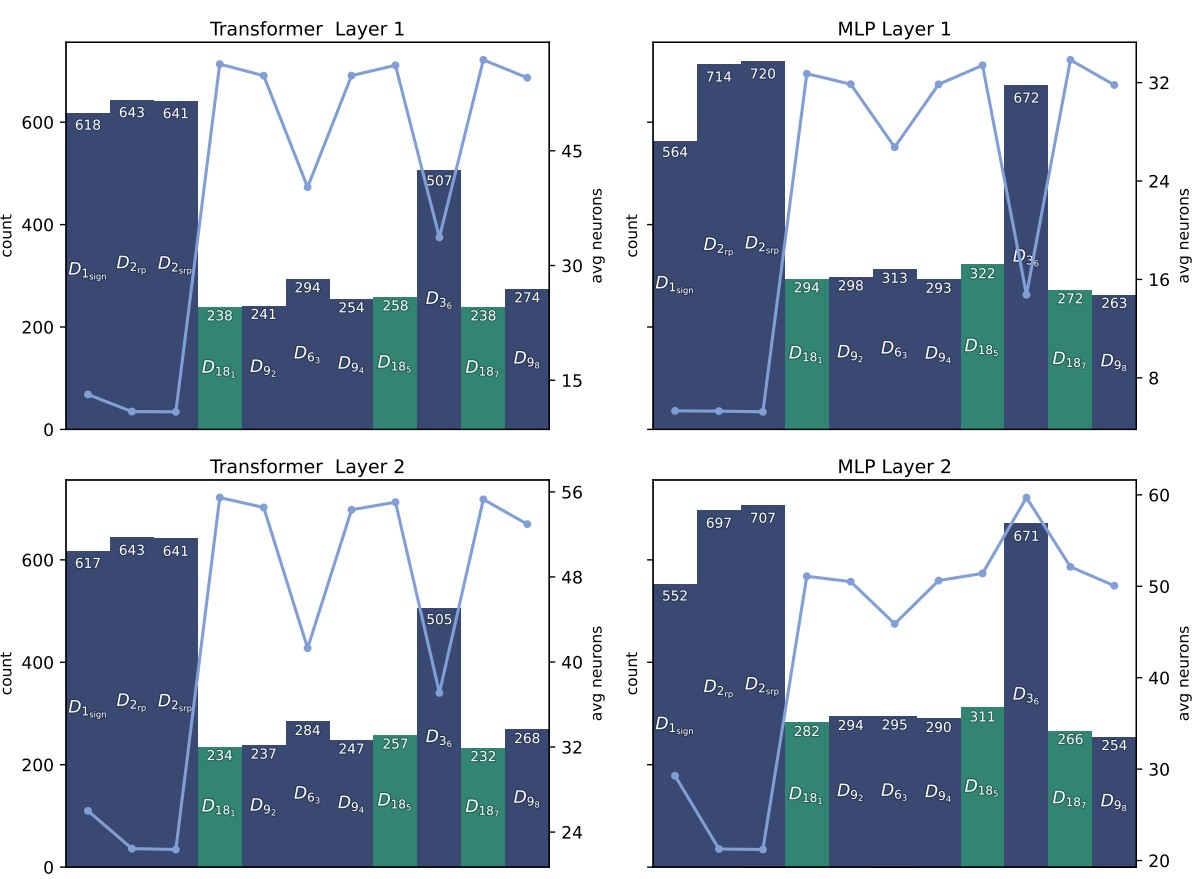

*Figure 19.* Across-seed average Cayley graph-frequency spectra learned by two-layer networks on $D_{18}$ and the average number of neurons per cluster (1000 random seeds). The two panels overlay results from transformers with a learning rate schedule and MLPs trained under the same protocol. Top: layer 1; bottom: layer 2.

*Table 6.* Mean of cluster-average $R^2$ per Fourier basis for 2-layer MLP and transformer preactivations. In Layer 1 we fit only first-order sinusoids, whereas in Layer 2 we fit both first- and second-order sinusoids. For the MLP and for the transformer with a learning rate schedule, replacing each neuron's preactivation (layerwise) by its fitted sinusoid(s) leaves test accuracy at $100\%$ in all runs.

| Fourier basis | MLPs | | Transformers with lr schedule | | Transformers without lr schedule | |
| --- | --- | --- | --- | --- | --- | --- |
| | Layer 1 | Layer 2 | Layer 1 | Layer 2 | Layer 1 | Layer 2 |
| rp | 0.9999 | 0.9988 | 0.9981 | 0.9994 | 0.9849 | 0.9096 |
| srp | 0.9999 | 0.9988 | 0.9986 | 0.9993 | 0.9812 | 0.9084 |
| 2D_1 | 0.9968 | 0.9879 | 0.9971 | 0.9735 | 0.9719 | 0.8642 |
| 2D_2 | 0.9990 | 0.9902 | 0.9970 | 0.9727 | 0.9673 | 0.8611 |
| 2D_3 | 0.9831 | 0.9769 | 0.9999 | 0.9906 | 0.9662 | 0.8638 |
| 2D_4 | 0.9987 | 0.9892 | 0.9952 | 0.9745 | 0.9673 | 0.8696 |
| 2D_5 | 0.9958 | 0.9863 | 0.9972 | 0.9740 | 0.9736 | 0.8707 |
| 2D_6 | 0.9997 | 0.9995 | 0.9946 | 0.9998 | 0.9814 | 0.9203 |
| 2D_7 | 0.9967 | 0.9881 | 0.9975 | 0.9745 | 0.9735 | 0.8677 |
| 2D_8 | 0.9989 | 0.9900 | 0.9941 | 0.9722 | 0.9705 | 0.8699 |

## D.2. Library of neuron classes that occur in $D_{18}$

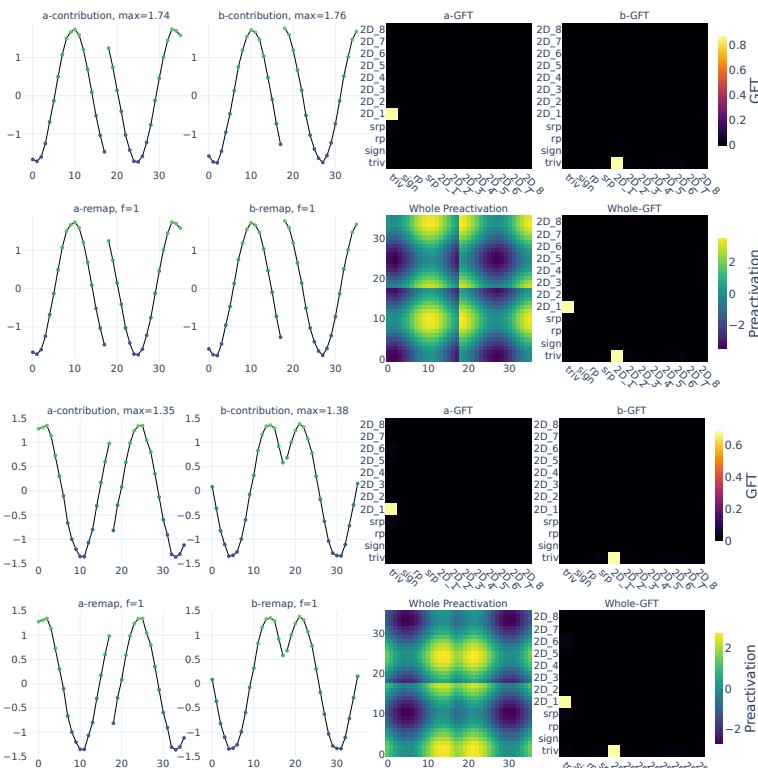

*Figure 20.* Visualization for $n = 18$, Fourier basis $2D_1$. Two neurons with highest preactivation in $2D_1$.

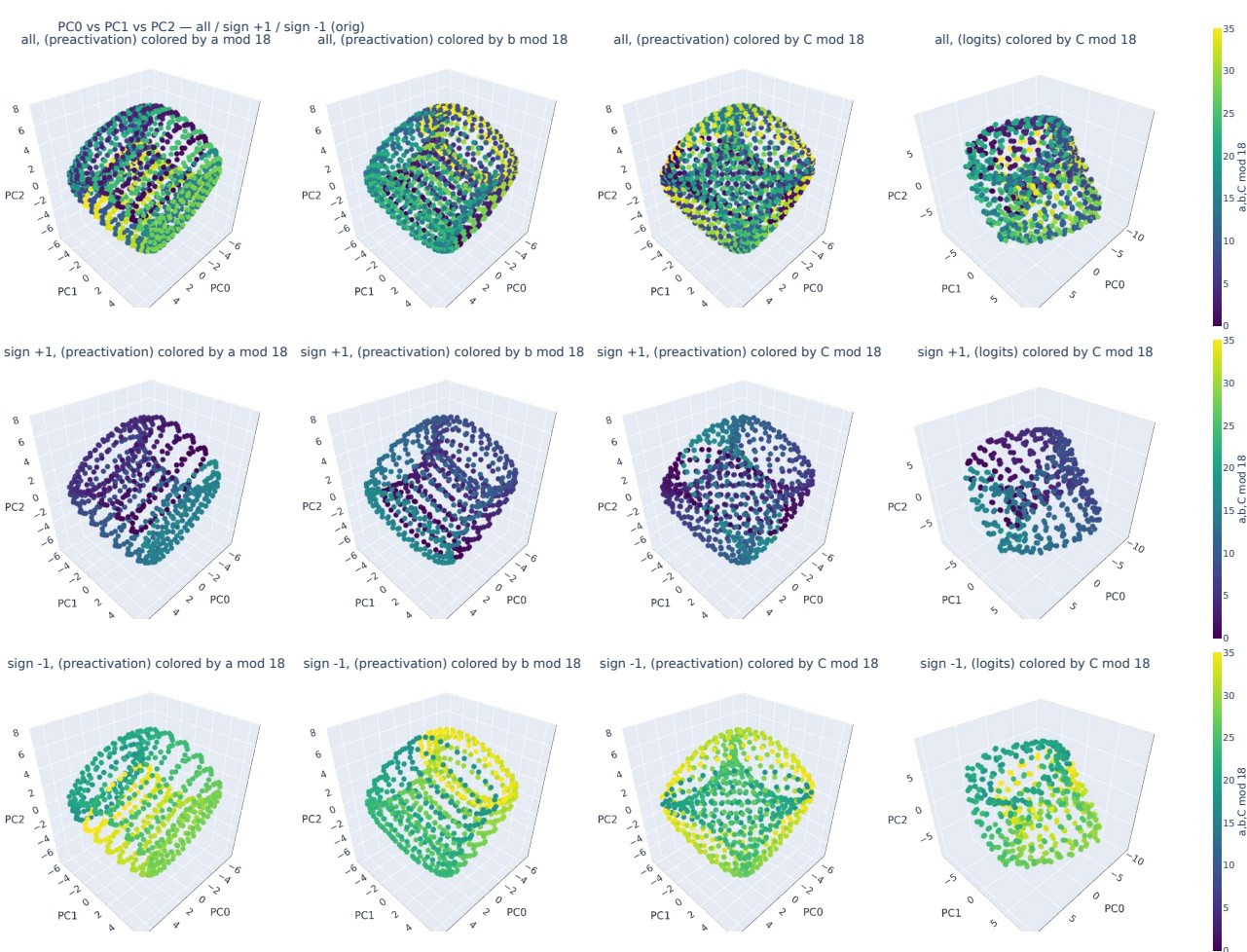

*Figure 21.* Visualization for $n = 18$, Fourier basis $2D_1$. PCA of preactivations and cluster contributions to logits, colored by $a$, $b$, and $C$ modulo $g = n/\gcd(n, f)$, with rotation classes $0, \ldots, g - 1$ and reflection classes $g, \ldots, 2g - 1$.

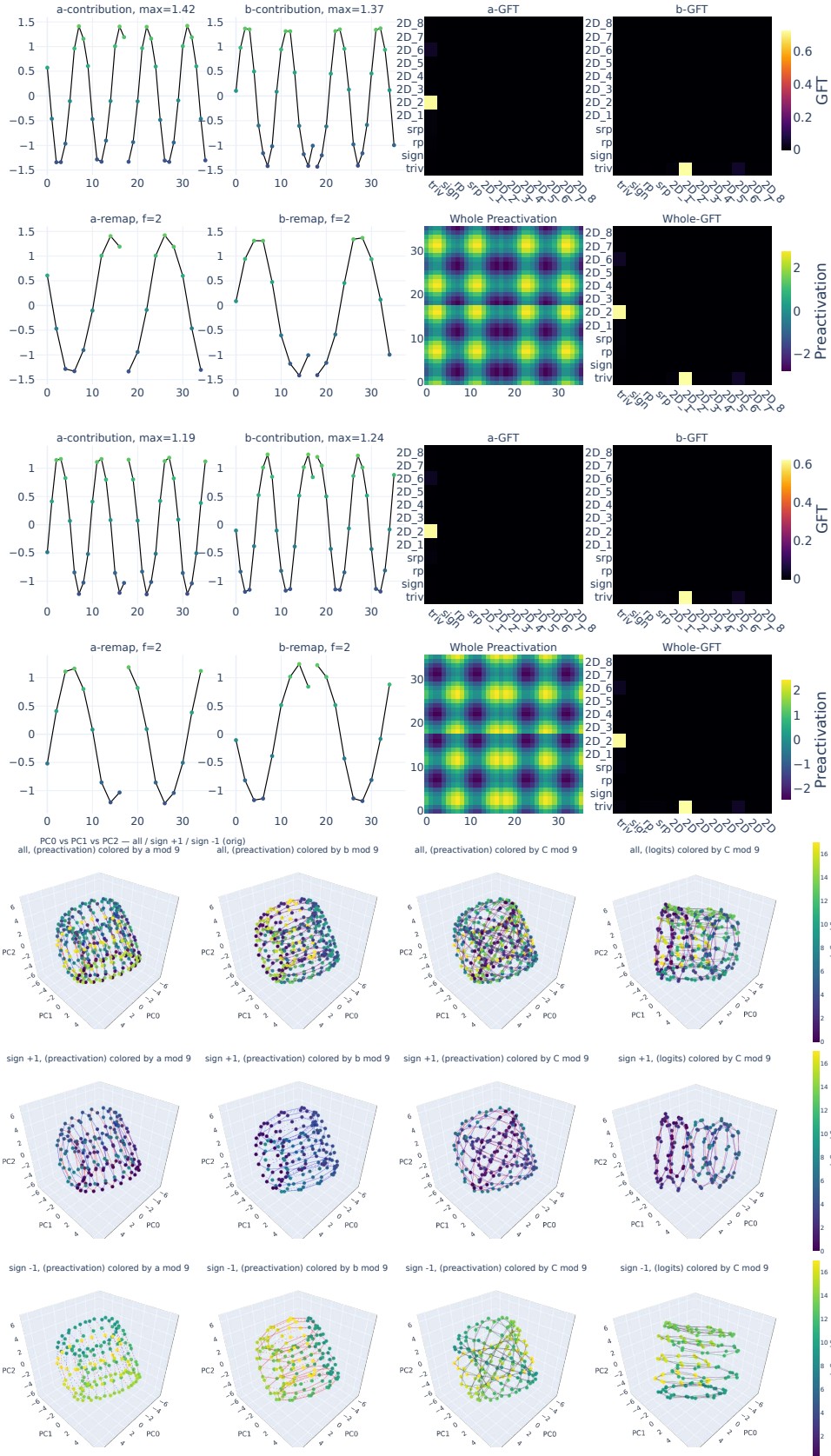

*Figure 22.* Visualization for $n = 18$, Fourier basis $2D_2$. Top: two neurons with highest preactivation in $2D_2$. Bottom: PCA of preactivations and cluster contributions to logits, colored by $a$, $b$, and $C$ modulo $g = n/\gcd(n, f)$, with rotation classes $0, \ldots, g - 1$ and reflection classes $g, \ldots, 2g - 1$.

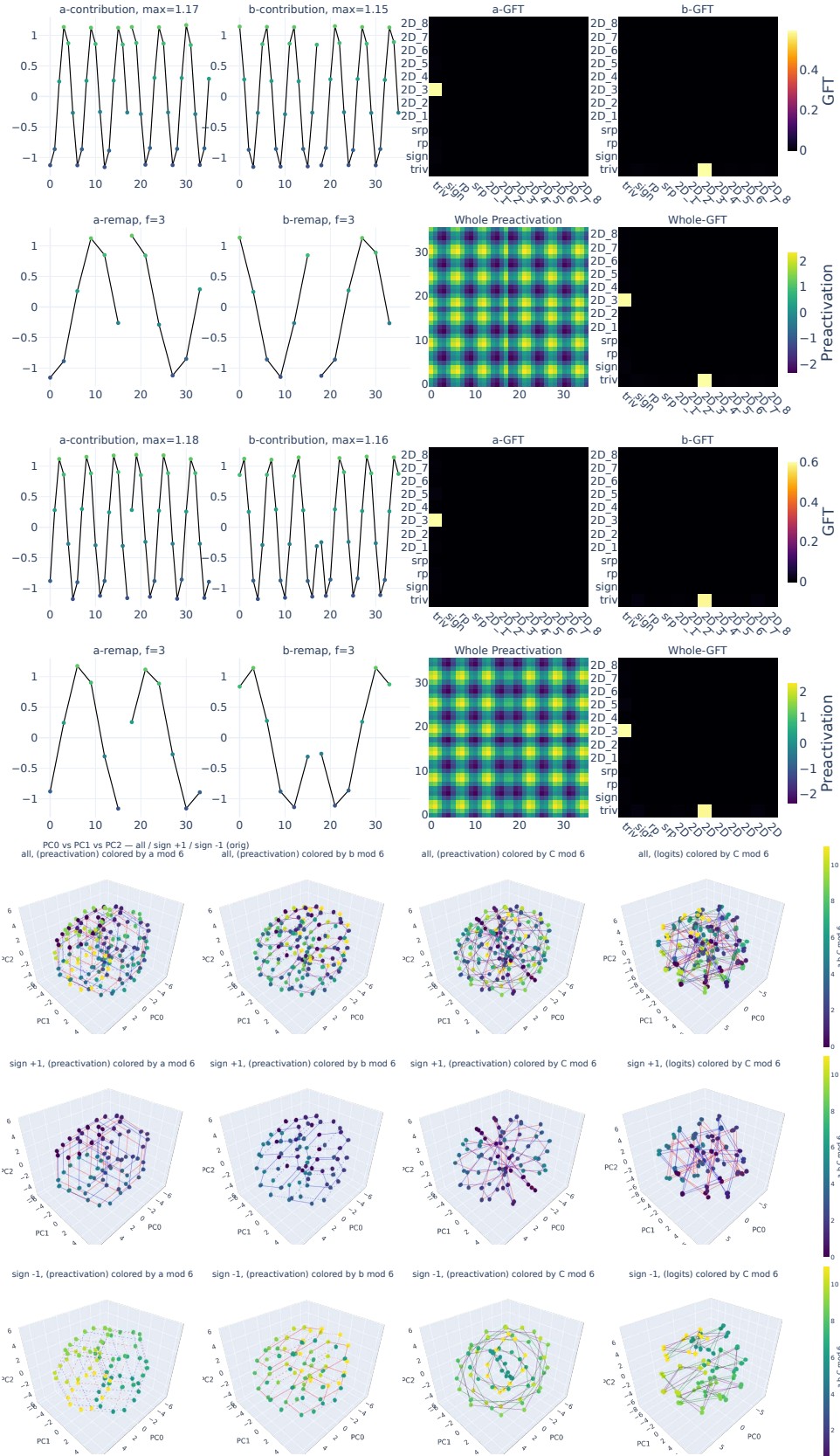

*Figure 23.* Visualization for $n = 18$, Fourier basis $2D_3$. Top: two neurons with highest preactivation in $2D_3$. Bottom: PCA of preactivations and cluster contributions to logits, colored by $a$, $b$, and $C$ modulo $g = n/\gcd(n, f)$, with rotation classes $0, \ldots, g-1$ and reflection classes $g, \ldots, 2g - 1$.

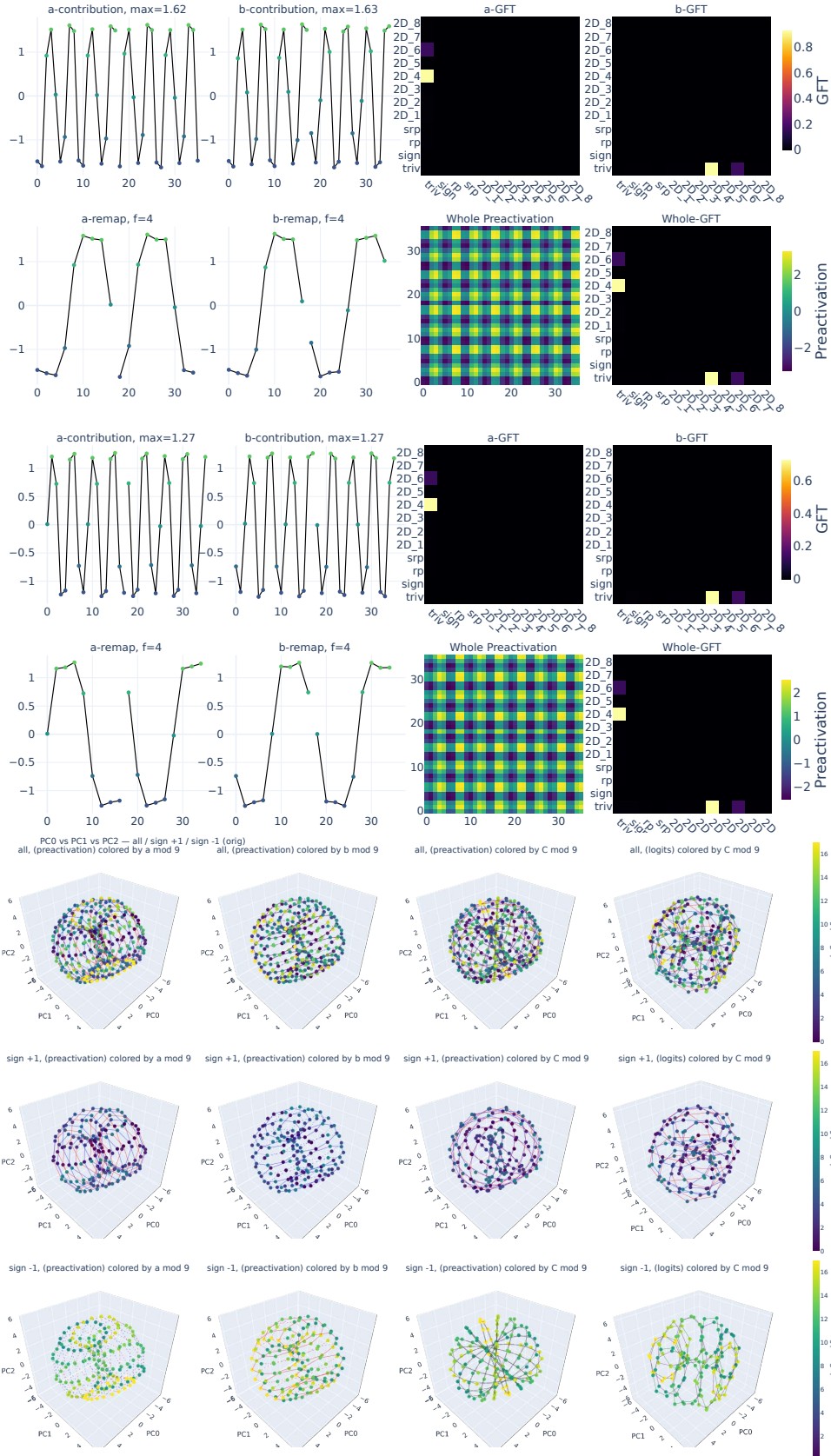

*Figure 24.* Visualization for $n = 18$, Fourier basis $2D_4$. Top: two neurons with highest preactivation in $2D_4$. Bottom: PCA of preactivations and cluster contributions to logits, colored by $a$, $b$, and $C$ modulo $g = n/\gcd(n, f)$, with rotation classes $0, \ldots, g-1$ and reflection classes $g, \ldots, 2g-1$.

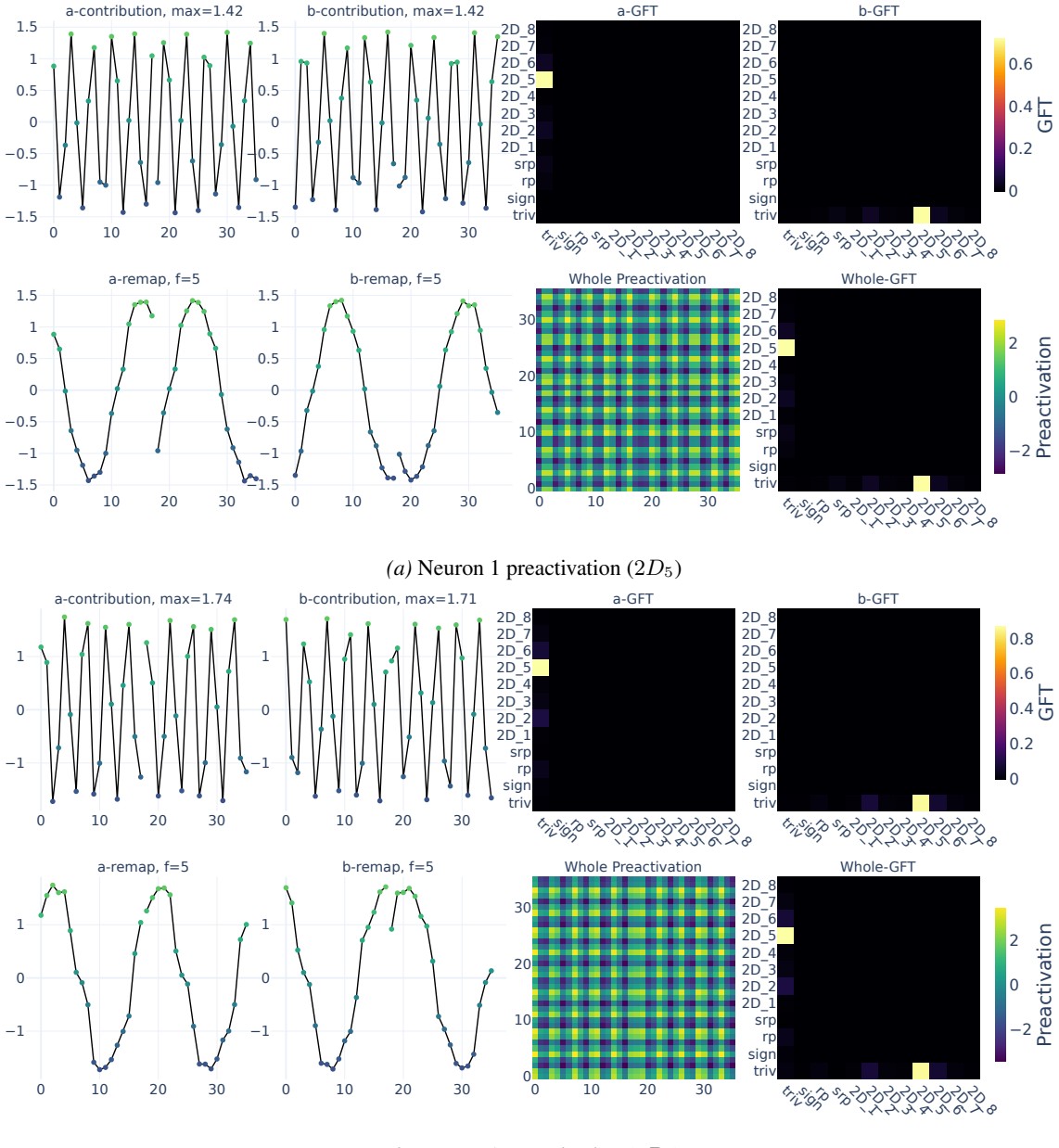

*(a)* Neuron 1 preactivation ($2D_5$)

*(b)* Neuron 2 preactivation ($2D_5$)

*Figure 25.* Visualization for $n = 18$, Fourier basis $2D_5$ (part 1/2).

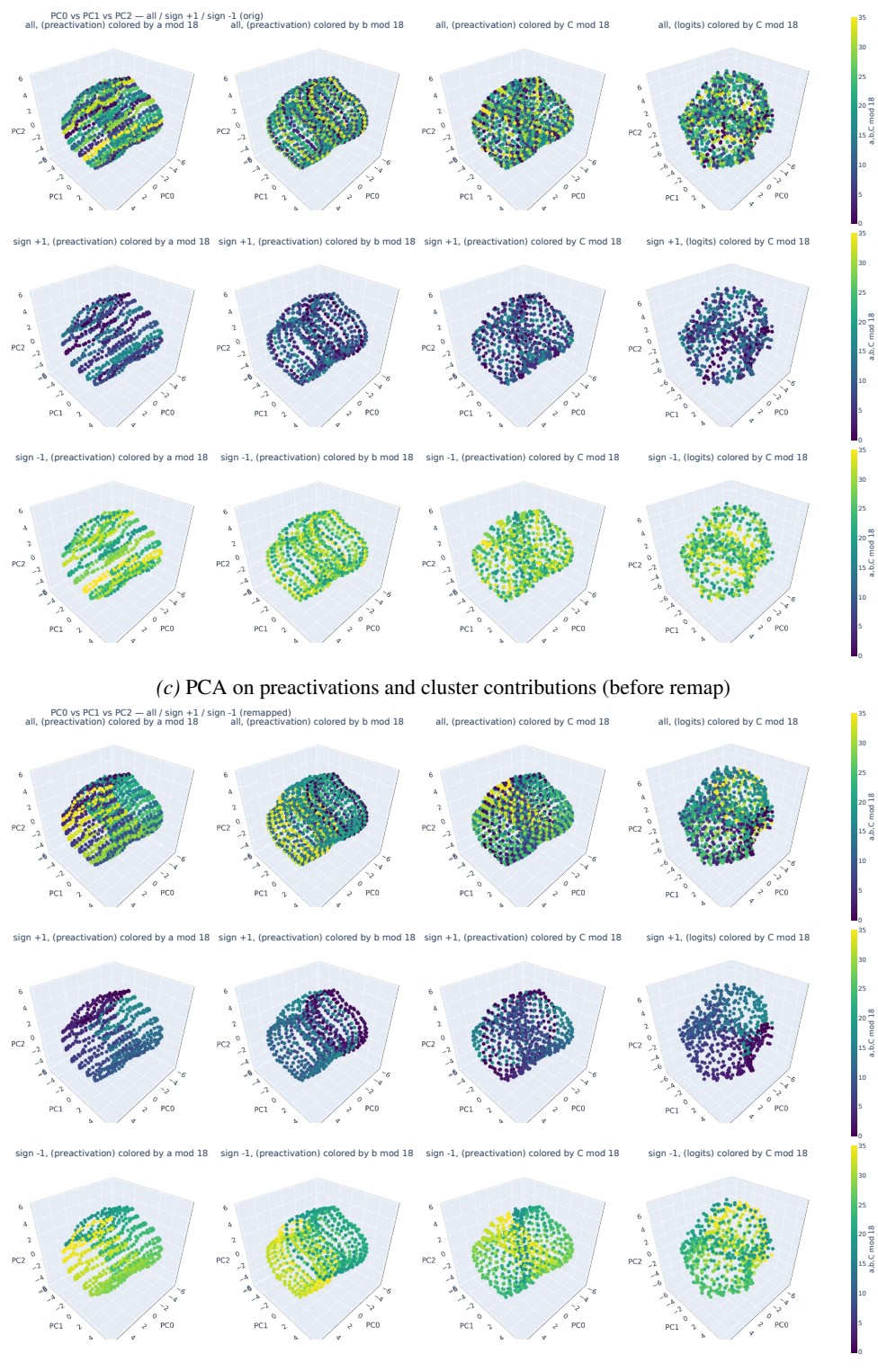

*(c)* PCA on preactivations and cluster contributions (before remap)

*(d)* Same as (c), after remap

*Figure 25.* Visualization for $n = 18$, Fourier basis $2D_5$ (part 2/2). Colored by $a$, $b$, and $C \mod 18$, before (c) and after (d) remapping.

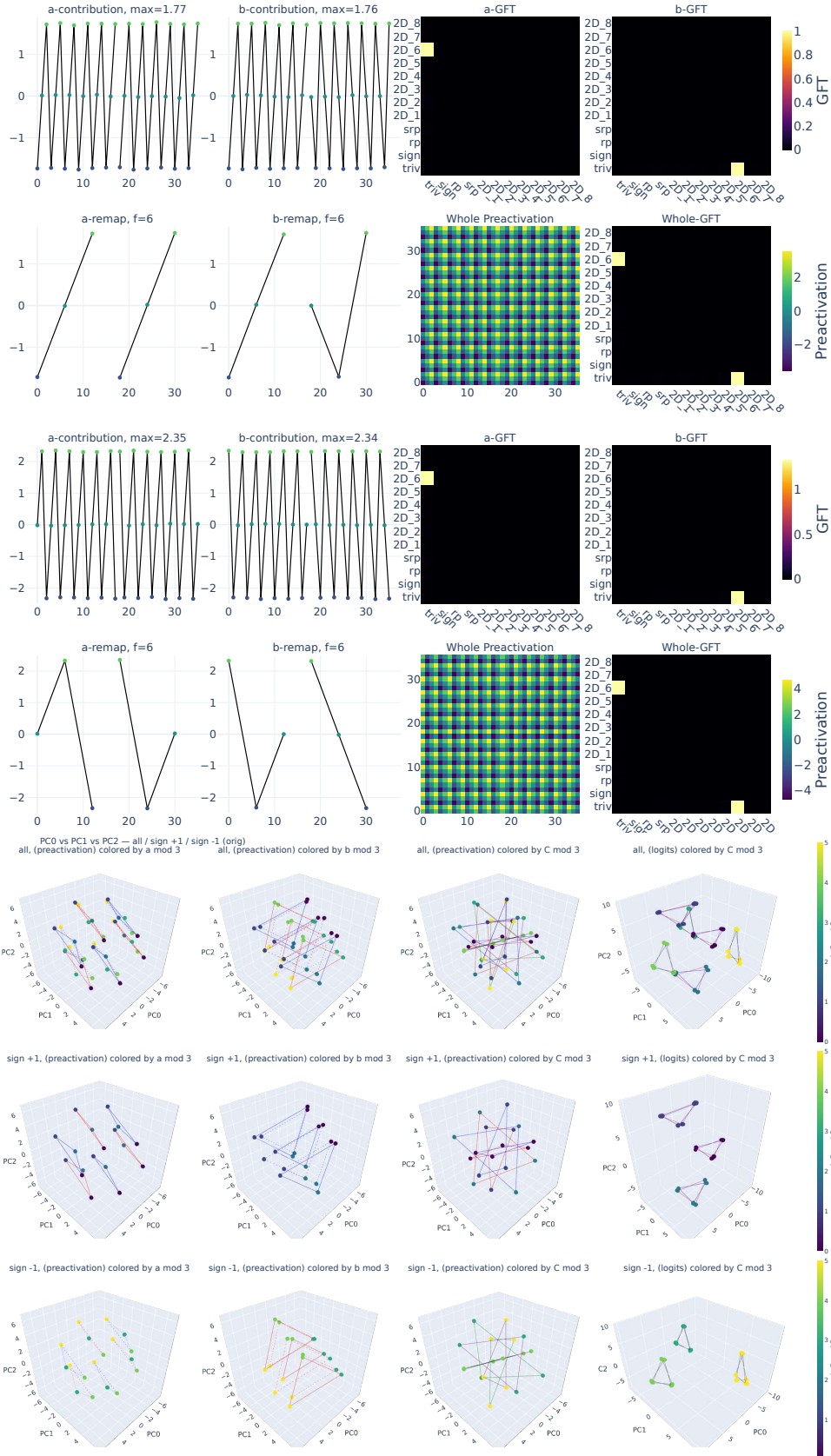

*Figure 26.* Visualization for $n = 18$, Fourier basis $2D_6$. Top: two neurons with highest preactivation in $2D_6$. Bottom: PCA of preactivations and cluster contributions to logits, colored by $a$, $b$, and $C$ modulo $g = n/\gcd(n, f)$, with rotation classes $0, \ldots, g - 1$ and reflection classes $g, \ldots, 2g - 1$.

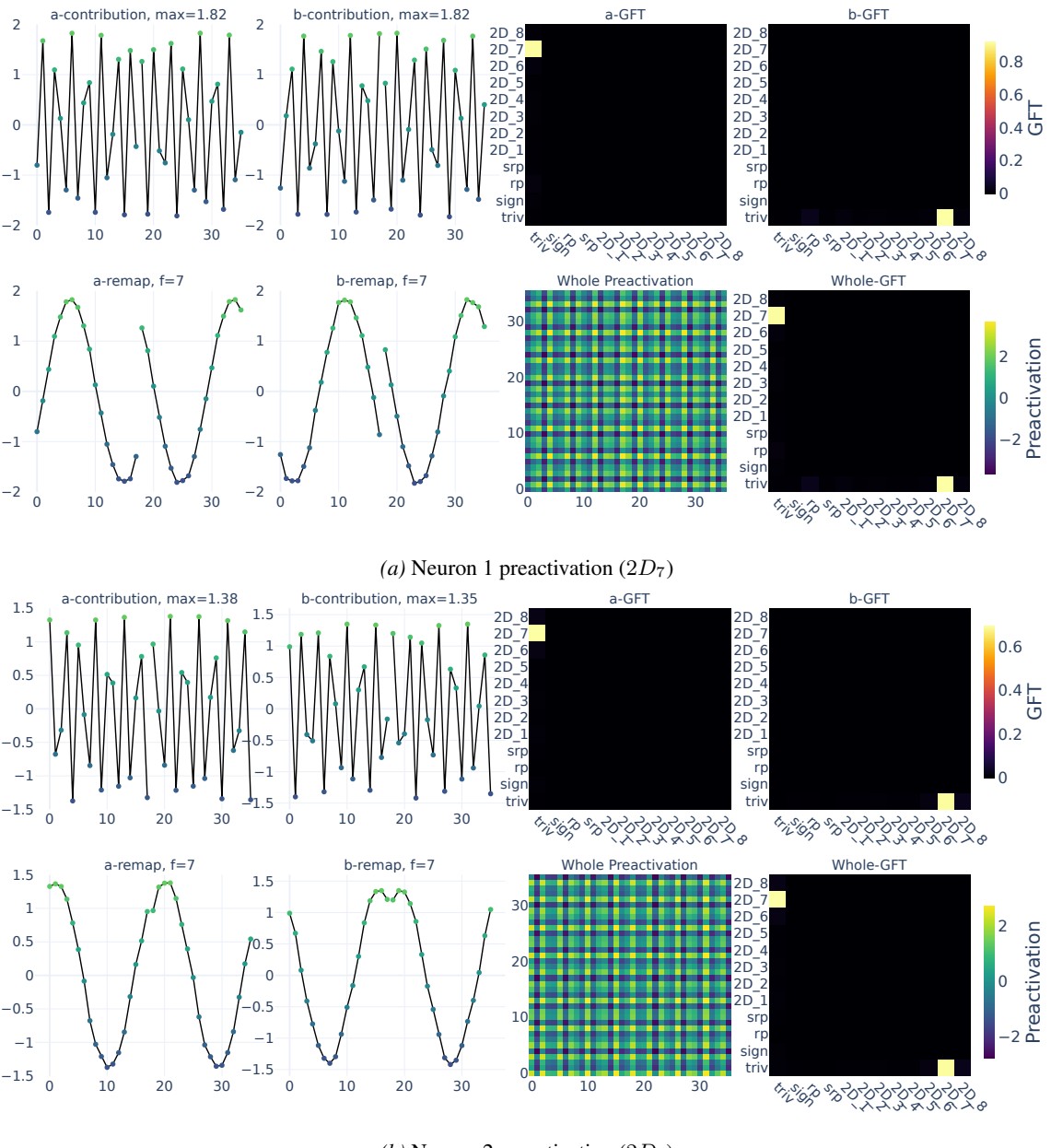

*(a)* Neuron 1 preactivation ($2D_7$)

*(b)* Neuron 2 preactivation ($2D_7$)

*Figure 27.* Visualization for $n = 18$, Fourier basis $2D_7$ (part 1/2). Two neurons with highest preactivation in $2D_7$.

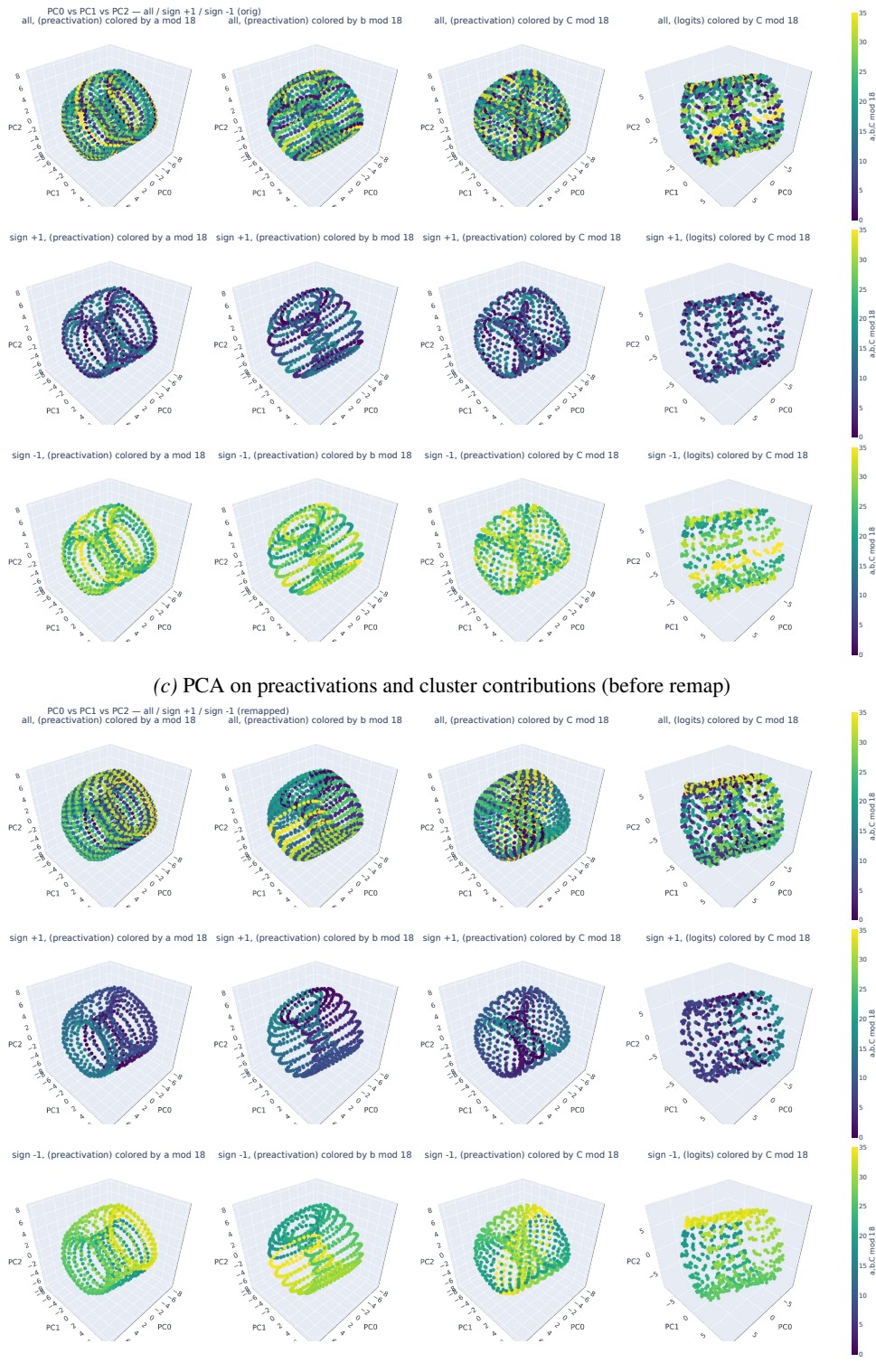

*(c)* PCA on preactivations and cluster contributions (before remap)

*(d)* Same as (c), after remap

*Figure 27.* Visualization for $n = 18$, Fourier basis $2D_7$ (part 2/2). PCA of preactivations and cluster contributions to logits, colored by $a$, $b$, and $C \bmod 18$, shown before and after remapping.

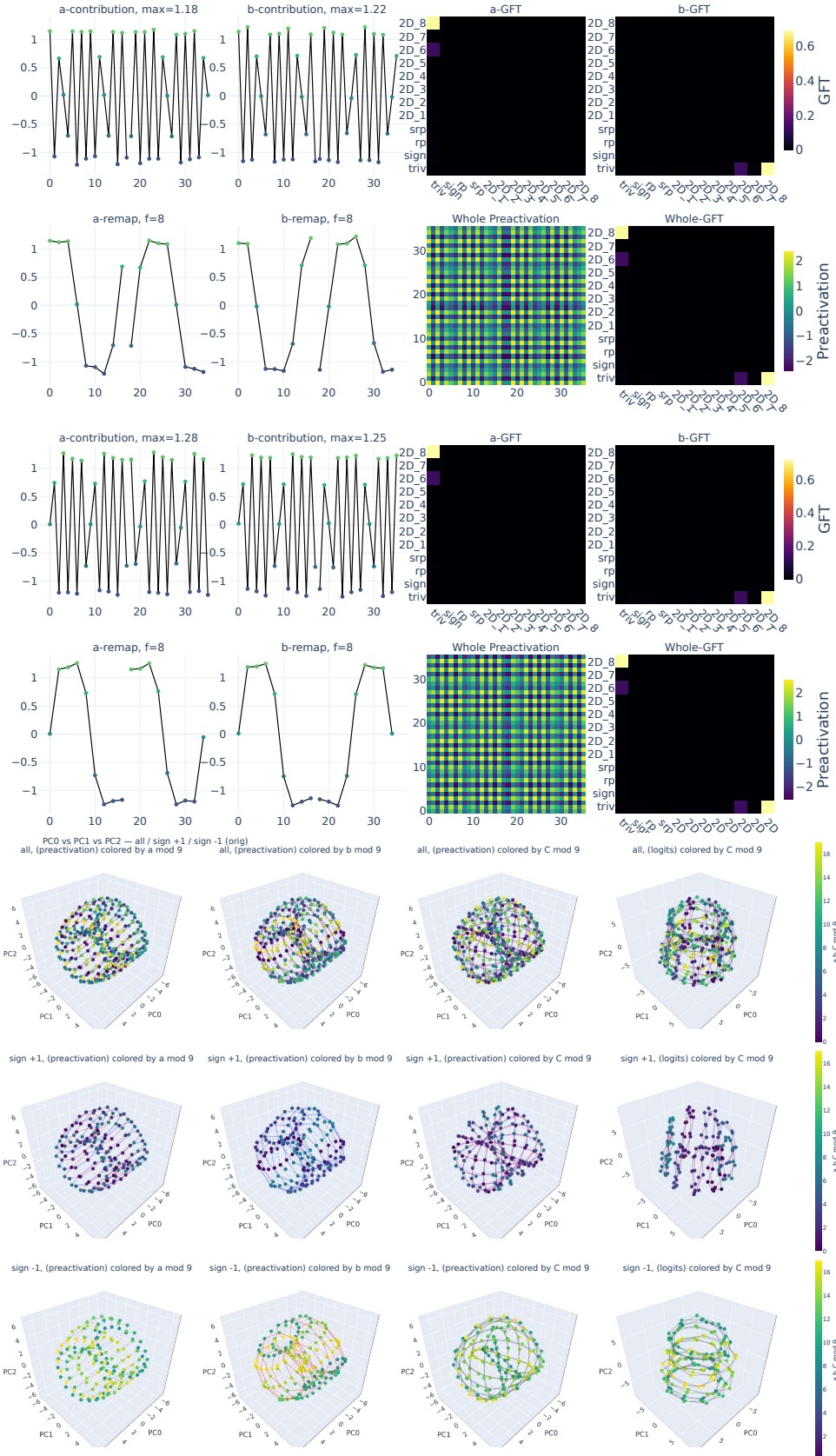

*Figure 28.* Visualization for $n = 18$, Fourier basis $2D_8$. Top: two neurons with highest preactivation in $2D_8$. Bottom: PCA of preactivations and cluster contributions to logits, colored by $a$, $b$, and $C$ modulo $g = n/\gcd(n, f)$, with rotation classes $0, \ldots, g-1$ and reflection classes $g, \ldots, 2g-1$.

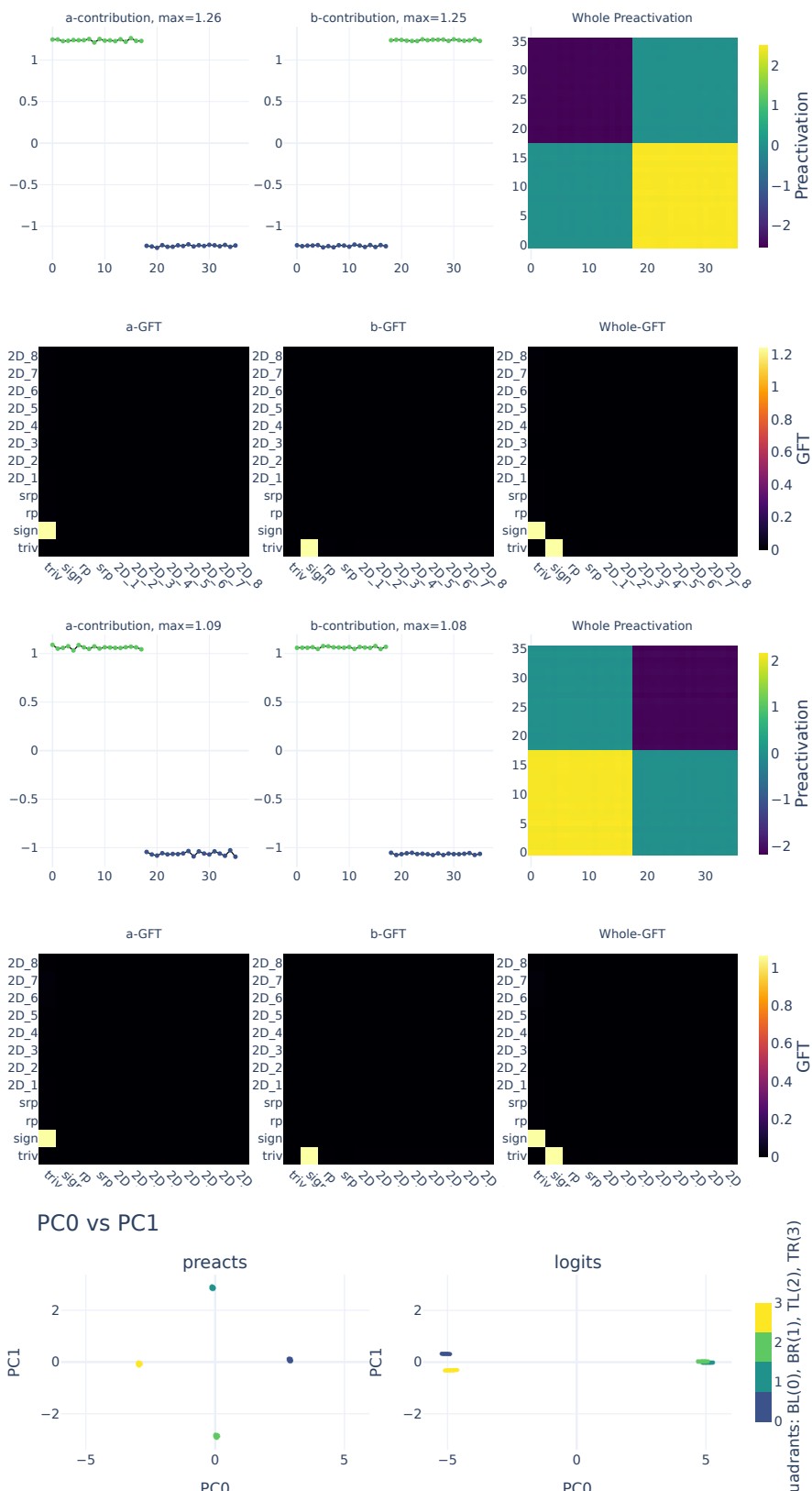

*Figure 29.* Visualization for $n = 18$, Fourier basis `sign`. Top: two neurons with highest preactivation in `sign`. Bottom: PCA of preactivations and cluster contributions to logits, colored by quadrant (BL, BR, TL, TR) of $(a, b)$ in the $2n \times 2n$ grid.

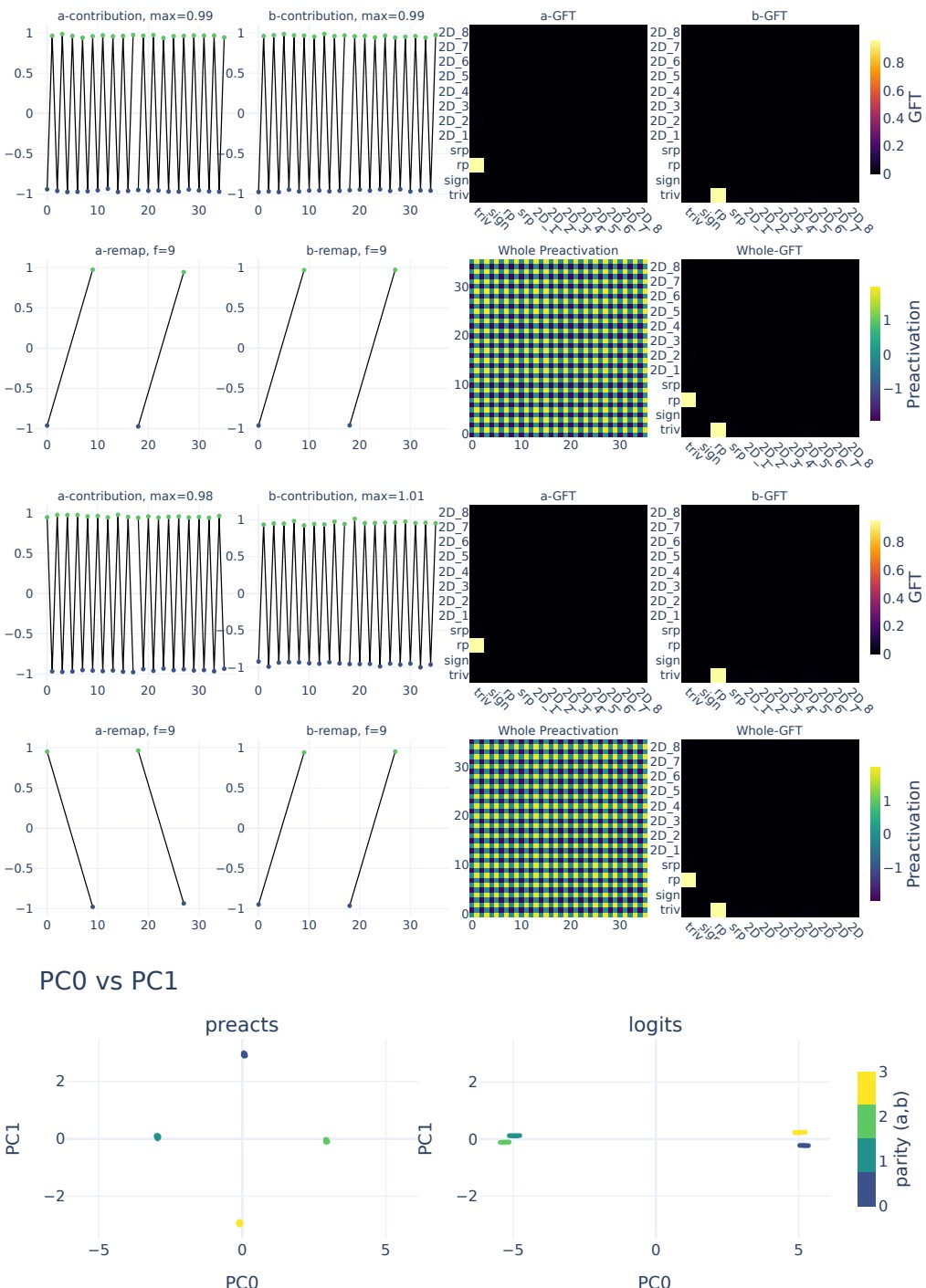

*Figure 30.* Visualization for $n = 18$, Fourier basis `rp`. Top: two neurons with highest preactivation in `rp`. Bottom: PCA of preactivations and cluster contributions to logits, colored by parity of $(a, b)$: code $= 2 \cdot (a \bmod 2) + (b \bmod 2)$.

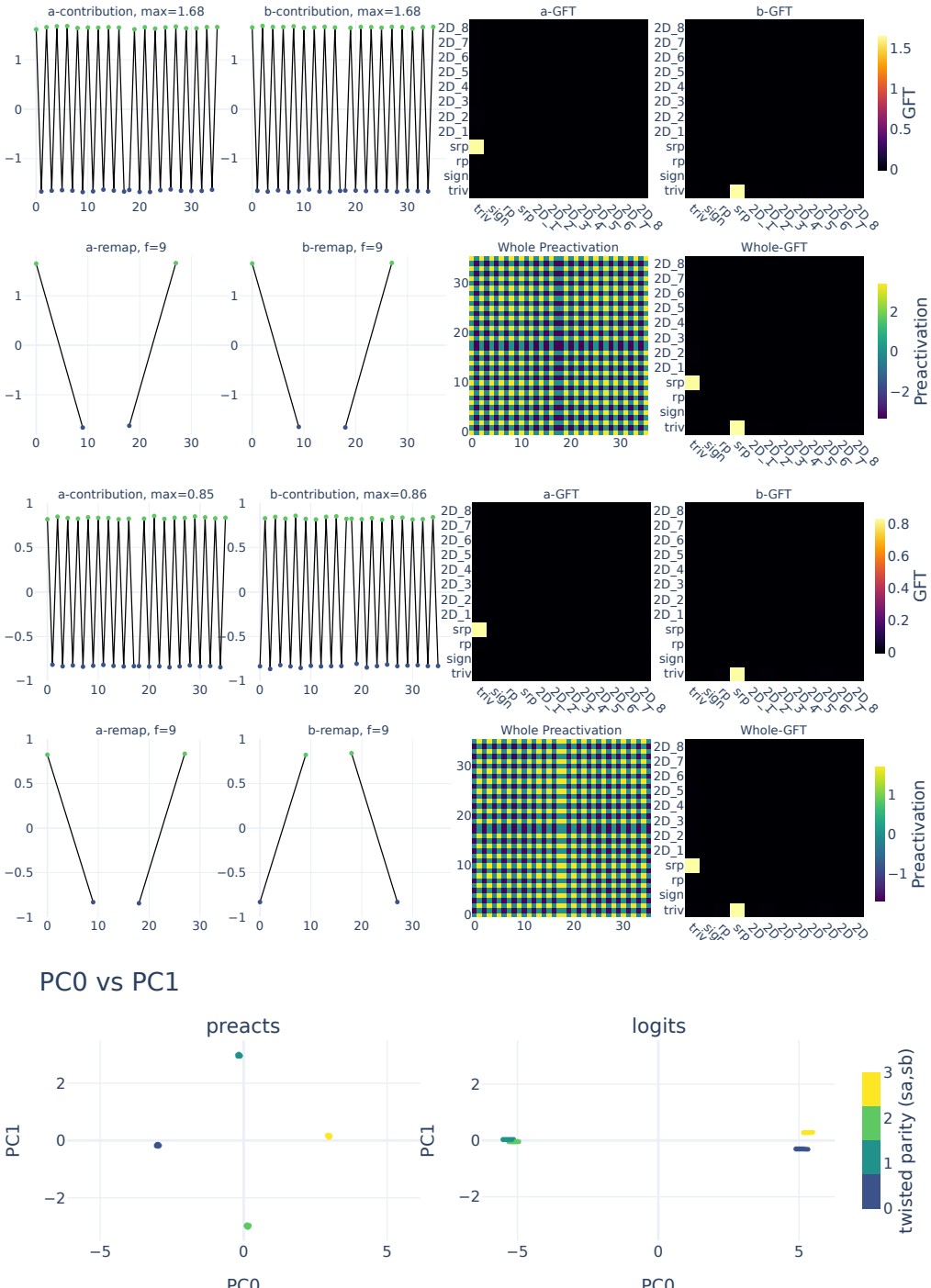

*Figure 31.* Visualization for $n = 18$, Fourier basis `srp`. Top: two neurons with highest preactivation in `srp`. Bottom: PCA of preactivations and cluster contributions to logits, colored by **twisted parity**: $s_a = (a \bmod 2) \oplus \mathbf{1}[a \geq n]$, $s_b = (b \bmod 2) \oplus \mathbf{1}[b \geq n]$, where $x \oplus y = (x + y) \bmod 2$, and code $= 2s_a + s_b$.

### D.3. Library of neuron classes that occur in $D_{19}$

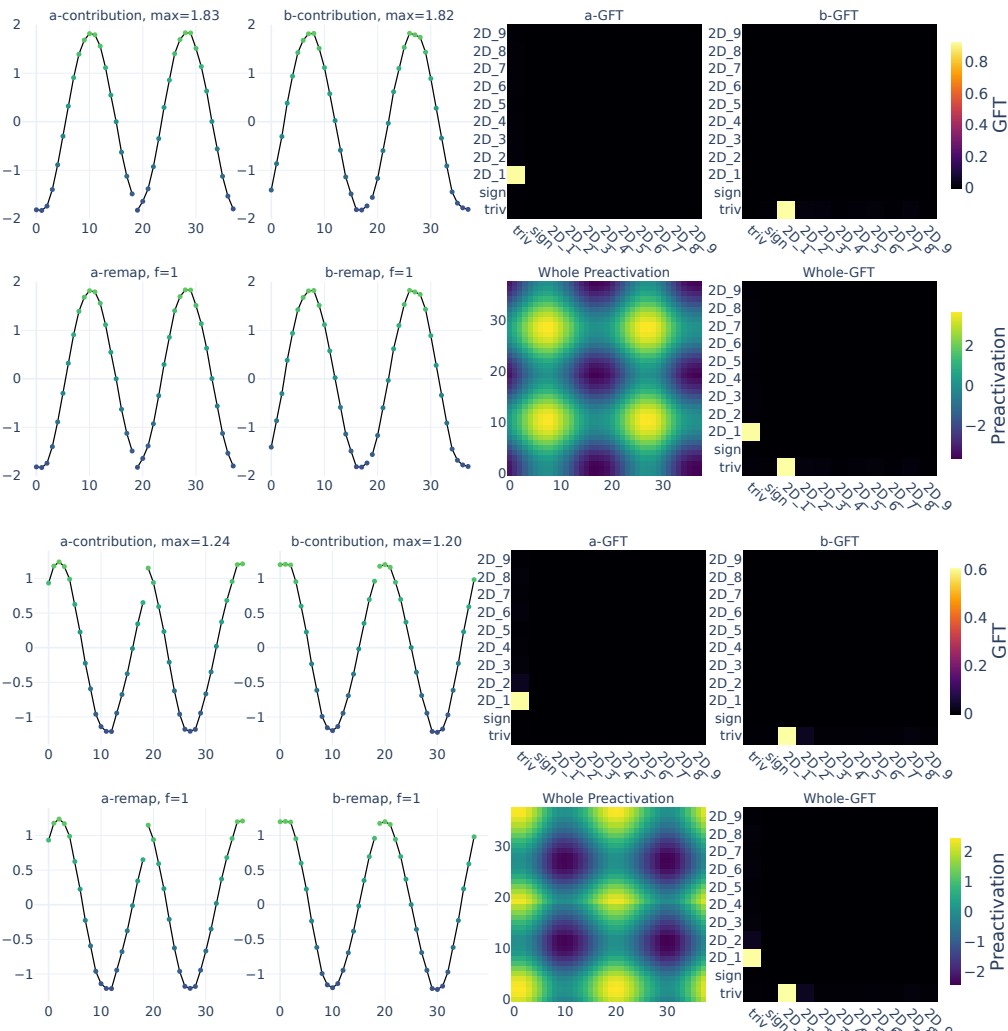

*Figure 32.* Visualization for $n = 19$, Fourier basis $2D_1$. Two neurons with highest preactivation in $2D_1$.

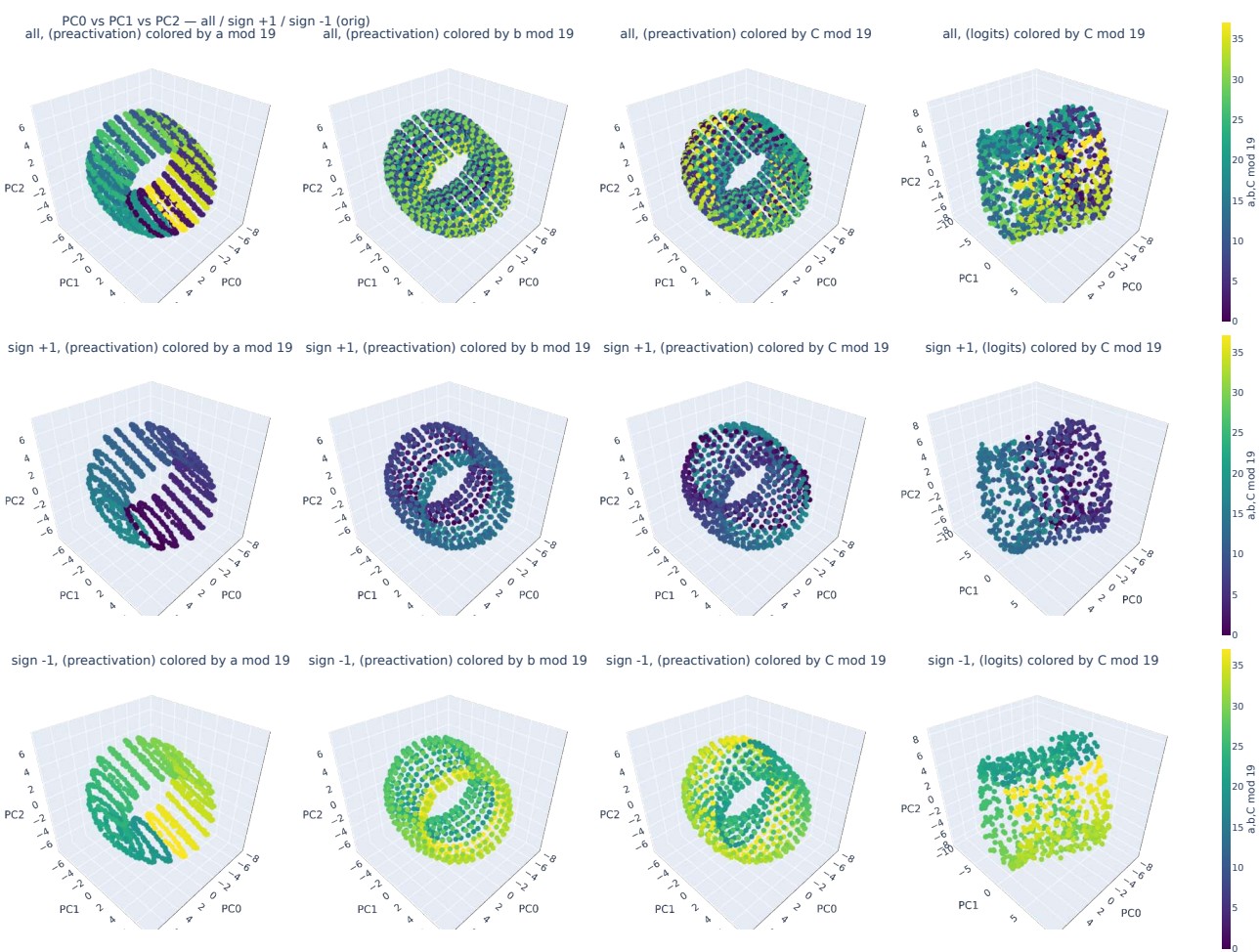

*Figure 33.* Visualization for $n = 19$, Fourier basis $2D_1$. PCA of preactivations and cluster contributions to logits, colored by $a$, $b$, and $C \bmod 19$ (rotation classes 0–18, reflection classes 19–37).

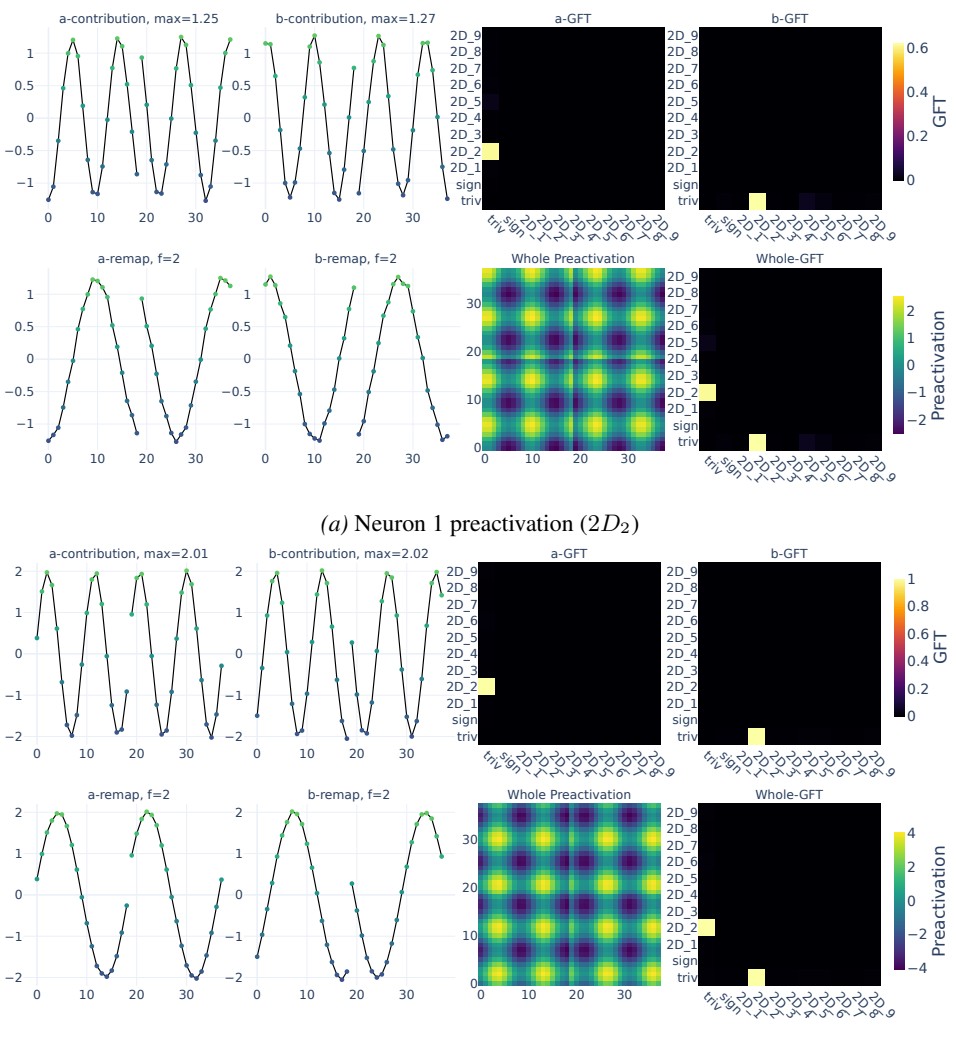

*(a)* Neuron 1 preactivation ($2D_2$)

*(b)* Neuron 2 preactivation ($2D_2$)

*Figure 34.* Visualization for $n = 19$, Fourier basis $2D_2$ (part 1/2). Two neurons with highest preactivation in $2D_2$.

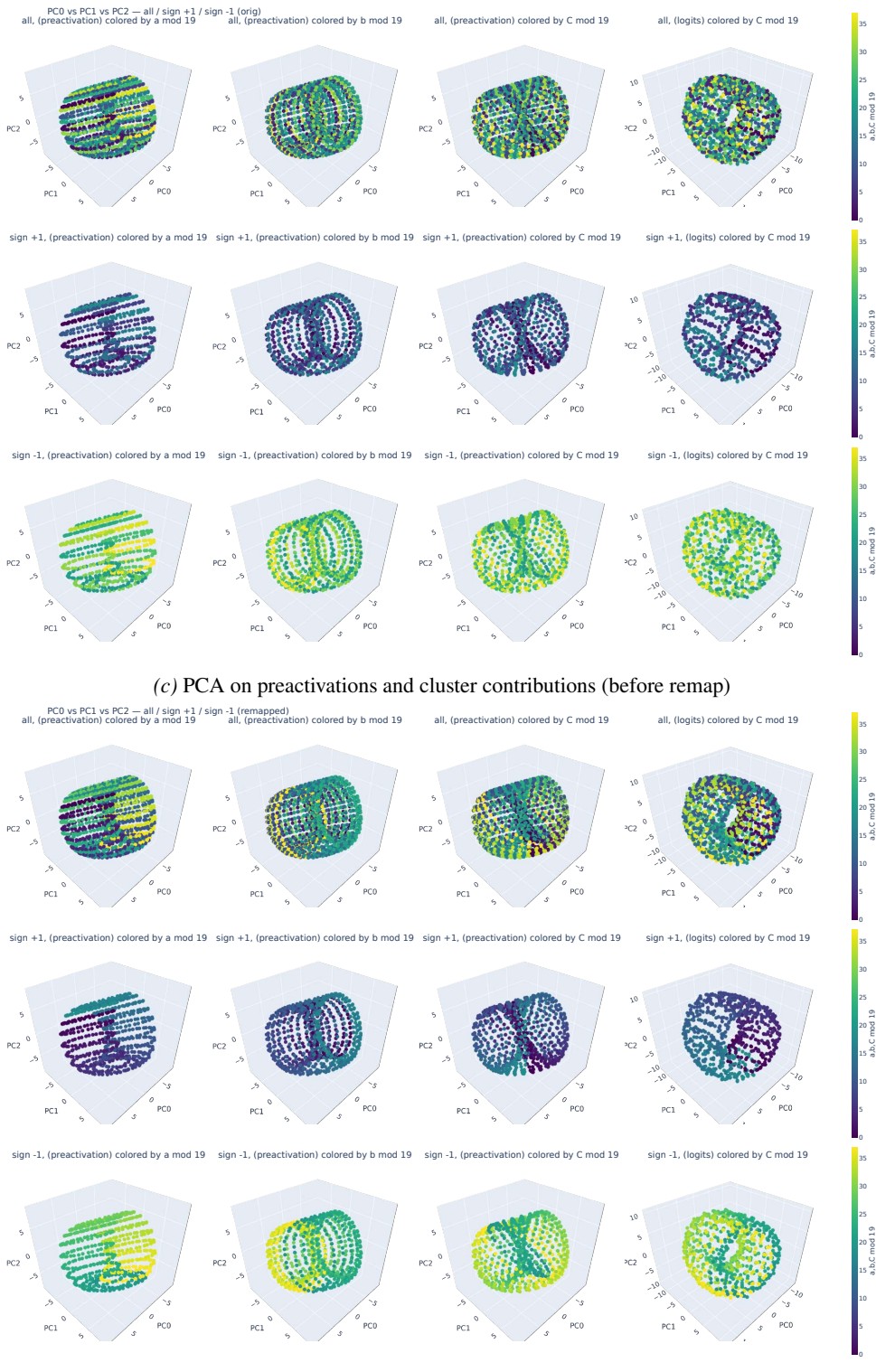

*(c)* PCA on preactivations and cluster contributions (before remap)

*(d)* Same as (c), after remap

*Figure 34.* Visualization for $n = 19$, Fourier basis $2D_2$ (part 2/2). PCA of preactivations and cluster contributions to logits, colored by $a$, $b$, and $C \bmod 19$, shown before and after remapping.

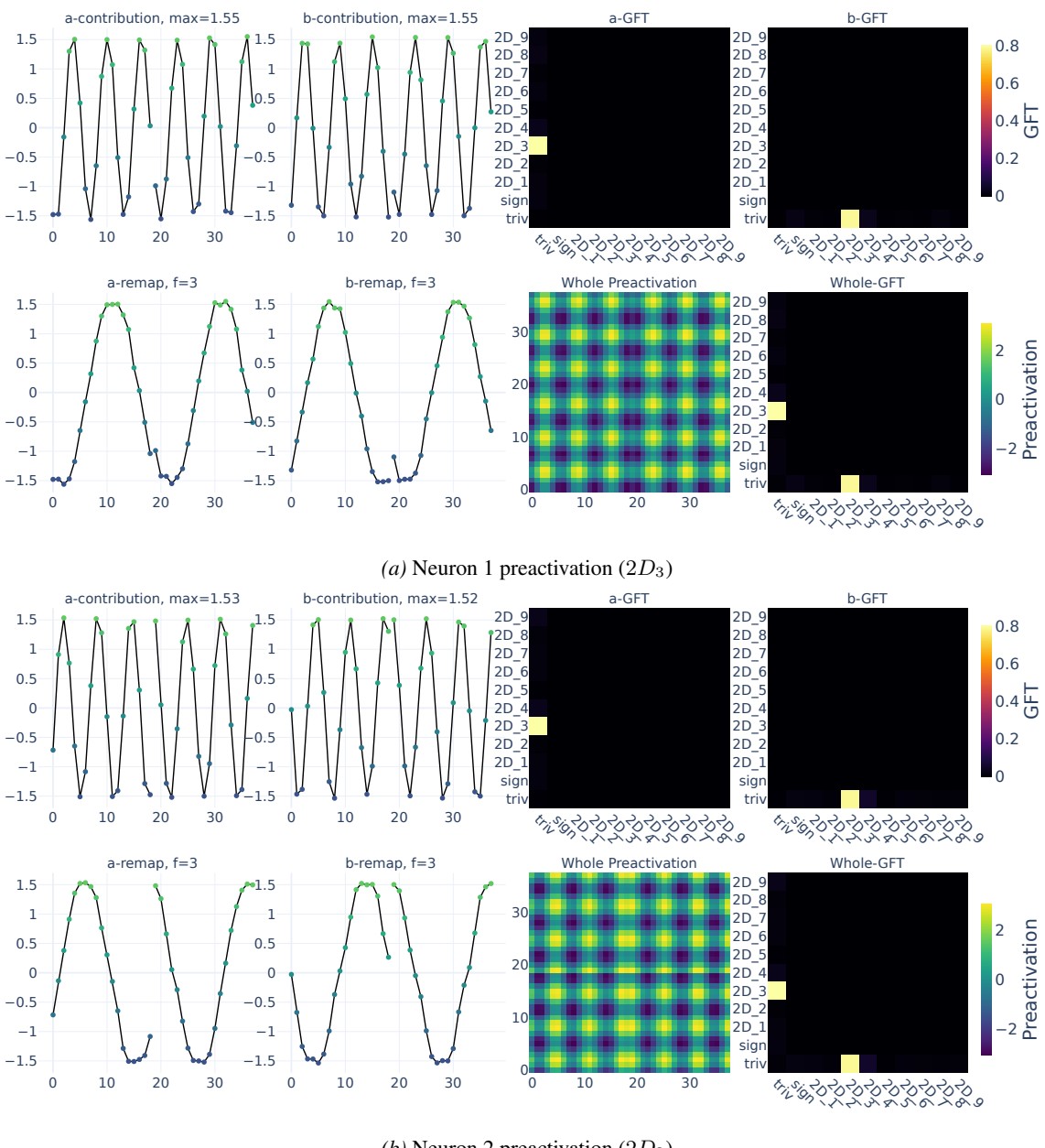

*(a)* Neuron 1 preactivation ($2D_3$)

*(b)* Neuron 2 preactivation ($2D_3$)

*Figure 35.* Visualization for $n = 19$, Fourier basis $2D_3$ (part 1/2). Two neurons with highest preactivation in $2D_3$.

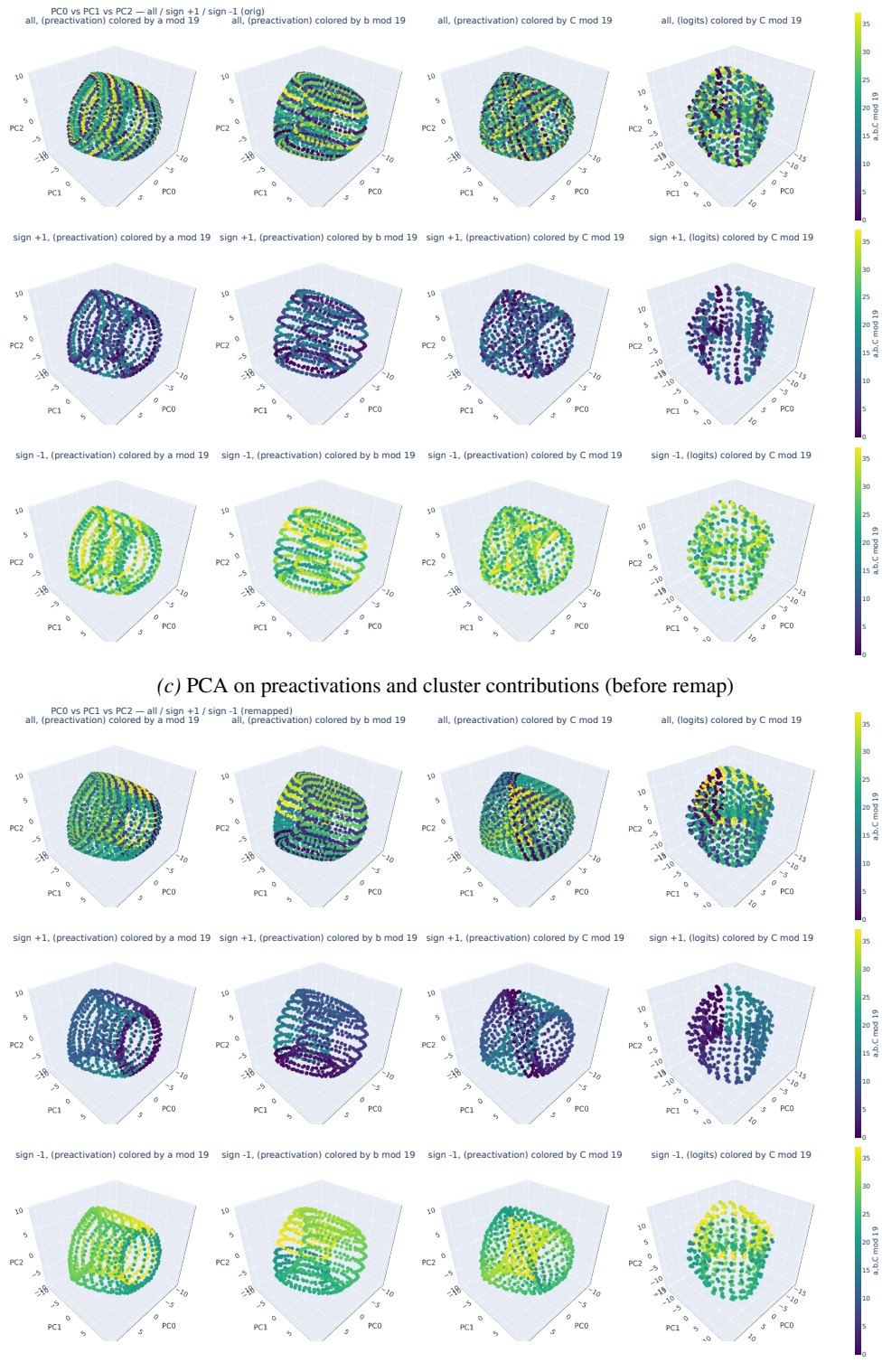

*(c)* PCA on preactivations and cluster contributions (before remap)

*(d)* Same as (c), after remap

*Figure 35.* Visualization for $n = 19$, Fourier basis $2D_3$ (part 2/2). PCA of preactivations and cluster contributions to logits, colored by $a$, $b$, and $C \bmod 19$, shown before and after remapping.

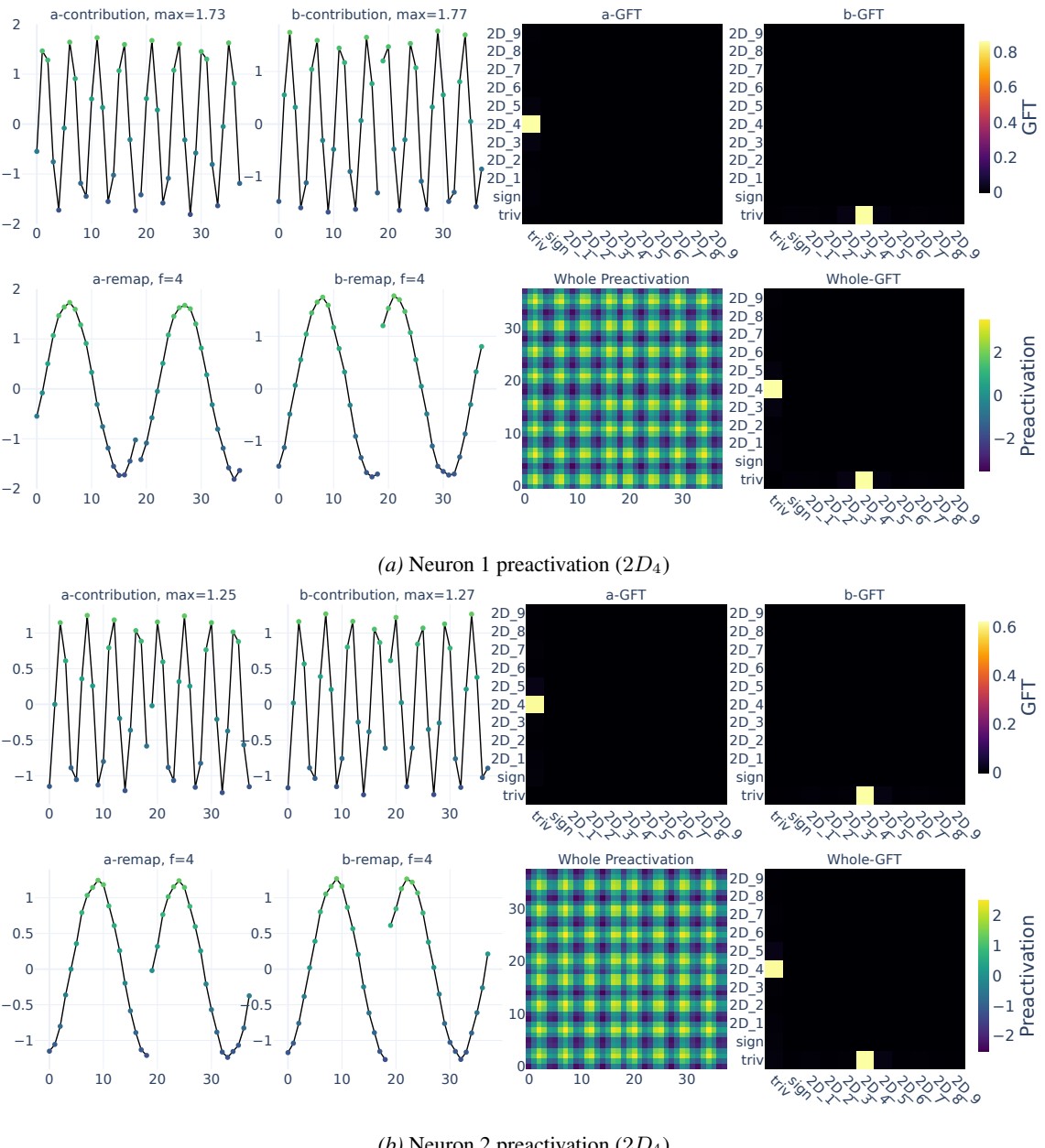

*(a)* Neuron 1 preactivation ($2D_4$)

*(b)* Neuron 2 preactivation ($2D_4$)

*Figure 36.* Visualization for $n = 19$, Fourier basis $2D_4$ (part 1/2). Two neurons with highest preactivation in $2D_4$.

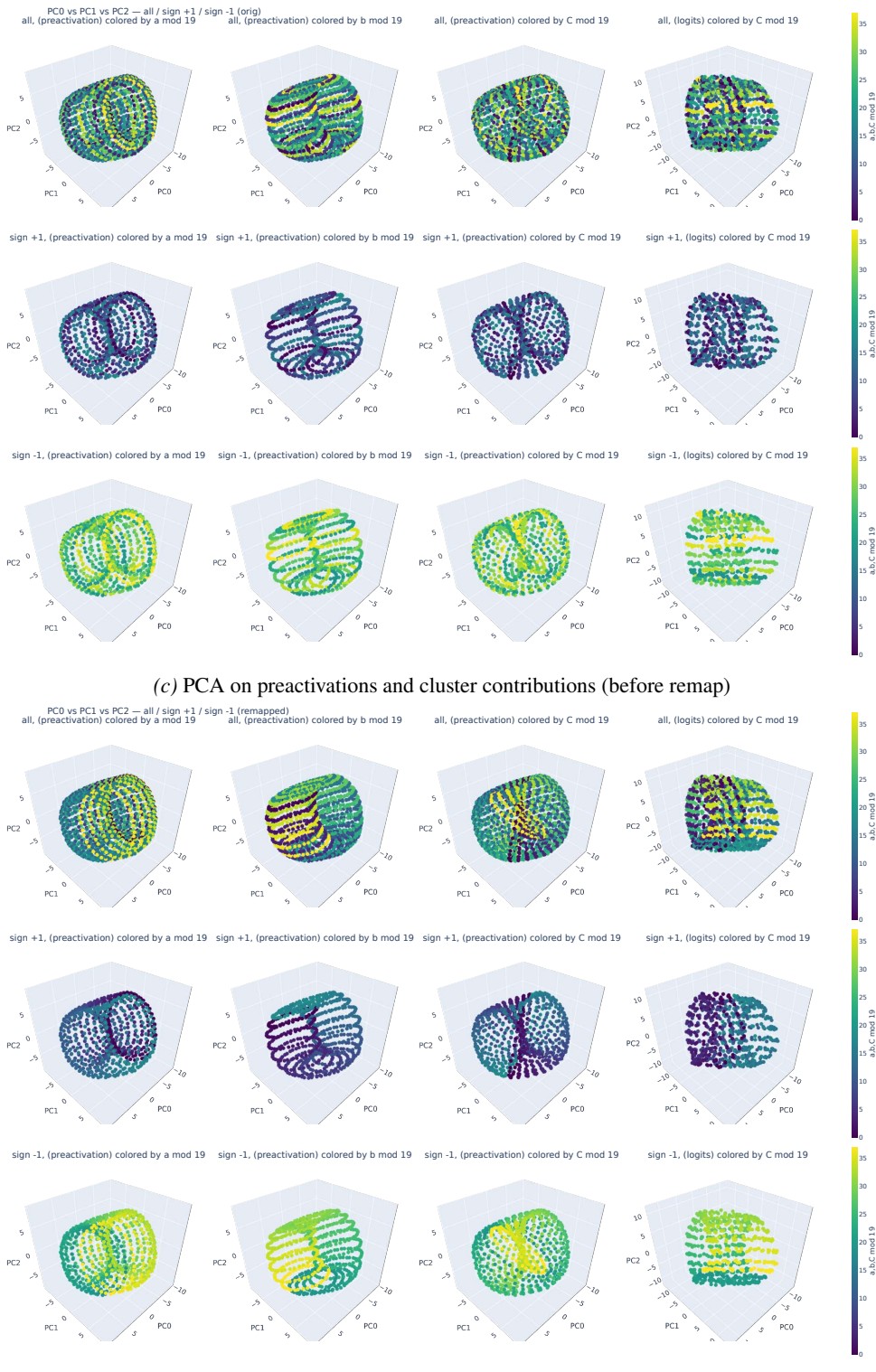

*(c)* PCA on preactivations and cluster contributions (before remap)

*(d)* Same as (c), after remap

*Figure 36.* Visualization for $n = 19$, Fourier basis $2D_4$ (part 2/2). PCA of preactivations and cluster contributions to logits, colored by $a$, $b$, and $C \bmod 19$, shown before and after remapping.

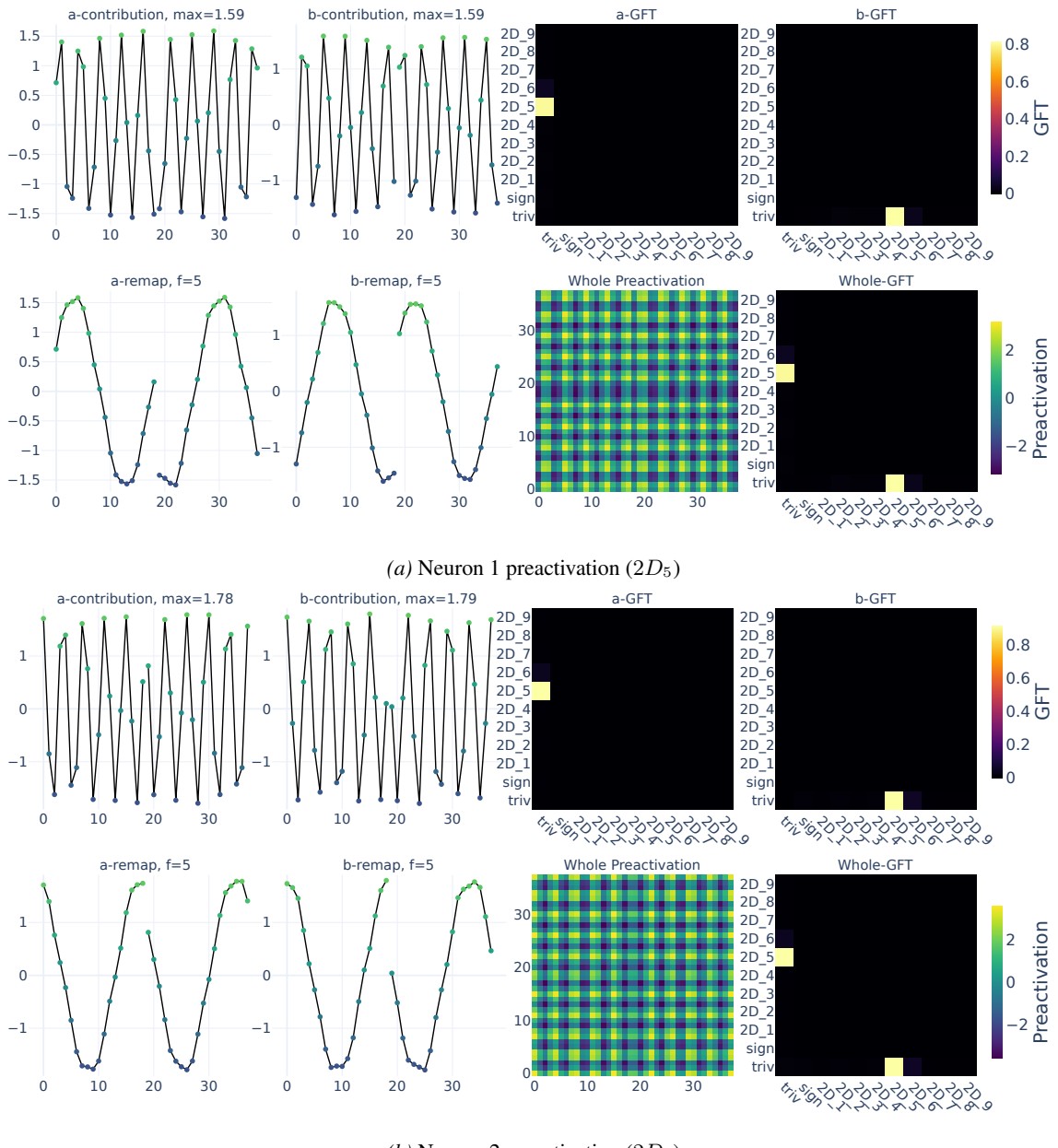

*(a)* Neuron 1 preactivation ($2D_5$)

*(b)* Neuron 2 preactivation ($2D_5$)

*Figure 37.* Visualization for $n = 19$, Fourier basis $2D_5$ (part 1/2). Two neurons with highest preactivation in $2D_5$.

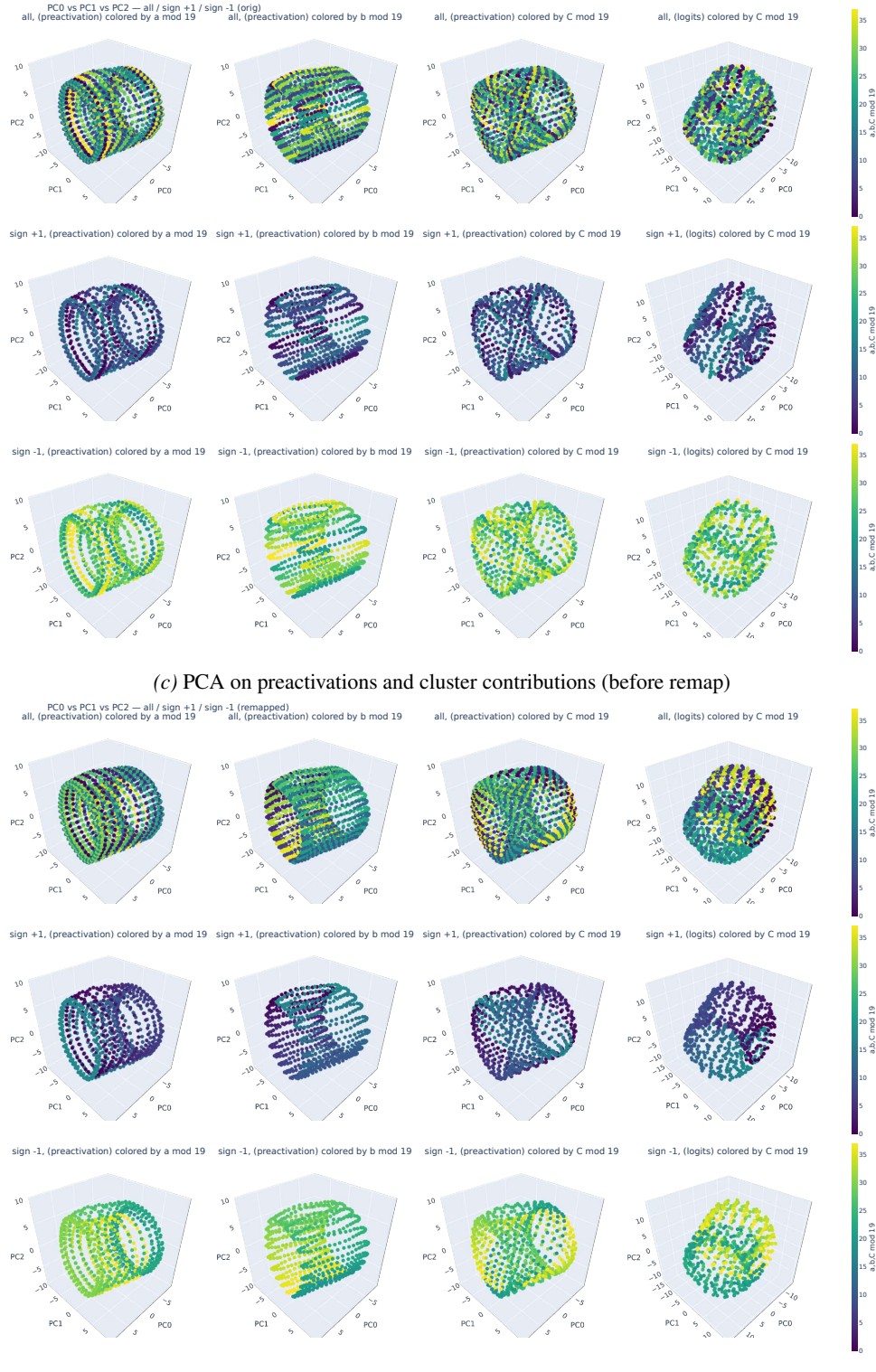

*(c)* PCA on preactivations and cluster contributions (before remap)

*(d)* Same as (c), after remap

*Figure 37.* Visualization for $n = 19$, Fourier basis $2D_5$ (part 2/2). PCA of preactivations and cluster contributions to logits, colored by $a$, $b$, and $C \bmod 19$, shown before and after remapping.

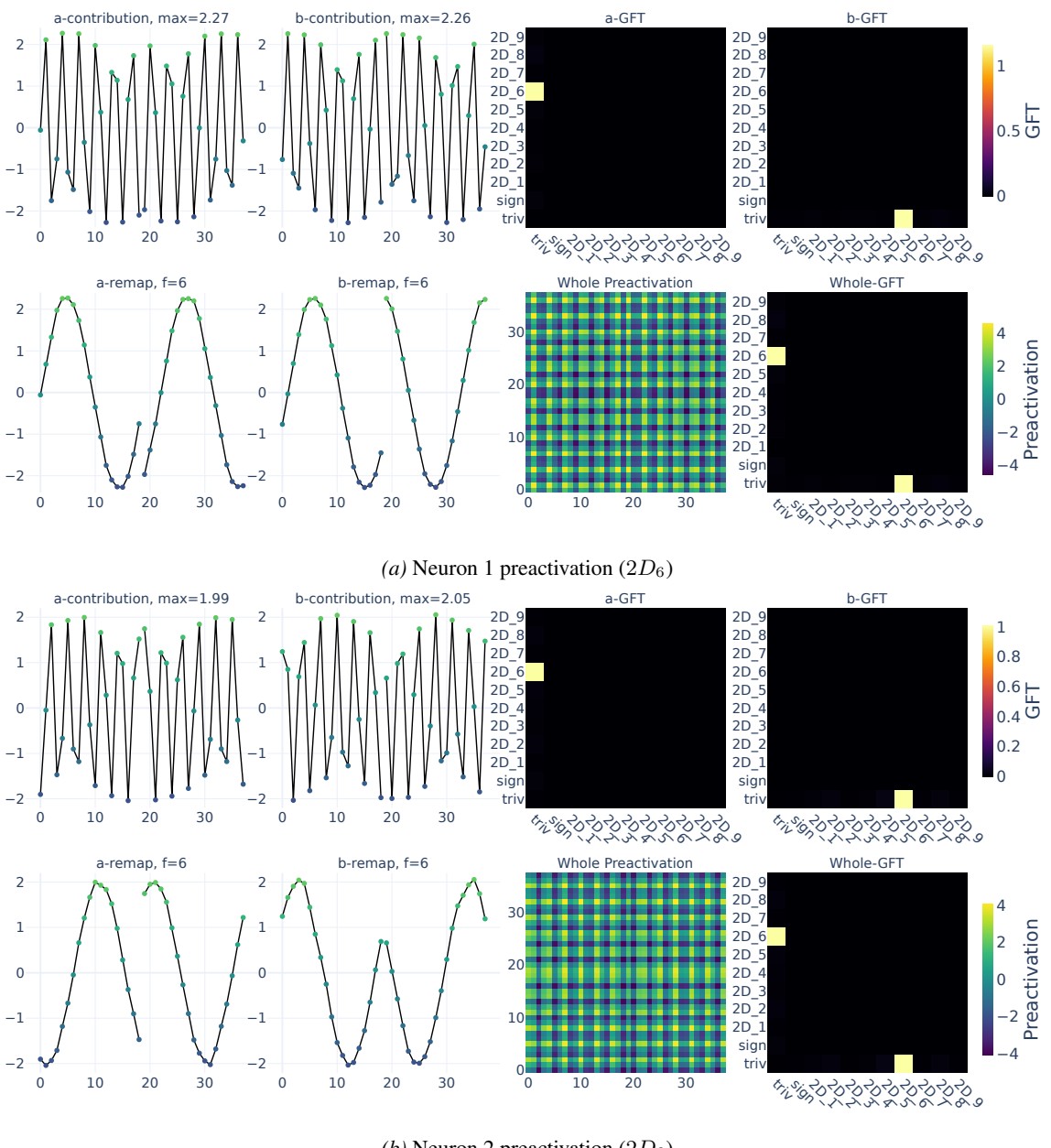

*(a)* Neuron 1 preactivation $(2D_6)$

*(b)* Neuron 2 preactivation $(2D_6)$

*Figure 38.* Visualization for $n = 19$, Fourier basis $2D_6$ (part 1/2). Two neurons with highest preactivation in $2D_6$.

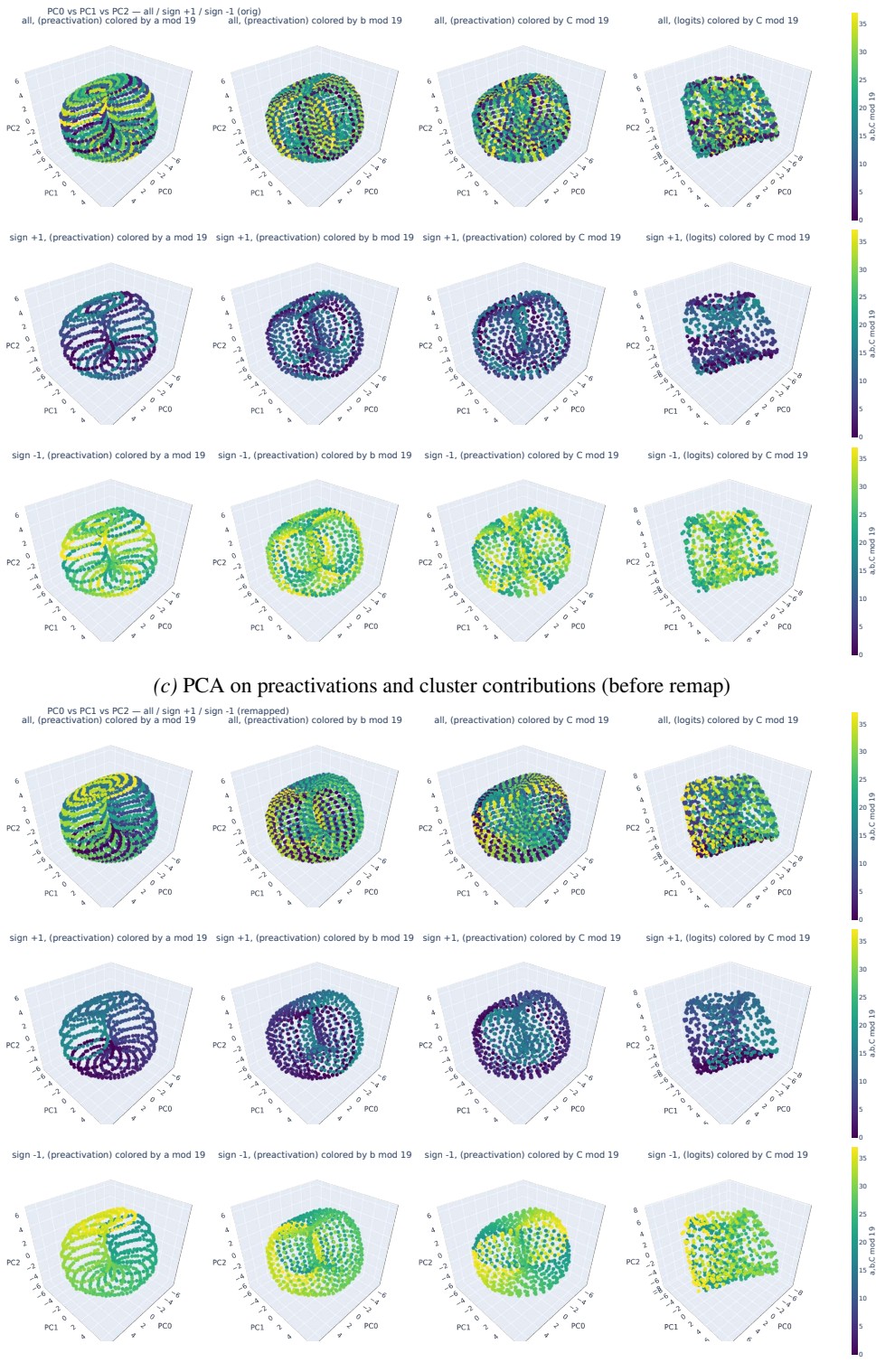

*(c)* PCA on preactivations and cluster contributions (before remap)

*(d)* Same as (c), after remap

*Figure 38.* Visualization for $n = 19$, Fourier basis $2D_6$ (part 2/2). PCA of preactivations and cluster contributions to logits, colored by $a$, $b$, and $C \bmod 19$, shown before and after remapping.

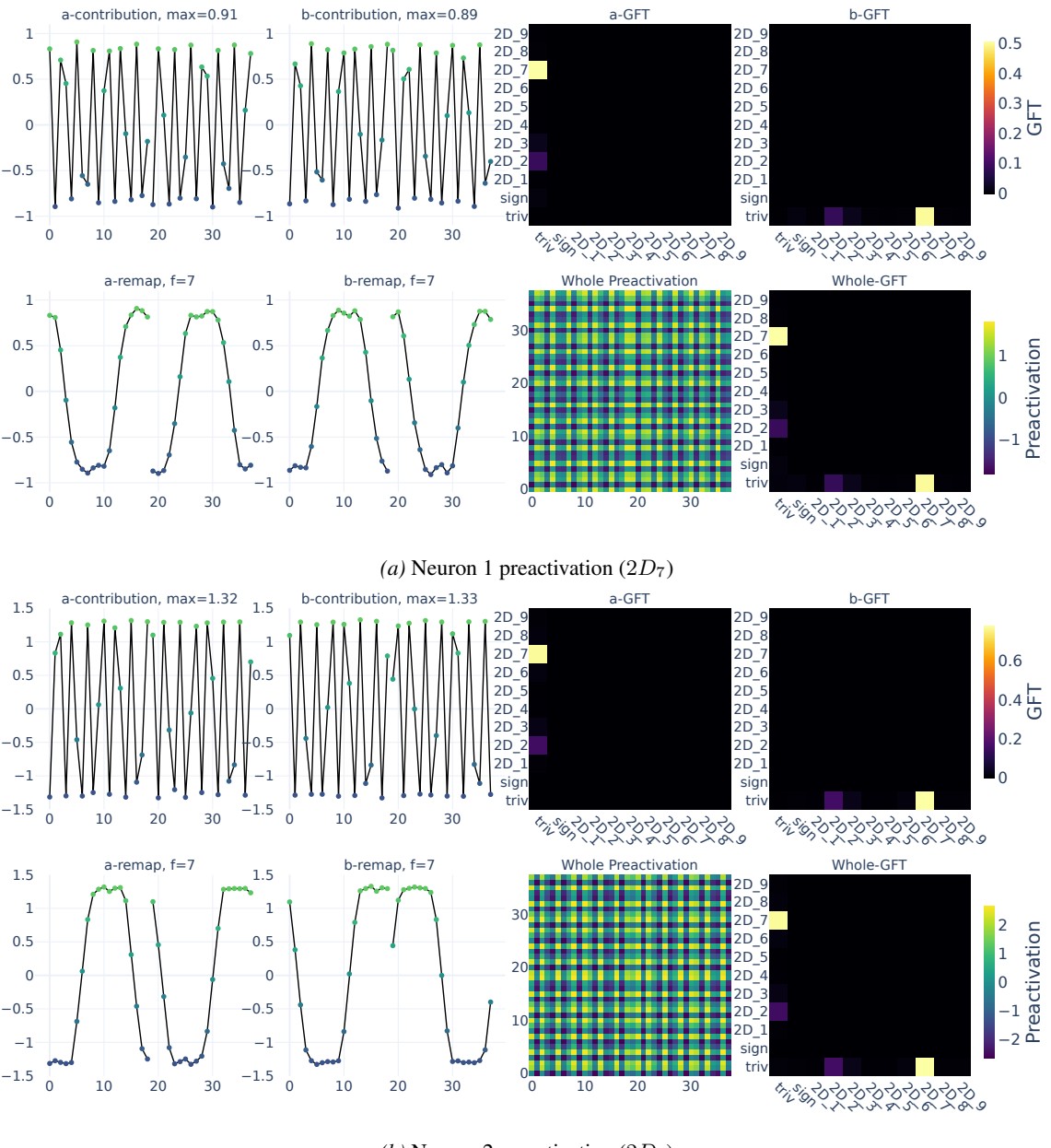

*(a)* Neuron 1 preactivation ($2D_7$)

*(b)* Neuron 2 preactivation ($2D_7$)

*Figure 39.* Visualization for $n = 19$, Fourier basis $2D_7$ (part 1/2). Two neurons with highest preactivation in $2D_7$.

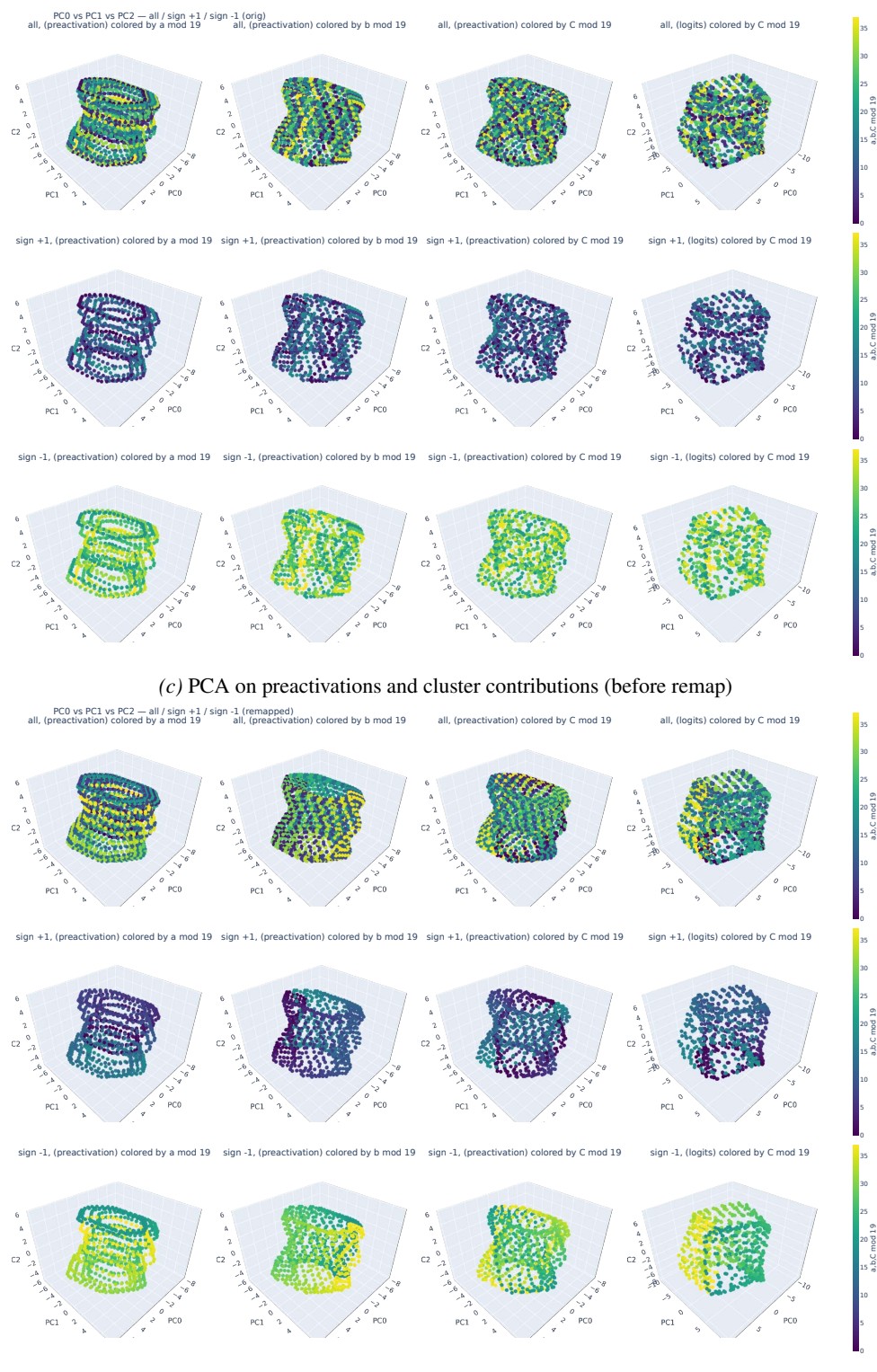

*(c)* PCA on preactivations and cluster contributions (before remap)

*(d)* Same as (c), after remap

*Figure 39.* Visualization for $n = 19$, Fourier basis $2D_7$ (part 2/2). PCA of preactivations and cluster contributions to logits, colored by $a$, $b$, and $C \bmod 19$, shown before and after remapping.

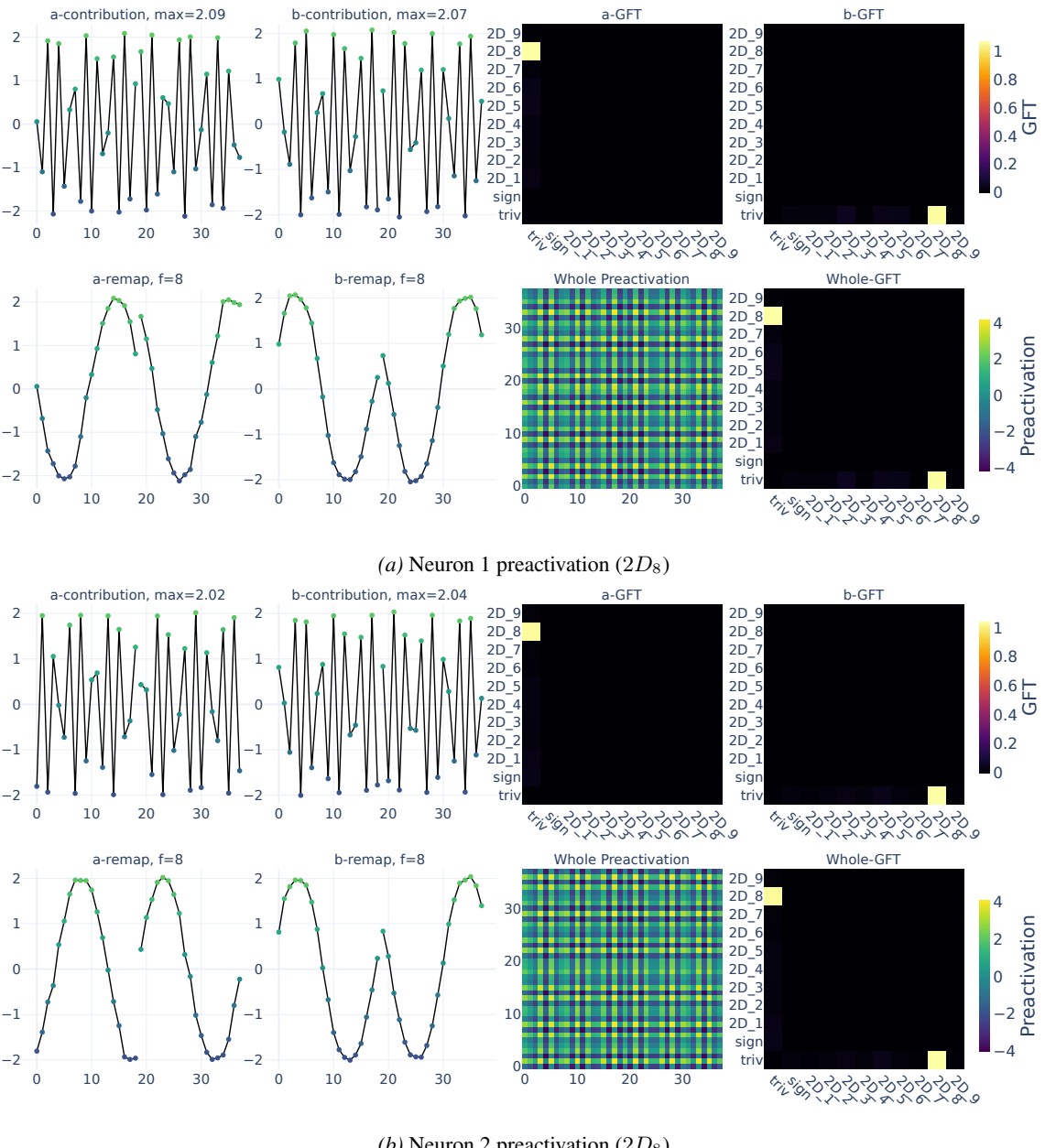

*(a)* Neuron 1 preactivation ($2D_8$)

*(b)* Neuron 2 preactivation ($2D_8$)

*Figure 40.* Visualization for $n = 19$, Fourier basis $2D_8$ (part 1/2). Two neurons with highest preactivation in $2D_8$.

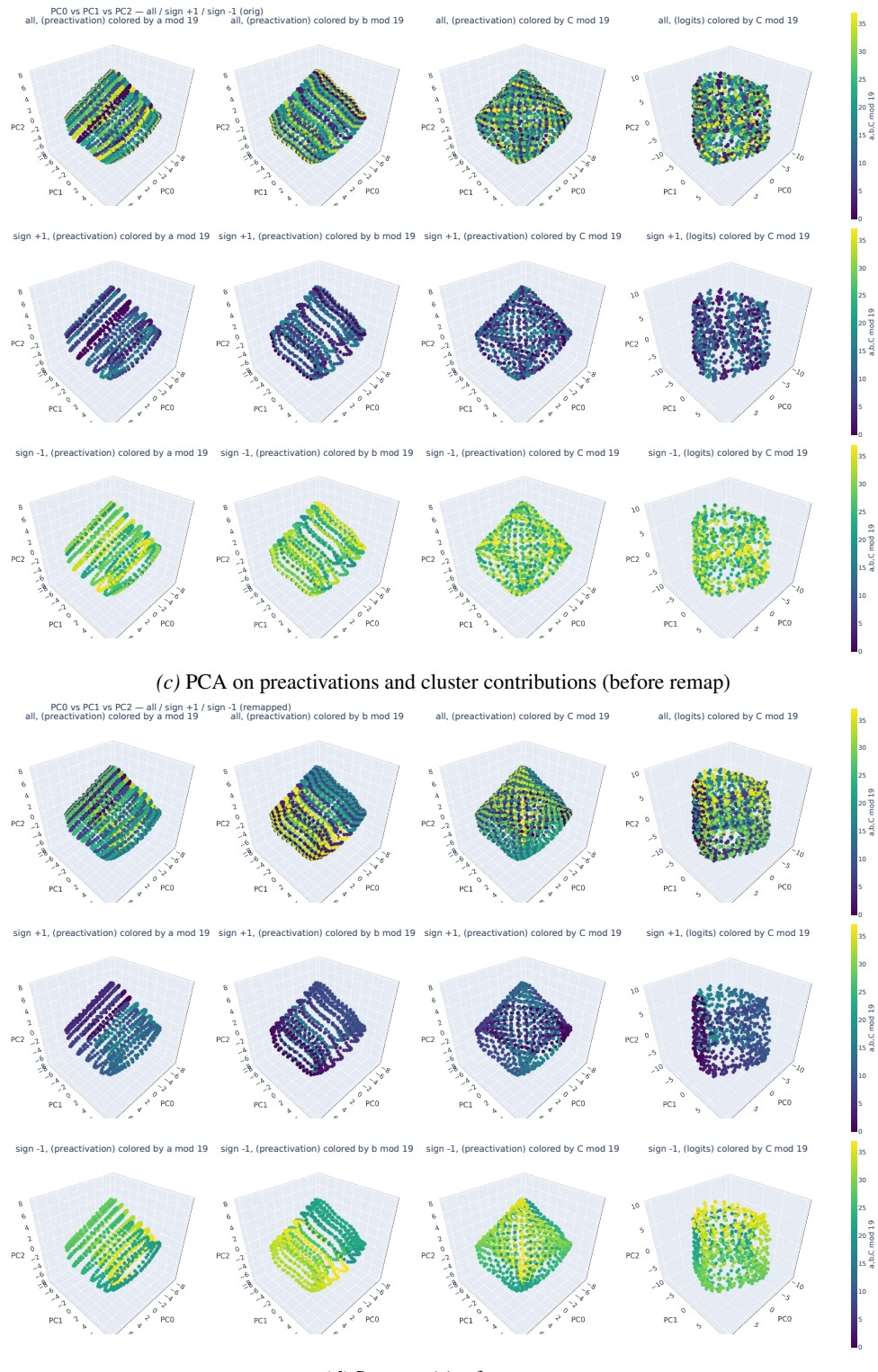

*(c)* PCA on preactivations and cluster contributions (before remap)

*(d)* Same as (c), after remap

*Figure 40.* Visualization for $n = 19$, Fourier basis $2D_8$ (part 2/2). PCA of preactivations and cluster contributions to logits, colored by $a$, $b$, and $C \mod 19$, shown before and after remapping.

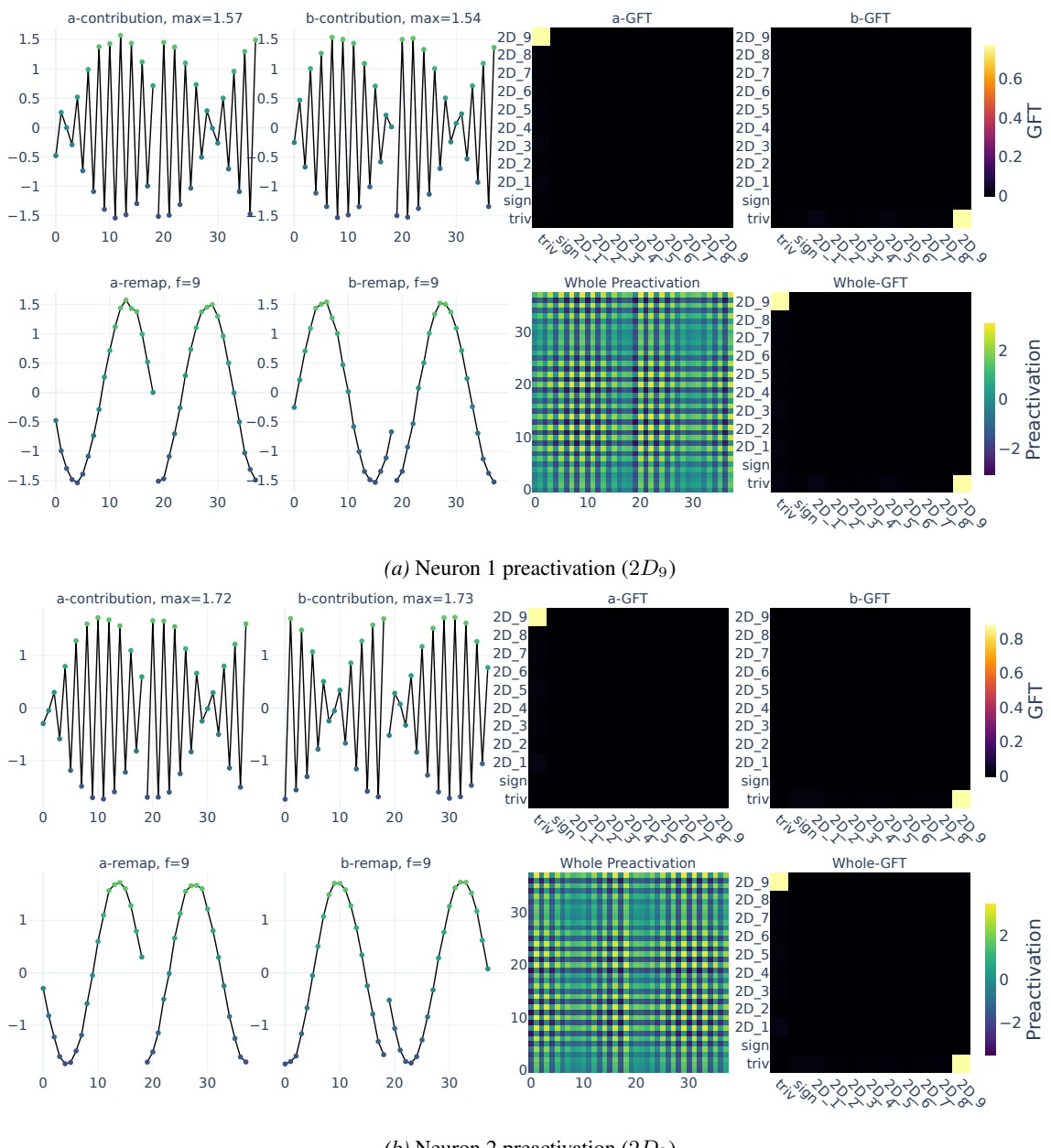

*(a)* Neuron 1 preactivation ($2D_9$)

*(b)* Neuron 2 preactivation ($2D_9$)

*Figure 41.* Visualization for $n = 19$, Fourier basis $2D_9$ (part 1/2). Two neurons with highest preactivation in $2D_9$.

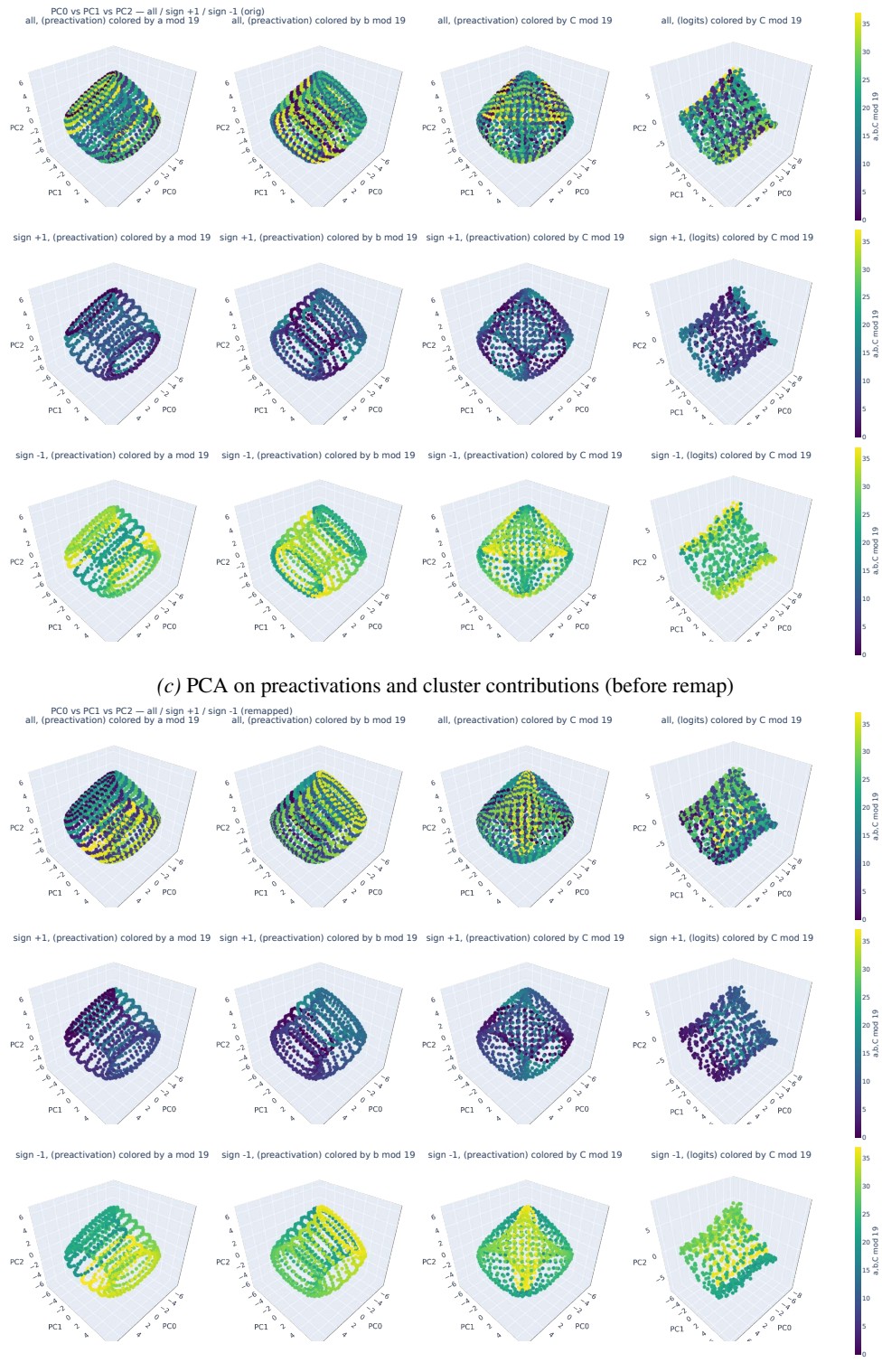

*(c)* PCA on preactivations and cluster contributions (before remap)

*(d)* Same as (c), after remap

*Figure 41.* Visualization for $n = 19$, Fourier basis $2D_9$ (part 2/2). PCA of preactivations and cluster contributions to logits, colored by $a$, $b$, and $C \bmod 19$, shown before and after remapping.

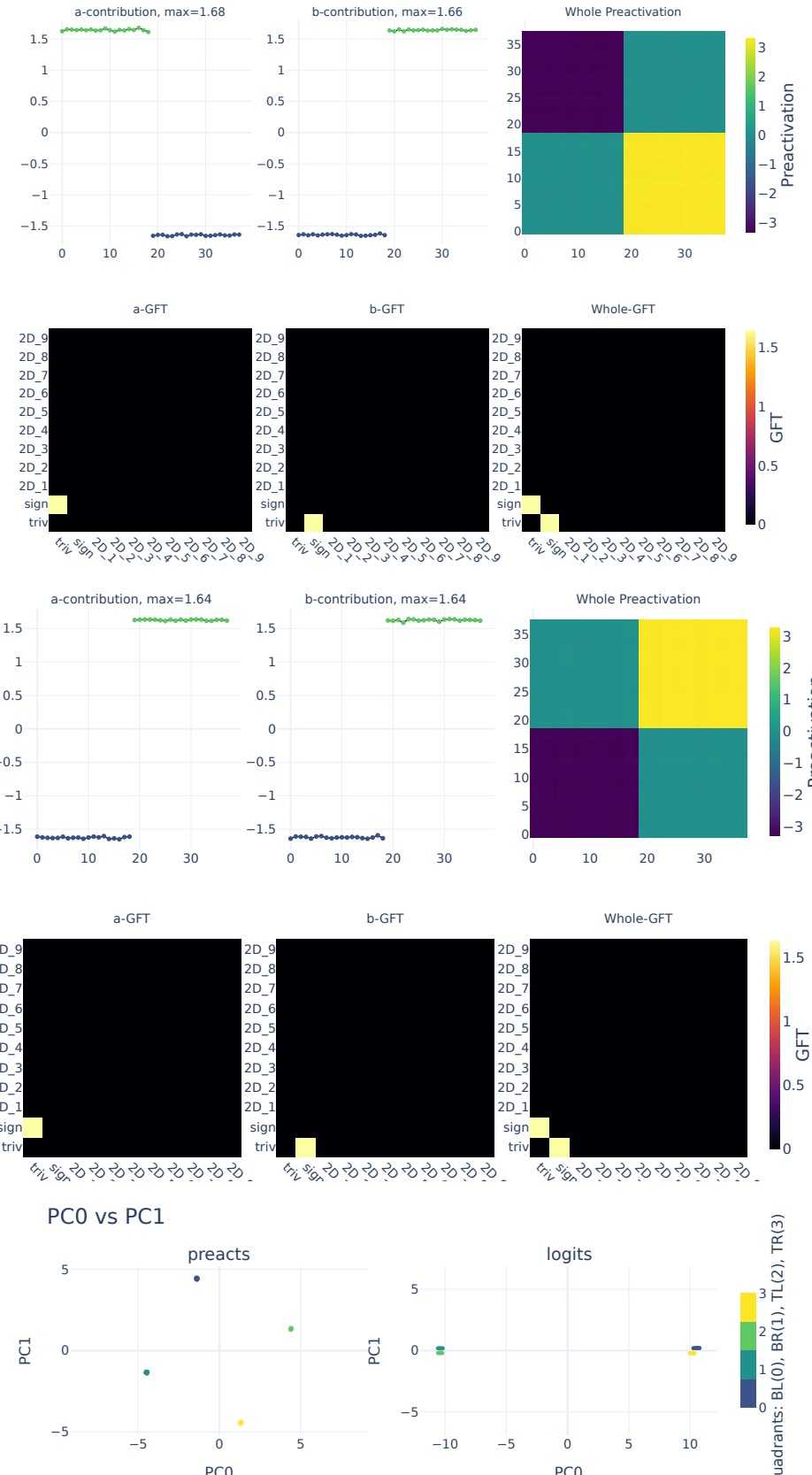

*Figure 42.* Visualization for $n = 19$. Top: two neurons with highest preactivation in Fourier basis `sign`. Bottom: PCA of preactivations and cluster contributions to logits, colored by quadrant (BL, BR, TL, TR) of $(a, b)$ in the $2n \times 2n$ grid.

## D.4. Transformer Cayley graph generator distribution

See Figure 43.

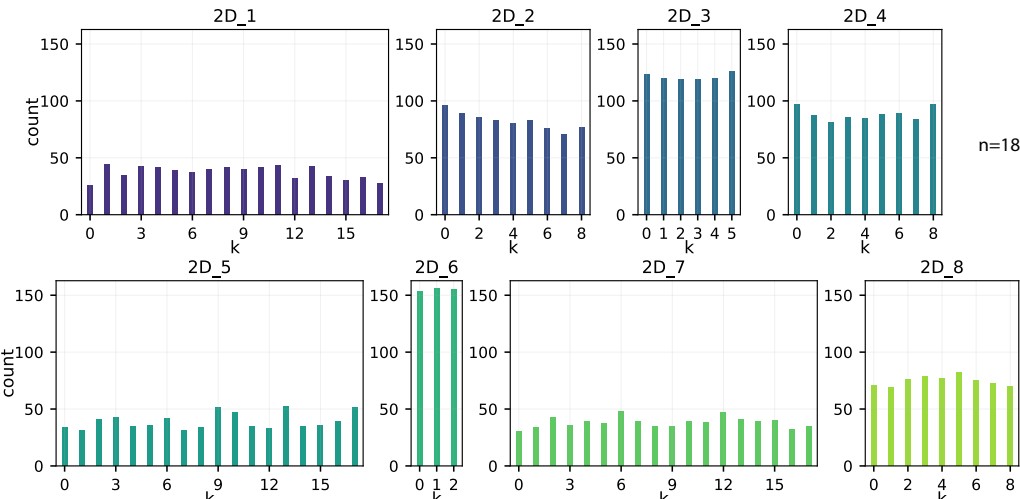

*Figure 43.* Distributions of how often each generator ($sr^k$) is used for each Cayley-graph type in the transformer. All distributions are approximately uniform, mirroring the behavior observed in the MLP. Representations of dimension less than 3 are filtered out.

## D.5. How were the plots made

For the neuron/PCA visualization plots in Appendix D.2- D.3, unless otherwise stated, we use a 90%/10% train/test split, and train with Adam (Kingma & Ba, 2014), cross-entropy loss, with learning rate 0.001 and batch size $= 2n$ for 5000 epochs.

## D.6. Additional algorithm for recovering Cayley graph generators from PCA space

Figure 44 illustrates the geometric pairing rule used to infer the Cayley graph generators from the preactivation PCA space. Algorithm 2 gives the greedy procedure used in the logit-contribution PCA space. The rotational generator is determined by the step size along each cyclic family, whereas the reflection generator is recovered by the pairing procedure described below.

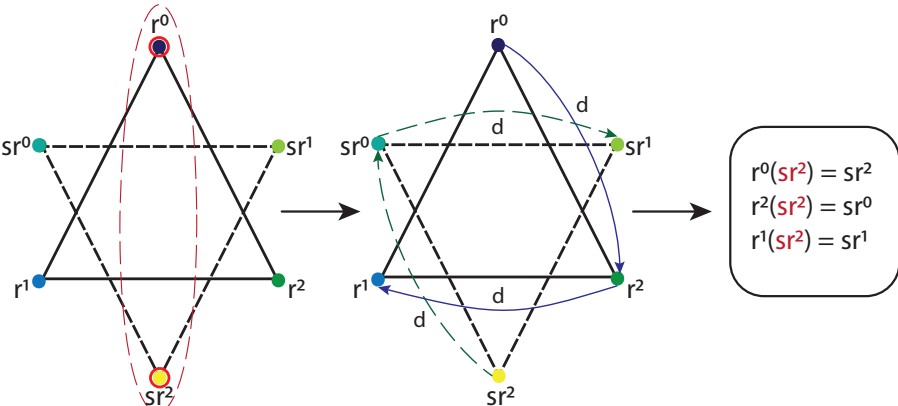

*Figure 44.* **Clockwise pairing on a stripe** ($g = 3$). Points are shown in a two-dimensional embedding, schematically corresponding to the PCA space. Each hexagram represents one fixed-$b$ stripe. The blue triangle denotes the rotation family $\{r^0, r^1, r^2\}$, indexed by $a$, while the green/yellow triangle denotes the reflection family $\{sr^0, sr^1, sr^2\}$. *Left:* we select the farthest cross-family pair as a shared reference point, here $r^0$ and $sr^2$. *Middle:* starting from this reference pair, we traverse both cyclic families clockwise using the same modular step size $d$, pairing the elements encountered at the same traversal time. The labels "$d$" indicate the step size in cyclic index space, rather than Euclidean distance. *Right:* in this example, the resulting pairs obey a stripe-wise constant-sum rule $i + j \equiv k \pmod{g}$, where $i \in \{0, 1, 2\}$ indexes the rotation elements $r^i$, $j \in \{0, 1, 2\}$ indexes the reflection elements $sr^j$, and the constant $k$ identifies the reflection generator $sr^k$. Here $k = 2$; for example, $r^0(sr^2) = sr^2$, $r^2(sr^2) = sr^0$, and $r^1(sr^2) = sr^1$.

---

**Algorithm 2** Greedy extraction of the reflection generator from logit-contribution PCA space

---

**Require:** Logit-contribution PCA coordinates, labels $(a, b)$, modulus $n$, frequency $f$

**Ensure:** Estimated reflection generator $\hat{k}$

1: Set the effective modulus $g = n/\gcd(n, |f|)$ and form residue–quadrant clusters modulo $g$.
2: Represent each cluster by a single prototype point, e.g., its centroid or nearest-to-centroid point.
3: Initialize an empty set of matched cluster pairs $\mathcal{M}$.
4: **for** each residue class **do**
5:     **for** each prescribed cross-quadrant pair **do**
6:         Compute all valid cross-quadrant distances between clusters in PCA space.
7:         Greedily match clusters in increasing order of distance, enforcing one-to-one assignments.
8:         Add the selected matches to $\mathcal{M}$.
9:     **end for**
10: **end for**
11: Convert each matched pair in $\mathcal{M}$ into a candidate reflection generator using the corresponding labels modulo $g$.
12: Return the modal candidate generator across all matched pairs.

---

*Table 7.* Generator agreement rates among phase-based inference, preactivation-PCA inference, and greedy logit-contribution-PCA inference for $n = 18$ under cosets across 500 seeds.

| Comparison | Match (%) |
|---|---|
| Phase-based inference vs. preactivation-PCA inference | 97.84 |
| Phase-based inference vs. greedy logit-contribution-PCA inference | 96.05 |
| Preactivation-PCA inference vs. greedy logit-contribution-PCA inference | 94.79 |

### D.7. Poor local minima—degenerate phase configurations

If the phase configurations of the sinusoidal functions learned by neurons aren't sufficiently diverse, the Cayley graph learned by the cluster of neurons effectively retains only the rotation generator. Thus, such a neural representation can't tell if the answer to a multiplication is in the rotation or reflection (sign +1 or sign -1) part of the dihedral group.

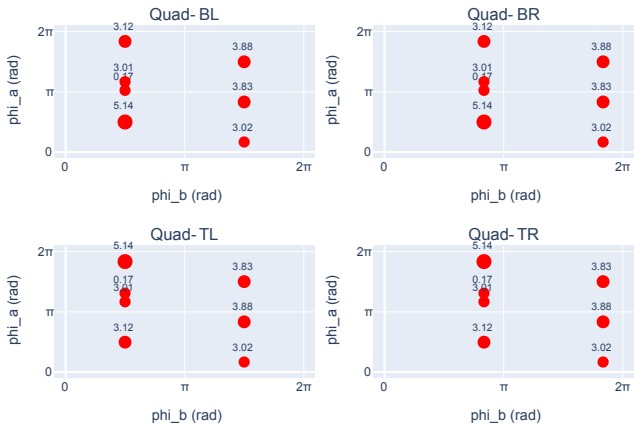

*Figure 45.* Here we see that the phases for this cluster of neurons in the top left (TL), bottom left (BL), bottom right (BR) and top right (TR) quadrants are aligned on two lines in every quadrant that the neurons activate on. In the case of non-degenerate neuron clusters, the phases appear to have something close to a diverse distribution over the grid.

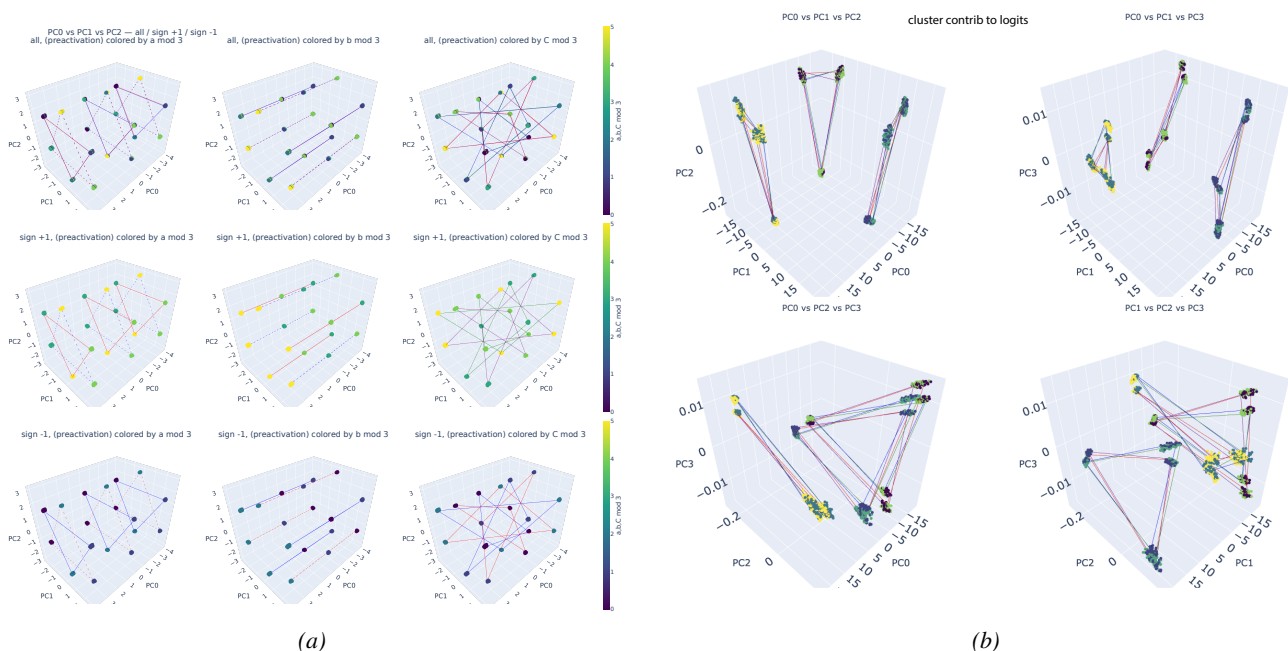

*(a)*                                    *(b)*

*Figure 46.* Degenerate solution for $n = 18$ and $f = 6$. **(a)** The representation contains only three $D_3$ structures, rather than the six observed in the main setting. This reduction is caused by the poor phase distribution shown in Figure 45. **(b)** The logits PCA shows a neuron cluster that fails to distinguish rotation and reflection equivalence classes. Each triangle has two equivalence classes of the correct answer $C$ on its vertices, indicating that this cluster does not separate answers with sign $+1$ from those with sign $-1$. This corresponds to a poorer local minimum, since a solution that distinguishes the two signs would further reduce the cross-entropy loss.

### D.8. Goodness of Fit for Length Extension

Here, length extension refers to training/evaluating models on products of more than two group elements, e.g., predicting $a_1 \cdot a_2 \cdot a_3$, $a_1 \cdot a_2 \cdot a_3 \cdot a_4$, or $a_1 \cdot a_2 \cdot a_3 \cdot a_4 \cdot a_5$.

*Table 8.* Mean $R^2$ averaged over 10 random seeds for different extended lengths using 2-layer MLPs and transformers ($n = 6$).

| Metric | MLPs | | | Transformers | | |
|---|---|---|---|---|---|---|
| | 3-variable | 4-variable | 5-variable | 3-variable | 4-variable | 5-variable |
| $R^2$ | 0.9997 | 0.9931 | 0.9987 | 0.9998 | 0.9945 | 0.9985 |

