# OpenReview forum: "Deep neural networks divide and conquer dihedral multiplication"
_ICML.cc/2026/Conference — ICML 2026 regular_

### Official Review · Reviewer_aqfb · 2026-02-26

**Soundness:** 3
**Presentation:** 3
**Significance:** 3
**Originality:** 3
**Overall Recommendation:** 4
**Confidence:** 2

**Summary:**

This paper makes a contribution toward the goal of universality hypothesis - deep neural networks trained on different, related datasets will use similar algorithmic strategies to solve the same problem. The authors consider specific multilayer perceptrons and transformers to solve dihedral multiplication and claim that they all essentially use the same divide and conquer algorithm that requires a logarithmic number of neural representations. Each neural representation corresponds to a Cayley graph. All neural representations are characterized at three levels of abstraction for dihedral multiplication (neuron; neural representation; and global algorithm). No theoretical basis is provided.

**Compliance With Llm Reviewing Policy:**

Affirmed.

**Final Justification:**

I am more or less satisfied with the answers by the authors.

**Key Questions For Authors:**

I believe I have a reasonable feel of the thorough analysis performed for dihedral multiplication on the architectures considered. What is the theoretical basis for the claims made in the paper? How about other architectures?

**Limitations:**

Needs theoretical justification of the claims made.

**Strengths And Weaknesses:**

Strengths:

This paper addresses a fundamental problem in deep learning and claims to present a full characterization of the algorithm networks universally learn in the simple case of dihedral multiplication.

This paper is the first work to characterize the classes of neural representations that can be learned across random seeds and architectures.

Weaknesses

The theoretical justification of the results are lacking. The whole methodology seems to depend on empirical results conducted on one or two layer MLP, and one and two-layer transformers.

---

> ### Author Rebuttal · Authors · 2026-03-31
>
> Thank you for your review and your feedback. We’re glad you recognize that our work *“addresses a fundamental problem in deep learning”* and is the first work to present a full characterization of the algorithm deep neural networks consistently learn across architectures, hyperaparameters, and seeds.
>
> Indeed, our work is empirical in nature, and we have made no claim to have theoretical guarantees. However, this would be a tall order: our results generalize across complex models, and we are demonstrating a remarkably consistent characterization of the local minima reached across these architectures. Proving theoretically which local minima will be reached for such highly nonlinear systems would be exceptionally challenging, and far beyond what has been achieved in the literature so far.
> Our argument is that, based on our empirical findings, theoreticians may have more guidance that may help prove such results in the future.
>
> > The whole methodology seems to depend on empirical results conducted on one or two layer MLP, and one and two-layer transformers.
>
> In the appendix, Figures 12-15 show how universal and robust our solution is. We sweep across hyperparameters, depths, widths, activation functions and architectures including MLPs and transformers (plus RNNs, see below). In all cases where hyperparameters are conducive to learning, our solution is learned. We’ve since extended our results to also cover RNNs (see https://anonymous.4open.science/r/fig-8659/rnn.pdf for heatmaps and table in the RNN case).

---

> > ### Author Rebuttal · Reviewer_aqfb · 2026-04-01
> >
> > I am more or less satisfied with the answers provided but still wish there is a way to provide a bit of a theoretical basis for the work.

---

### Official Review · Reviewer_JAKU · 2026-03-01

**Soundness:** 3
**Presentation:** 3
**Significance:** 3
**Originality:** 3
**Overall Recommendation:** 5
**Confidence:** 3

**Summary:**

This paper takes a significant step toward testing and refining the universality hypothesis in deep learning by examining whether different neural architectures learn shared algorithmic mechanisms when trained on related structured tasks​. The authors study how MLPs and transformers learn dihedral group multiplication an show that networks consistently develop a divide‑and‑conquer algorithm: individual neurons learn (approximate) coset structures, clusters of neurons form interpretable Cayley‑graph‑shaped neural representations, and the model solves the task by combining $\mathcal O(\log n)$ such representations. The work provides a complete characterization at the neuron, representation, and algorithmic levels, revealing that the learned clusters correspond to minimal Cayley graphs generated by the group’s rotation and reflection operations. Experiments over 1000 seeds demonstrate strong universality: networks reliably recover the same generators, prefer smaller subproblems and assemble these into a final algorithm that isolates the correct output logit.

**Compliance With Llm Reviewing Policy:**

Affirmed.

**Final Justification:**

My initial concerns centered on the lack of theoretical guarantees, limited architectural diversity, and the broader significance of studying group multiplication as a proxy for real-world tasks. The rebuttal addressed these points convincingly. In particular, the authors clarified the intended role of the work as a foundation for future theoretical results, provided additional empirical evidence demonstrating robustness across architectures, and offered a clearer motivation connecting group multiplication to broader algorithmic problems such as permutation-based tasks. While the results remain primarily empirical and the broader generalization is still an open question, the rebuttal improved my confidence in both the robustness and relevance of the findings.

Overall, I weigh the strengths (clear empirical evidence, strong interpretability insights, and a coherent conceptual contribution) above the remaining limitations, which are mainly about scope and future theoretical development rather than correctness. The rebuttal directly addressed my main concerns and positively changed my evaluation. I therefore support acceptance.

**Key Questions For Authors:**

# Questions
## 1. Theoretical guarantees
Your results are entirely empirical. Can you clarify which parts of your mechanism you believe could be formally proven and which you see as empirical regularities. A clear argument would strengthen confidence in the generality and impact of the work, with a potential increase of the soundness, significance and originality score
## 2. Architectural robustness
Shallow 1-2-layered MLPs/transformers have been evaluated. Do deeper networks, sparse architectures, or varied nonlinearities still recover the same Cayley‑graph structures and generator patterns. Evidence that the mechanism persists across more architectures would substantially improve the evaluation of robustness and increase my opinion on soundness and significance
## 3. Motivation for group multiplication
I understand that current state-of-the-art research investigates group multiplication for answering the universality hypothesis and mechanistic interpretability. For me as a non-expert, I am lacking a clear motivation on why insights on these problems should generalize to real-world cases. A convincing answer would raise my score on significance
# Remarks
- "Submission and Formatting Instructions for ICML 2026" as running title
- Write "First", or use 1.) for visible clarity (lines 15 and 65)
- "divide-and-conquer" (lines 33, 61, 378)
- "is depicted in" (line 97)
- $DC$ component? (line 105)
- $2D_k$? (line 55)
- "nor in a related setting"? (line 111)
- "cluster$f$"? (150)
- Paragraph headline "2. Coset disjointness" ended with "." while earlier paragraph headline ends with ":" (line 262)
- "neuron-based" (line 273)
- ")" missing (line 315)
- Universality Hypothesis is italic and capital, while it was not earlier in introduction (line 83)

**Limitations:**

yes

**Strengths And Weaknesses:**

# Strengths
## Soundness
- The paper provides a complete characterization of what the network learns at the neuron, representation, and algorithmic levels. The authors ground each claim in explicit mathematical tools, including group Fourier transforms, neuron‑cluster PCA geometry, and generator recovery via phase‑relation analysis
- The experiments are repeated over 1000 random seeds for both MLPs and transformers, demonstrating that the learned structures (cosets, Cayley‑graph representations, and the final divide‑and‑conquer mechanism) arise consistently and not by coincidence or architectural quirks
- First‑layer neuron preactivations are extremely well‑explained by coupled sinusoidal models, validating the theoretical interpretation that neurons correspond to Fourier components aligned with dihedral irreps. This precision supports the paper’s core claim that neuron behavior is structured, interpretable, and mathematically grounded
- Although the insights are derived from experimental analyses on small toy group‑multiplication tasks, the discussion section directly addresses the possible implications for answering different important hypothesis (universality, manifold, platonic representation) and highlights open questions and limitations
## Presentation
- The paper is well-written and well‑structured around three interpretability layers: neurons, neural representations, and the global algorithm. This makes the progression of ideas intuitive and highlights where each technical insight contributes
- Figures (e.g., PCA embeddings of clusters, Cayley‑graph layouts) are informative and help concretize abstract algebraic concepts by directly visualizing coset structures, equivalence classes, and generator actions in learned representations
- The paper situates its contributions within the history of conflicting claims about universality in group‑multiplication tasks, then shows how its results resolve previous contradictions and unify the emerging picture
## Significance
- The work unifies previously conflicting claims from studies on cyclic, permutation, and modular‑addition groups, showing that approximate‑coset‑based mechanisms generalize to non‑commutative dihedral groups as well. This strengthens the broader universality hypothesis and meaningfully extends its empirical foundation
- Demonstrating that neural representations correspond to minimal Cayley graphs offers a mathematically grounded lens for analyzing learned representations in other settings
- The finding that networks systematically implement an O(log n) divide‑and‑conquer algorithm via multiple Cayley‑graph representations is significant and theoretically generalizing such a result could let to deep insights on the efficiency of deep learning
## Originality
- The paper claims to be the first to fully describe every neural representation that a DNN learns when trained on a specific task
- The identification that neuron clusters reliably form PCA manifolds matching minimal Cayley graphs is an original conceptual bridge between group theory and mechanistic interpretability
- The finding that networks universally learn O(log n) Cayley‑graph representations to solve group multiplication is new
# Weaknesses
## Soundness
- While the experiments are extensive, there are no formal proofs that the recovered Cayley‑graph structures, coset behaviors, or the $\mathcal O(\log n)$ divide‑and‑conquer algorithm are necessary or guaranteed to arise in broader settings
- The work focuses on shallow 1-2‑layer MLPs and transformers with fixed design choices. It does not test whether the same mechanisms persist under alternative architectures (e.g., deeper networks, sparse connectivity, different activation functions) limiting claims of robustness across design spaces
## Presentation
- While the introduction clearly states the contributions, it does not explain why the dihedral group is an important test case and that tools from group Fourier transforms are used, making the motivation less immediately transparent
- Although the paper is self‑contained, the background section is dense and assumes familiarity with topics like the Group Fourier Transform (GFT) and cosets. Concepts like irreps, sign conventions, and frequencies are introduced in rapid succession, which may overwhelm readers unfamiliar with harmonic analysis on groups
- The manuscript frequently presents complex mathematical expressions directly in the text, which makes sentences visually dense and harder to parse
## Significance
- The clear emergence of cosets, Cayley graphs, and generator actions is partly due to the strict symmetries of dihedral groups. It is uncertain whether such clean neuron‑cluster geometries will be recoverable in tasks lacking exact group structure, limiting the broader interpretability significance
- Although the analysis is deep and rigorous, the results apply only to small structured group‑multiplication tasks

Even though I am not an expert in the field, this paper provides a highly-detailed and exhaustive mechanistic analyses of learning dihedral multiplication and seems to be an intriguing step towards obtaining insights on the universality hypothesis. The work is empirically robust, clearly structured across multiple abstraction levels, and offers valuable contributions to the ongoing discussion surrounding universality and interpretability in deep learning. I am very much open to increase my score if the authors answer questions on possible theoretical guarantees, architectural robustness and the overall motivation of analyzing group multiplication (see Key Questions).

---

> ### Author Rebuttal · Authors · 2026-03-31
>
> Thank you for your comprehensive review.
>
> **Q1. Theoretical guarantees**
>
> The reason that we went to lengths to provide the full empirical story of what DNNs learn on dihedral multiplication is because we believe all parts of the story can be proven theoretically, and that future work should expend the effort to do so.
>
> We believe that DNNs will universally learn group multiplications with a logarithmic number of minimal Cayley graphs that are appropriately intersected at the logits to select the correct answer. A general proof of this across all groups implies that on a massive space of problems: DNNs trained with SGD learn not only divide-and-conquer algorithms, but with logarithmic neural representation efficiency.
>
> Stander et al.’s finding that neurons learn coset structure, isn’t enough to imply neural representations are Cayley graphs and valid solutions exist without Cayley graph representations. For example, a network could learn neural representations that are partial Cayley graphs (only some of the edges and vertices are learned). These solutions have enough coverage across the many “partial graphs” to compute all correct answers, but require many more neural representations than logarithmic. The fact SGD never learns these partial Cayley graphs, and always “couples” enough neurons together to cover each graph is remarkable and is a hint about the type of solutions SGD is biased toward finding.
>
> A proof that SGD finds the log(n) intersecting Cayley graph solution for any group, would be a monumental step toward classifying the types of solutions learned by SGD.
>
> To summarize, our goal is to guide theoreticians, by showing them how DNNs universally learn the task, toward proving the above statements (first for dihedral groups, and eventually for all groups). To be clear, we believe future work will theoretically prove that the O(log(n)) Cayley graph neural representation algorithm is a valid local minima, and that networks converge to it when trained via SGD.
>
> **Q2. Architectural robustness**
>
> In the appendix, Figures 12-15 show how universal and robust our solution is. We sweep across hyperparameters, depths, widths, activation functions and architectures including MLPs and transformers (plus RNNs, see below). In all cases where hyperparameters are conducive to learning, our solution is learned.
>
> As far as studying deeper MLPs and transformers go, we investigated them and saw the same result Moisescu-Pareja et al. reported on modular addition, where the network preserves the representation from layer 2 in later layers, until finally projecting it to the logits. In essence, the computation is finished after the first non-linearity. We will add plots showing that the deeper neural representations also align with our solution.
>
> Your question inspired us to check if RNNs learn our solution: we trained 1k seeds of RNNs finding they learn our solution (see https://anonymous.4open.science/r/fig-8659/rnn.pdf for heatmaps and table in the RNN case). We believe our results over different training conditions and architectures convincingly show SGD has a bias to learn Cayley graph neural representations.
>
> **Table: Results for RNNs using the same settings as in the main paper.**
>
> | $s_{\text{cluster}}$ | coset disjointness | $R^2$            |
> |----------------------|--------------------|------------------|
> | $0.9968 \pm 0.0032$  | $0.9999 \pm 0.0009$| $0.9934 \pm 0.0078$ |
>
>
> **Q3. Motivation for group multiplication**
>
> Many problems can be framed as group multiplication. For example, the problem of sorting a list can be framed as determining which element in a permutation group permutes a list into sorted order. Thus, a proof that DNNs learn universal divide-and-conquer solutions across all groups would explain a lot about how neural networks solve a broad class of algorithmic tasks.
>
> Indeed, this doesn’t transfer to all possible datasets, as many tasks aren’t group multiplications. It’s the case however that group multiplications have opened a window into understanding “the unreasonable effectiveness of deep learning”, by showing that at least in the case of some groups networks learn very efficient divide-and-conquer algorithms. The overall goal of the field is to continually increase the difficulty of synthetic tasks in the hope that universal principles are uncovered that will help guide interpretability and AI safety work with natural data.
>
>
> **References:**
>
> Stander et al. "Grokking group multiplication with cosets". ICML 2024.
>
> Moisescu-Pareja et al). "On the geometry and topology of representations: The manifolds of modular addition". ICLR 2026.

---

> > ### Author Rebuttal · Reviewer_JAKU · 2026-04-03
> >
> > Thank you for the detailed and thoughtful rebuttal. The clarifications regarding the theoretical perspective are particularly helpful. While the results remain empirical, I now better understand the intended role of this work as a foundation for future formal analysis and as a guide toward proving stronger statements about SGD’s inductive biases.
> >
> > I also appreciate the additional evidence on architectural robustness, including the extended sweeps and the new RNN experiments, which increases my confidence that the observed mechanism is not an artifact of specific design choices. Finally, the motivation connecting group multiplication to broader algorithmic tasks (e.g., permutations and sorting) provides a clearer picture of why this line of work is relevant beyond toy settings, even if full generalization remains an open question.
> >
> > Overall, the rebuttal successfully addresses my main concerns and strengthens both the interpretation and robustness of the results. I will therefore raise my score accordingly.

---

### Official Review · Reviewer_ctyR · 2026-03-13

**Soundness:** 3
**Presentation:** 3
**Significance:** 2
**Originality:** 3
**Overall Recommendation:** 3
**Confidence:** 3

**Summary:**

The authors investigate whether multilayer perceptrons and transformers learn universal representations when solving dihedral multiplication, whose underlying structure combines a cyclic rotation component of order $n$ with a binary flip component of order 2. The authors empirically show that MLPs and transformers trained on dihedral multiplication learn neuron-level approximate/coset structure, cluster-level Cayley-graph representations, and a global divide-and-conquer strategy using only $O(\log n)$ neural representations.

**Compliance With Llm Reviewing Policy:**

Affirmed.

**Key Questions For Authors:**

How much of the reported coset/Cayley-graph structure is specific to neural-network representations, versus being intrinsic to dihedral multiplication itself? -- is it possible to perform such analysis by looking into representations learnt by other models, like decision trees for the same task?

I am slightly confused about the role of GFT in the analysis. I am not an expert in this direction, but it seems to me that GFT creates basis for the group structure of dihedral multiplication, and then using it as method to group neurons may create confirmation bias (potentially similar to the cases authors discuss in the Introduction)? I will be interested in understanding that why is this not the case? can similar results be obtained using some other more uninformative clustering choice?

**Limitations:**

Yes

**Strengths And Weaknesses:**

Strengths:
-  For dihedral multiplication, the paper provides identification of neuron-level coset structure, cluster-level Cayley-graph representations, and a global divide-and-conquer interpretation, giving a complete mechanistic analysis.

- Extensive and thorough empirical evaluation.

- The main take away on MLPs and transformers learning an efficient strategy for a simple task, where potentially memorization can work equally well, is a very interesting insight.

Weakness:
- My main concern is that how canonical/universal are the learnt representations to NNs vs how universal are they to the task itself?
- The scope of the paper is limited to an extremely toy setting

---

> ### Author Rebuttal · Authors · 2026-03-31
>
> Thanks for your review. We’re glad you found that MLPs and transformers learn “an efficient strategy for a simple task” to be “a very interesting insight.” We see you raised three points: the scope of the contribution; whether it’s possible for networks to learn solutions without Cayley graphs; and if using the Group Fourier Transform (GFT) to collect neurons into neural representations could bias/force the results to be Cayley graphs.
>
> **1. Scope of the paper**
>
> Dihedral multiplication is a task that some prior work investigated and others suggest for future work. This interest is because there's a plethora of publications about modular addition (the cyclic group multiplication) which is Abelian (commutative). Naturally, commutative group multiplications aren’t technically challenging because the order of actions doesn’t matter (commutative), the agent can do nothing (add 0), and actions are reversible (you can add 5, or subtract 5 to reverse it).
>
> The dihedral multiplication is a natural step up in difficulty from modular addition: it's a task where the order of actions matters (not commutative), but still involves cyclic groups as base components. Thus, it provides an opportunity to test the universality hypothesis across datasets that have something in common (cyclic groups), but very different geometric structures.
>
> Thus, Chughtai et al. investigated universality between dihedral and cyclic group multiplications. At the time, the tools we used weren’t being used and so they used their observation that the GFT was concentrating on irreducible representations of the group, which are matrices, to claim DNNs learn group multiplications by performing matrix multiplications on irreducible representations. This is a bit of a jump because the GFT only tells you functions on the group have been learned using that irreducible representation (basis). It doesn’t mean the matrix itself was learned.
>
> Our work revisits dihedral multiplication with new methods and shows quantitatively and qualitatively that 3 levels of abstraction (neurons, neural representations, and globally) all imply the DNNs learned a divide-and-conquer algorithm on Cayley graphs, not Chughtai et al.’s claimed matrix multiplication algorithm.
>
> **2. How canonical/universal are the learnt representations to NNs vs how universal are they to the task itself?**
>
> These representations are learned in all architectures tested but are not the only way to solve dihedral multiplication. For example, Chughtai et al.’s algorithm requires DNNs to learn the irreducible representation matrices in the embeddings and weights and combine the values such that matrix multiplication is performed computing the answer.
>
> Additionally, Stander et al.’s finding, neurons learn coset structure, isn’t enough to imply neural representations are Cayley graphs. Valid solutions exist without Cayley graph structure. A network could learn neural representations that are partial Cayley graphs (only some of the vertices). This solution would have enough coverage across the many “partial graphs” to compute all correct answers, but would require many more neural representations.
>
> This is where our result contributes: we show DNNs learn remarkably structured solutions: O(log(n)) different Cayley graphs, each graph distributed over multiple neurons and each neural representation is a full graph – not parts of it. Our solution explains every neuron in the network and how they collaborate to form neural representations to compute the answer.
>
> We believe this finding is significant: from an interpretability standpoint, SGD is learning seemingly the cleanest neural representations possible. Since we can characterize so precisely, and this is a math dataset (not natural data), future theory can hope to prove why this solution is found, gaining insight into how/why/if SGD has a bias towards structured solutions.
>
> **3. Possible confirmation bias in GFT.**
>
> We actually demonstrated our GFT methodology doesn’t have a bias of showing Cayley Graphs in the appendix. We did this by training networks (MLPs + transformers) on a random multiplication task, where the answer to (a x b) is randomly chosen instead of the dihedral multiplication. After generating the dataset, we train, then apply our methodology, grouping neurons based on similar GFTs. The result is a noisy cloud of points – not a highly structured Cayley graph (see Fig. 10). This shows composing a neural representation with our methodology doesn’t “force” Cayley graph structure to be found. Indeed, we confirm this quantitatively in Figure 11 over 1000 seeds, showing neural representations from random multiplication tasks have poor coset clustering scores and poor disjointness, implying these structures aren’t learned.
>
> **References:**
>
> Chughtai et al. "A toy model of universality: Reverse engineering how networks learn group operations". ICML 2023.
>
> Stander et al. "Grokking group multiplication with cosets". ICML 2024.

---

> > ### Author Rebuttal · Reviewer_ctyR · 2026-04-03
> >
> > I remain concerned about the GFT-based methodology.
> >
> > Q1. Is the GFT analysis fundamentally task-informed by the dihedral group structure? I think this to be the case right now.
> >
> > Q2. Does Appendix B.5 only show that the same GFT pipeline stops finding this structure once the task is no longer true dihedral multiplication? -- This is good, but it only shows that GFT is not suited for other tasks. But it does not ameliorate my concern that GFT is already aligned to find these universal representations, and any model that solves dihedral multiplication will admit some sort of universal representations if seen through a GFT lens.
> >
> > Q3. Do you show any case where dihedral multiplication is solved successfully, but the GFT analysis does not recover the claimed universal representations?
> >
> > At a high level, I worry that the paper may not yet fully rule out a methodological circularity similar in spirit to the interpretability issues it attributes to related work.

---

> > > ### Author Response · Authors · 2026-04-04
> > >
> > > Q1. Yes every group has its own irreducible representations, thus a specialized GFT exists for every group where the bases of the GFT are the irreducible representations.
> > >
> > > Q2. Yes B.5 shows our dihedral GFT pipeline doesn’t force universal Cayley graph representations to be found when they do not exist. Your interpretation is correct that in order for our analysis to yield fruit, it was necessary to know a priori that our network learns dihedral multiplication (giving which GFT to use, explaining why B.5 shows no structure – neurons were clustered with the “wrong” GFT). That said, analyzing networks this way still sheds light on universal properties of representations learned by neural networks via SGD. In particular, and explained more in our response to Q3, the Cayley graph representations that we formalize are not necessary for solving the task, yet we find all networks we trained do converge to circuits that employ these representations. This generalizes the interpretations of [1] and [2] on cyclic groups — the representations they find are direct analogues of the Cayley graph structure learned here for dihedral groups. While our analysis does not provide a universal technique revealing which group operation is learned by a network (again, you must know a priori what the task is), it provides evidence of a **non-trivial universal structure** that is learned by networks across group operations.
> > >
> > > Q3: This is an interesting question and we maintain that plausible solutions exist for networks to solve dihedral multiplication without learning Cayley graph representations, but we show that SGD doesn’t find these other solutions.
> > >
> > > It is important to distinguish between the GFT and Cayley graphs. The statement that the GFT of many neurons concentrates on the same irreducible representation is a spectral statement and does not imply the existence of Cayley graphs. The Cayley graphs come from an additional geometric statement, being that neurons concentrating on the same irreducible representation also all learned the same geometric structure, which is observed in the *neuron phases*. We report this in the main paper as a key result: for every neuron in the same neural representation, the sum of the $a$-rotation and $a$-reflection phases is the same constant and the difference in $b$-rotation and $b$-reflection phases is the same constant. Note this constant can be different in different neural representations. The fact that those same constants are learned by every neuron that concentrates on the same spectral basis is what results in the neurons having learned the same coordinate system, which distributes a Cayley graph over the cluster of neurons. This is seen as the neurons all having learned the same generators for the Cayley graph at an individual neuron level. Were these phase relations not learned, the PCA wouldn’t show clean Cayley graphs since neurons wouldn’t have learned the same generators.
> > >
> > > To contrast, consider [4]’s model that assumes networks learn to multiply the irreducible representation matrices $\rho(a) \rho(b) = \rho(ab)$ with fixed irreducible representation $\rho$ of the dihedral group. Networks implementing this need not learn coupled sinusoids with linked phases (which we show are learned across seeds and architectures with a consistently high R^2 of 0.9988). Instead, it can learn coordinates of the representation matrices, or arbitrary linear projections of those coordinates that lack shared phase alignment. That matters because those coordinates do not force the phase relation we universally observe to be learned. Thus [4]’s algorithm, while theoretically valid and representable by networks, our finding is that this algorithm is never learned, but rather the Cayley graph algorithm is always learned. This is what we mean by the universal property we show generalizes prior work in our Q2 response.
> > >
> > > We hope our responses clear up your concerns about circular reasoning. To summarize:
> > > 1. All our analyses show networks solving dihedral multiplication learn particular structures in neuron phases (encoding Cayley graphs).
> > > 2. This phase structure isn't required to solve dihedral multiplication; a counterexample is the model proposed by [4], which networks could plausibly learn, but if they did, clustering neurons with the GFT would select neural representations, but wouldn't show Cayley graphs since the required phase structure is missing.
> > > 3. This phase structure generalizes patterns found in prior work on other groups (cyclic groups [1,2], elementary p-groups [3])
> > > 4. Thus we provide evidence suggesting this phase structure is universally learned, restricting the space of possible solutions learned by networks solving group tasks trained by SGD.
> > >
> > > We appreciate the insightful comments and are happy to discuss any points in more detail.
> > >
> > > [1] https://arxiv.org/abs/2505.18266
> > > [2] https://arxiv.org/abs/2512.25060
> > > [3] https://openreview.net/forum?id=NTXCFymNWu
> > > [4] https://arxiv.org/abs/2302.03025

---

### Official Review · Reviewer_HXc8 · 2026-03-16

**Soundness:** 3
**Presentation:** 3
**Significance:** 3
**Originality:** 3
**Overall Recommendation:** 4
**Confidence:** 1

**Summary:**

This work analyses MLPs and transformers' behavior on dihedral group and finds that they learn the same divide and conquer strategy. By clustering neurons, this work finds clear structure in trained neural network and understand its full behavior rather than neural-level behavior only. It shows that neural network learns to solve dihedral multiplication problem by three step. 1. coset structure identification. 2. dividing into cayley graph 3. linear combine subproblem logits. This work provides a full explanation of mlp and transformer on a specific problem.

**Compliance With Llm Reviewing Policy:**

Affirmed.

**Final Justification:**

All my concerns are solved. I keep my score to 4.

**Key Questions For Authors:**

1. As shown in Table 1, transformer has larger score deviation than MLP. Could you please explain it?
2. Could you please show how to generalize the explanation method to real-world problem or a larger range of math problem?

**Limitations:**

yes

**Strengths And Weaknesses:**

Strength
1. Clear presentation. Background in section 2 is easy to understand. Figure 2 provides a comprehensive illustration of the interpretation method used.
2. Soundness. The claim is well supported, and experiments are conducted with 1000 repeats, and standard deviation is reported. These results make the experimental finding sound.
3. Significance. Full explanation of a neural network is rare. And the explanation tricks used (clustering score and disjointness) seem novel.

Weakness:
1. The toy problem they use limit the significance.
2. The explanation method seems only useful for a small range of problems. For the dihedral multiplication problem, cayley graph structure is already known to be important. But such structure is rare in real-world problem.

---

> ### Author Rebuttal · Authors · 2026-03-31
>
> Thanks for your thorough review and helpful feedback. We appreciate your recognition of the rigor of our empirical results, as well as the novelty and significance of our analysis techniques.
>
> **Scope of the problem domain.**
>
> As you suggest, our results apply only to dihedral multiplication. However, we argue that the explanation method (that is, our analysis technique), is much more broadly useful.
> For example, the recent works of McCracken et al. (2025) and Moisescu-Pareja et al. (2026) found similar patterns analyzing the simpler cyclic group multiplication task. Our work here provides a natural, yet nontrivial, generalization of these results to a broader class of group operations.
>
> As such, our work should not be interpreted as an end in itself; rather, it pushes towards a larger research program geared at interpreting circuits for general group operations. This can be highly influential, as many algorithmic tasks can be cast as group multiplication. One such example is the classical problem of sorting lists, which is equivalent to acting on a list with an element of the permutation group. Thus, further generalization of our methods to broader groups can indeed lead to strong mechanistic interpretability for complex algorithmic tasks, shedding light into the mechanics of the solutions stochastic gradient descent learns across a wide and diverse scope of problems.
>
>
> > Q1. As shown in Table 1, transformer has larger score deviation than MLP. Could you please explain it?
>
> This is a good question and it’s an interesting artifact for future work to explore. For the sake of this paper, we wanted to ensure transformers were learning Cayley graphs and our proposed solution. Moisescu-Pareja et al. inspected the manifolds learned within DNNs on the modular addition group multiplication and found that the neural representations in transformers were “noisier” than the ones learned in MLPs. This is consistent with our findings as well, as you pointed out.
>
> > Q2. Could you please show how to generalize the explanation method to real-world problem or a larger range of math problem?
>
> Yes. Sorting a list is a classical computer science problem and it would be interesting to see an in depth interpretation of how DNNs learn to do it. Sorting a list can be rewritten as a group multiplication within the permutation group. Given an unsorted list, the network must learn the permutation group element that when applied to the list, sorts the list. Thus, the problem of sorting lists is a group multiplication (indeed, libraries like numpy can return the permutation that would’ve sorted the original list when you do np.argsort). Our method would work on this problem, or any other group problem.
>
>
> **References:**
>
> McCracken et al. "Uncovering a universal abstract algorithm for modular addition in neural networks". NeurIPS 2025.
>
> Moisescu-Pareja et al. "On the geometry and topology of representations: The manifolds of modular addition". ICLR 2026.

---

> > ### Author Rebuttal · Reviewer_HXc8 · 2026-04-01
> >
> > All my concerns have been resolved. I will keep my score.

---

### Decision · Program_Chairs · 2026-04-30

**Decision:**

Accept (regular)

**Comment:**

The reviewers discussed the merits of the submission with the authors and the authors convinced them mostly about the strengths of the paper. The remaining negative review points out how the paper could be further strengthened. I would appreciate if the authors could take that into account when preparing future versions of the paper.